# Could seismo-volcanic catalogus be improved or created using weakly supervised approaches with pre-trained systems?

Manuel Titos[1,2], Carmen Benítez[1,2], Luca D'Aria[3], Milad Kowsari[4], and Jesús M. Ibáñez[5,6]

[1]Department of Signal processing, Telematics and Communications, University of Granada, Granada, 18014, Spain
[2]Research Center on Information and Communication Technologies of the University of Granada (CITIC-UGR)
[3]Volcanological Institute of the Canary Islands, Tenerife, 38400, Spain
[4]University of Iceland, Faculty of Civil and Environmental Engineering, Reykjavík, 102, Iceland
[5]Instituto Andaluz de Geofísica, University of Granada, Granada, 18071, Spain
[6]Department of Theoretical Physics and the Cosmos, University of Granada, Granada, 18071, Spain

**Abstract.** Real-time monitoring of volcano-seismic signals is complex. Typically, automatic systems are built by learning from large seismic catalogs, where each instance has a label indicating its source mechanism. However, building complete catalogs is difficult owing to the high cost of data-labelling. Current machine learning techniques have achieved great success in constructing predictive monitoring tools; however, catalog-based learning can introduce bias into the system. Here, we show that while monitoring systems trained on annotated data from seismic catalogs achieve performance of up to 90% in event recognition, other information describing volcanic behavior is not considered or either discarded. We found that weakly supervised learning approaches have the remarkable capability of simultaneously identifying unannotated seismic traces in the catalog and correcting misannotated seismic traces. When a system trained with a master dataset and catalog is used as a pseudo-labeller within the framework of weakly supervised learning, information related to volcanic dynamics can be revealed and updated. Our results offer the potential for developing more sophisticated semi-supervised models to increase the reliability of monitoring tools. For example, the use of more sophisticated pseudo-labelling techniques involving data from several catalogs could be tested. Ultimately, there is potential to develop universal monitoring tools able to consider unforeseen temporal changes in monitored signals at any volcano.

## 1 Introduction

Understanding the dynamics of active volcanoes and, even more so, carrying out Early Warning protocols for volcanic eruptions require multiparametric observations focused on accomplishing accurate and effective monitoring (Sparks, 2003). The objective of identifying precursors that warn of a possible volcanic eruption involves the analysis of long temporal series of data, characterizing and relating them with source models associated with the internal dynamics of the volcano (Witze, 2019; Palmer, 2020). Currently, the availability of multiparametric long-time data series, such as seismology, deformation, measure-

ments of volcanic gases and fluids, space imaging, and other processes, is limited to a few volcanoes around the world. For this reason, volcanic seismology continues being the backbone of the analysis, both in real time and using data from previous eruptive episodes (Chouet, 2003; McNutt, 2015). This is because the installation and acquisition of seismic data continues to be the most efficient procedure of volcanic monitoring, and because the existence of numerous open access repositories allows the scientific community reviewing consolidated databases to understand what occurred in the past for modelling future eruptions. In volcanic seismology, the presence of various seismic signals—such as volcano-tectonic earthquakes (VT), long-period events (LP), ultra-long-period (ULP) events, hybrid (HY) events, explosions (EXP), and volcanic tremors (TR)—indicates the existence of multiple seismic sources, which can sometimes operate simultaneously and must be considered. Thus, models of brittle rock fracturing, conduit resonance, pressure transients in fluids, bubbles, cracking in viscoelastic mediums, elastic energy transfer by fluid flow, debris flows, and many others are used (Ibáñez et al., 2000; McNutt and Roman, 2015; Minakami, 1974)) (Table 1 summarizes the source models and classifications for different authors). The complexity of seismic sources leads to varying interpretations of volcanic dynamics, influenced by the predominant signal type and its spatio-temporal evolution. Comprehending the underlying physics behind the eruptions, and thus understanding why they occur, cannot be solely explained through such signal processing. It requires knowledge of the frequency and types of seismic events that take place. This understanding is primarily gained by constructing seismic catalogs, which are then analyzed to infer volcanic dynamics in future crises. However, building complete catalogs presents significant challenges due to factors such as noisy signals, human error, intense seismic activity, and overlapping signals, all of which complicate the identification and classification of seismic events.

Historically, seismic catalogs have been manually created by experts, with the classification of seismic signals based on time-frequency characteristics and wave-field properties. The process relies heavily on expert knowledge, which, while essential, can introduces potential biases. These biases may arise from various factors, such as the prevailing scientific understanding at the time of labeling, or the occurrence of intense seismic activity where, due to time constraints, only the most energetic events are highlighted, or even when the energies are not high enough, overlapping signals are classified as a single event, leading to the combination of different types of signals under a single label. This issue was notably observed during the 2011 eruption on the island of El Hierro, where continuous VT events resulted in a high-frequency signal resembling volcanic tremor due to the overlap of hundreds of VTs per hour (Ibáñez et al., 2012; Díaz-Moreno et al., 2015). Despite the efforts made, such challenges remain widespread across seismic databases worldwide, highlighting the need for improved methods of signal classification and event labeling.

The introduction of automatic recognition procedures for earthquake-volcanic signals almost two decades ago (e.g. Ohrnberger 2001, Scarpetta et al., 2005; Alasonati et al., 2006; Benítez et al., 2006; Ibáñez et al., 2009, Curilem et al., 2009, Bhatti et al. 2016; Canario et al., 2020 ; Cortés et al., 2021; Bueno et al. (2021, 2022); Martínez et al. 2021; Titos et al. (2017, 2018, 2019), Bicego et al., 2022, etc) has made the process of identifying and characterizing signals more efficient, faster and comprehensive, allowing progress in both building robust catalogs and real-time monitoring of active volcanoes. However, the results obtained have begun to reveal potential problems: *monitoring systems loss effectiveness when recognizing events over time, which biases the construction of seismic catalogs and, in turn, affects experts' ability to analyze and understand volcanic*

| Ibáñez, J.M et al. (2000) | McNutt, S. and Roman, D. (2015) | Minakami, T. (1974) | Frequency [Hz] | Example source models |
|---|---|---|---|---|
| Volcano Tectonic Earthquakes Tectonic Short Period Earthq. | High Frequency (HF) | A-Type | >5 | Shear failure or slip along faults, usually as swarms within the volcanic edifice |
| Long Period Event Volcanic Long Coda Event Tornillo | Low Frequency (LF) | B-Type | 1-5 | Fluid driven cracks, pressurization processes (bubbles), and attenuated waves |
| Hybrid Event Medium Frequency | Mixed Frequency (MX) | - | 1-12 | Mixture of processes (e.g., cracks and fluids, frictional melting) |
| Explosion Volcanic Explosion | Explosion Quake (EXP) | Explosion Quake | >10 | Accelerated emissions of gas and debris to the atmosphere |
| Volcanic Tremor Harmonic Tremor | Volcanic Tremor (TRE) | Volcanic Tremor | 1-12 | Pressure disturbance, gas emissions, debris processes, and pyroclastic flows |

**Table 1.** Representative volcano-seismic scientific labels and associated source models proposed by Ibáñez, J.M. et al. (2000). Other labels and associated source models proposed by different authors have been included for comparison.

*dynamics.*

These outcomes raise open questions that should be efficiently addressed to adequately comprehend and solve such problems: a) Why do monitoring systems lose effectiveness? Could it be because volcanoes do not behave uniformly over time, displaying different unrest patterns from eruption to eruption and from one volcano to another? (b) Could it be that automatic monitoring systems show weakness due to seismic catalog-induced bias in their development? That is, is the database used during the development process properly labeled? Are the signal names or labels accurately identified? (c) Finally, how do seismic attenuation processes or source radiation patterns influence changes in the appearance of a signal, thus confounding the associated source models? How could background seismic noise affect the identification of seismic events?

For the last open question, it is well-know that seismic waves carry information not only on volcanic activity but also on the intricate internal structure of the volcanic edifice, which influences the seismic wave-field and complicates its interpretation (Titos et al. (2018)). At many volcanoes, rugged and pronounced topography introduces additional complexities, such as wave interference, high attenuation, and path alterations for direct seismic waves. Consequently, even for the same volcano and the same originating seismic source, recordings vary in shape and wave-field characteristics depending on seismometer placement. Furthermore, even at the same seismic station, similar sources may produce different signal patterns due to variations in the

source's energy radiation. These effects are broadly categorized into path-related (attenuation) and source-related (energy and radiation pattern) influences (Titos et al. (2018)). As a potential solution, experts propose using a network of multiple seismic stations for signal recognition and defining rules or conditions to identify signals simultaneously.

The first and second open questions may potentially be more difficult to resolve. Volcanic behavior is highly variable, exhibiting different signs of unrest between eruptions and between volcanoes. Environmental and geological factors, such as geology, magma composition, and the volcanic edifice, influence how seismic signals propagate and are recognized. This variability poses a challenge for automatic recognition systems, which are typically built by learning from large seismic catalogs, where each instance has a label indicating its source mechanism. The more diverse the data, the better the system's adaptability. However, as stated before, constructing complete catalogs is challenging because of the high cost of data labeling, which often leads to inaccuracies or mislabeling in seismic catalogs. Such inaccurate or mislabeled seismic catalogs could bias the effectiveness of the systems, meaning that their performance may be influenced not only by changes in volcanic dynamics, but also by inadequate modeling of those dynamics.

In this work, we propose a comprehensive analysis of seismic catalog-induced bias when developing automatic recognition systems. We evaluated the ability of several monitoring systems trained using a master seismic catalog from Deceptio Island volcano (referred to as the 'Master database') to adapt to new different volcanic environments from Popocatépetl (Mexico) and Tajogaite (Canary Island, Spain) volcanoes. We hypothesize that, often, automatic recognition systems are not capable of modeling the spatial-temporal evolution of seismic events. Instead, they learn to recognize the probabilistic pattern-matching observed in their training data. In other words, rather than simply learning to characterize volcanic dynamics by describing the latent physical model, catalog-induced learning biases the system's performance as it learns the description of the data annotated in the catalog, potentially discarding useful data that describes volcanic dynamics. Therefore, we conclude that using systems trained with a master database (complete and large) as pseudo-labeler, could help create less biased catalogs from which the systems can be retrained and adapted to different volcanic environments.

To test our hypothesis, we conduct three independent experiments with three different automatic monitoring systems. In the first experiment, aimed at demonstrating that any state-of-the-art machine learning model can effectively learn the information contained in a seismic catalog, we will build monitoring systems within the Transfer Learning framework. In this approach, systems that have previously been trained on Deception Island volcano, will be re-trained using a seismic catalog from the Popocatépetl volcano. Once trained, the models will be evaluated in terms of performance and analyzed in detail. The outcomes reveal a key issue: when the catalog is not meticulously constructed, and events are not accurately annotated—where multiple events are combined as a single label—the systems fail to recognize each individual event, leading to the loss of valuable data that describes volcanic dynamics. In the second experiment, instead of re-training the pre-trained systems using a given catalog, we use the pre-trained systems as a foundational seed (pseudo-labeler) for labeling the new database and construct a new catalogs. Using these new catalogs as training knowledge, we will re-train the systems. Afterwards, we will compare and analyze the results obtained from both approaches. The outcomes reveal that a significantly higher number of events, compared to those annotated in the original catalog, are recognized. This finding could offer a new potential perspective on the volcanic dynamics. Finally, to prove the robustness of our hypothesis, we will conduct a new experiment with data from

the eruption of Tajogaite volcano in 2021, for which only an earthquake catalog is available, demonstrating that the application of automatic seismo-volcanic monitoring systems based on weakly supervised techniques can offer an effective alternative for both building and revising seismic catalogs.

The rest of this paper is organized as follows. Section II describes the seismic dataset and signals used in this study. . Section III provides the experimental framework, and describes how weakly supervised techniques can be used for developing automatic volano-seismic recognition systems. Section IV and V presents the results and discussions. Section VI concludes this paper.

## 2    Seismic data and catalogs

As previously stated, in this study, we will use three datasets from three volcanoes of different nature: Deception Island (Antarctica), Popocatépetl (Mexico) and Tajogaite (Canary Island, Spain). Due to the extensive expertise and in-depth knowledge that our research group has on Deception Island volcano, providing a comprehensive understanding of its structure and dynamics through numerous campaigns conducted since 1994 (Ibáñez et al., 2000; Martínez-Arévalo et al., 2003; Zandomeneghi et al., 2009; Carmona et al., 2012; Ibáñez et al., 2017), we will consider the dataset associated with this volcano as the reference or "master" dataset, thus granting it a high level of reliability and robustness. Therefore, to corroborate our hypotheses, we will use the Popocatépetl and Tajogaite databases as benchmarks.

Deception Island (62°59'S, 60°41'W) is a horseshoe-shaped volcanic island that emerged during the Quaternary period. It is located within a marginal basin-spreading center of the Bransfield Strait, where the South Shetland Islands and the Antarctic Peninsula are separating (Smellie, 1988; Martí et al., 2011; Carmona et al., 2012). The Deception Island dataset (hereafter referred to as MASTER-DEC) was created using seismic data collected during the 1994-1995 campaign organized by the Andalusian Institute of Geophysics (IAG) with a short-period array of 8 channels. The array consisted of a three-component Mark L4C seismometer with a lower frequency band of 1 Hz and 5 Mark L25 sensors with a vertical component frequency of 4.5 Hz, electronically extended to 1 Hz. After analyzing the 8 channels, the one with the highest Signal-to-Noise Ratio (SNR) was selected (Ibáñez et al., 2000). The data were sampled at a frequency of 100 Hz. Since this sampling frequency allows for the analysis of frequencies up to 50 Hz and our parameterization workflow primarily operates within the 1-20 Hz range, the data were filtered within this range. This filtering minimizes the influence of the sensorization used for signal recording and ensuring the comparability of the data recorded by different sensors over various time periods or at different volcanoes. By integrating our understanding of the structural, source, and dynamic models of Deception Island volcano with advancements in signal processing and Machine Learning (ML), MASTER-DEC has played a crucial role in the development of seismo-volcanic signal segmentation and classification. It has also served as the foundation for studies involving hidden Markov models, artificial neural networks, parameter reduction algorithms, and more (e.g., Bueno et al., 2021; López-Pérez et al., 2020; Titos et al., 2018, 2019, 2023; Cortés et al., 2021). Therefore, we can confidently assert that this database is both highly reliable and ideally suited for our intended purpose: serving as a reference seed (pseudo-label) for constructing other seismic catalogs or improving existing ones, particularly those designed for early warning systems for volcanic eruptions. While it is true that

not all types of signals are represented in MASTER-DEC—especially those associated with ongoing eruptive processes—its primary objective aligns with our ML application, which focuses on understanding pre-eruptive processes.

For the current study, we extracted a subset of reliable data, consisting of 2,193 seismic events. These data are categorized into five classes, which align with the volcano-seismic scientific labels and the accompanying source models proposed by Ibáñez et al. (2000) (Table 1 summarizes the source models and classifications). Table 2 presents a detailed summary of the seismic events and their distribution. Figure 1 depicts an example of each type of event corresponding to the prototypes in the database. Figure 2 illustrates the UMAP (Uniform Manifold Approximation and Projection) projection, showing the distribu-

tion of the five MASTER-DEC event types within the feature representation space. This visualization highlights how different seismic events occupy unique but sometimes overlapping regions, revealing potential challenges in distinguishing between event categories. The projection provides an intuitive view of the clustering tendencies and the proximity of events with shared characteristics, underscoring the inherent variability and possible misclassification risk in automatic seismic event recognition systems even in thoroughly analyzed and refined datasets.

| Class | nEvents | min(sec) | mean(sec) | max(sec) | total(sec) | std(sec) |
|-------|---------|----------|-----------|----------|------------|----------|
| BGN | 1222 | 0.3 | 15.4 | 128.2 | 18835.2 | 11.8 |
| TRE | 77 | 10.4 | 93.3 | 150.0 | 7184.2 | 43.63 |
| HYB | 54 | 7.8 | 29.4 | 136.8 | 1587.1 | 18.9 |
| VTE | 75 | 5.4 | 19.1 | 89.9 | 1434.5 | 12.88 |
| LPE | 765 | 2.4 | 9.8 | 30.7 | 7469.8 | 3.81 |

**Table 2.** MASTER-DEC summary. The table reflects statistics on the duration of the signals and the number of events for each class. Seismic categories: Background Seismic Noise (BGN), Volcanic Tremor (TRE), Long Period Events (LPE), Volcano-Tectonic Earthquakes (VTE), and Hybrid Events (HYB). Duration) is in seconds (sec).

Popocatépetl Volcano (19°1'N, 98°37'W) is placed within a different geodynamic framework and exhibits a different eruptive style compared to Deception Island; a subduction region in confront to a rift area. Popocatepetl is a large dacitic–andesitic stratovolcano covering > 500 km2 of the eastern Trans-Mexican volcanic belt (Alaniz-Álvarez et al., 2007; Siebe et al., 2017). It is surrounded by a densely populated area with around 25 million inhabitants (Arango-Galván et al., 2020). The volcano is highly active, with the current active period beginning in December 1994 (Arango-Galván et al., 2020). The dataset used in this

study (hereinafter called POPO2002) was collected during a seismic experiment conducted between November and December 2002, using short-period seismic stations. There is no detailed information regarding the type or specifications of the sensors used to record the seismic signals. Data labelling was manually performed by a group of geophysicists with extensive knowledge and experience of the volcano's dynamics. It consists of 4,883 events, divided into similar classes as the MASTER-DEC catalog (again aligning with the volcano-seismic scientific labels and accompanying source models proposed by Ibañez et al.

2000). Additionally, the catalog includes noisy events (labelled as GAR)-2739 events, and due to Popocatepetl's activity, there is a category for explosions (EXP). Along with the event catalog, we have continuous seismograms from this period that will be used for segmentation and identification processes. Table 3 summarizes the POPO2002 catalog. With the aim of minimizing

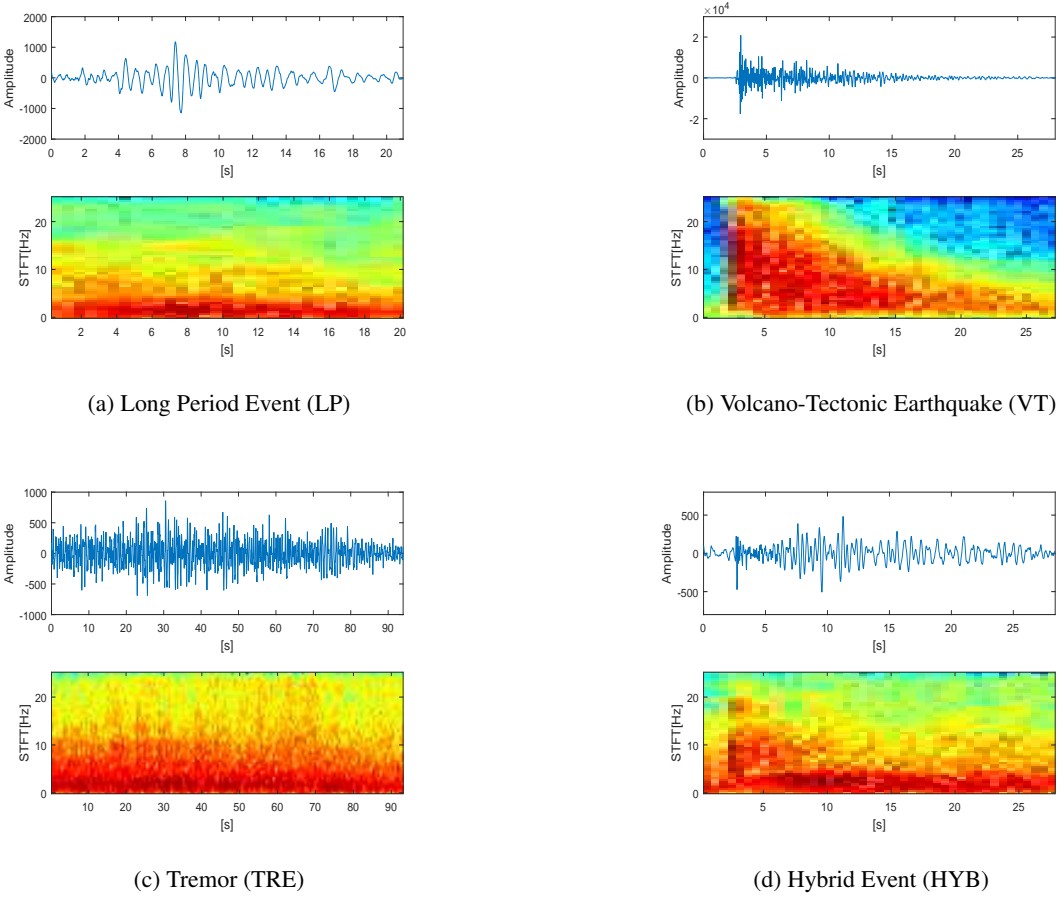

**Figure 1.** Amplitude and spectrograms of the main four prototypes of volcano-seismic events recorded at *Deception Island* volcano, during three seismic surveys: 1994-1995, 1995-1996, and 2001-2002.

the influence of the sensors used for signal recording and ensure data comparability, the signals were first filtered to match the frequency range of MASTER-DEC, followed by a subsampling process to adjust the sampling frequency accordingly.

Tajogaite volcano (28º40'N, 17º52'E) is located on the island of La Palma in the Canary Islands, Spain. The eruptive activity started in September 19, 2021, following a period of seismic activity, marked by several VT swarms and then carried by continuous volcanic tremor, becoming the first eruption on La Palma since 1971. The eruption started with the opening of a fracture in the southwest part of the island, and the emission of material persisted for nearly three months, generated extensive lava flows and pyroclastic deposits (D'Auria et al. (2022)). This event significantly affected the surrounding environment, infrastructure, and regional air traffic. The volcanic process yielded comprehensive seismic and geochemical data, providing valuable insights into volcanic behavior in the Canary Islands and serving as a key reference for improvements in volcanic monitoring and hazard assessment. The seismic catalog for this volcano (from this point forward referred to as LAPALMA2021) differs from

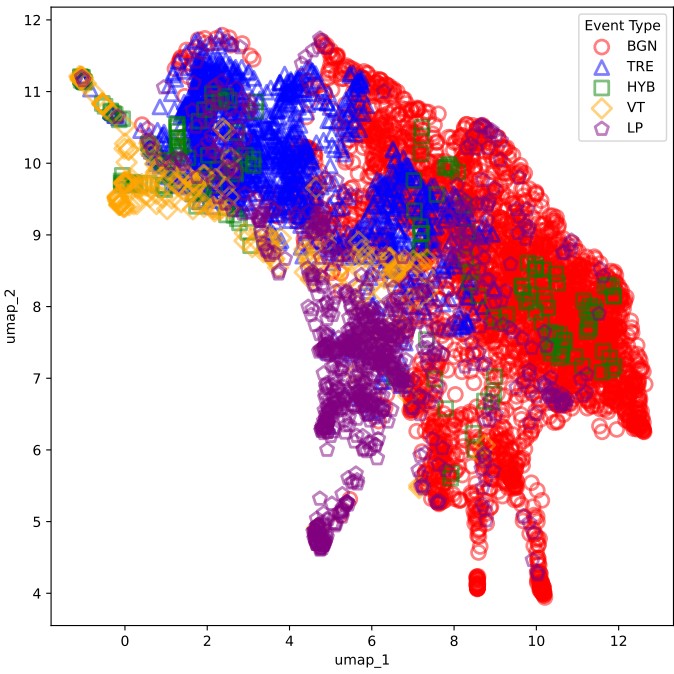

**Figure 2.** UMAP (Uniform Manifold Approximation and Projection) projection obtained for the input vector forming the original data of MASTER-DEC dataset. Different seismic categories may have elements located in overlapping areas of the representation space, where they share similar projected features.

previous seismic catalogs since it only includes annotations of the occurrence of VT-type events. That is, the catalog consists
solely of a series of entries describing the date of the event's occurrence, along with its magnitude and depth. There is no detailed information regarding the type or specifications of the sensors used to record the seismic signals. Given the nature of this catalog and database, we believe that the inclusion of this use case could be of interest for evaluating the capability of the proposed approach to improve a catalog from scratch. Once again, to further minimize the impact of sensor differences and ensure data comparability, the signals were first filtered to match MASTER-DEC's frequency range, then adjusted to the same
sampling frequency.

| Class | nEvents | min(sec) | max(sec) | total(sec) | mean(sec) | std(sec) |
|-------|---------|----------|----------|------------|-----------|----------|
| BGN | 340 | 0.63 | 5048.09 | 311359.63 | 915.76 | 995.18 |
| TRE | 273 | 10.14 | 357.17 | 8798.0 | 97506.93 | 880.23 |
| HYB | 1 | 32.63 | 32.63 | 32.63 | 32.63 | 0.0 |
| VTE | 371 | 6.33 | 1202.7 | 25363.44 | 66.82 | 94.40 |
| LPE | 1155 | 8.95 | 1227.99 | 72866.73 | 63.09 | 43.88 |
| EXP | 4 | 76.82 | 240.59 | 551.86 | 137.97 | 61.52 |
| GAR | 2739 | 0.78 | 14228.95 | 2747967.0 | 1003.27 | 1705.2 |

**Table 3.** POPO2002 summary. The table reflects statistics on the duration of the signals and the number of events for each class. The table reflects statistics on the duration of the signals and the number of events for each class. Seismic categories: Explosions (EXP), Garbaje (GAR), Hybrids (HYB), Long Periods (LP), Volcano-Tectonic Earthquakes (VT), Background Seismic Noise (BGN), Volcanic Tremor (TRE). Duration is in seconds (sec).

## 3 Methodology and experimental framework

This section details the methodolgy and experiments conducted to test our hypothesis that, beyond the changing dynamics of volcanoes between eruptive periods, and intrinsic factors like attenuation effects and source characteristics that alter the shape and spectrum of seismic signals, the effectiveness of automatic seismic monitoring systems is further compromised by the *incompleteness* of the seismic catalogs on which they rely. To accomplish this task, the proposed algorithm will first be described, and then, once its functioning is understood, the three experiments conducted will be detailed. The results of each of these experiments will be detailed in the results section.

### 3.1 Methodology

Building on the architectural strengths and integrating the advanced temporal modeling capabilities of machine learning techniques, this work proposes using a weakly supervised transfer learning algorithm to create new seismic catalogs from which the systems can be retrained with minimal initial human supervision. In other words, instead of retraining the pre-trained systems with a given catalog, this approach proposes using the pre-trained systems as a foundational seed (pseudo-labeler) to weakly label the new database and construct new catalogs. These new catalogs will then serve as the training knowledge for retraining the systems to the new volcanic environment.

Weakly supervised learning is a branch of machine learning covering the construction of predictive models with minimal or indirect supervision (Zhou, 2018). Such techniques focus on learning with incomplete, inexact, and/or inaccurate information derived from noisy, limited, or imprecise supervision processes. The objective is to automatically provide supervision for labeling large amounts of data using labeling functions derived from domain knowledge. This approach replaces the costly and impractical hand-labeled process with inexpensive weak labels, understanding that although imperfect, they can be used to

create a strong predictive model. In this framework, the source domain (denoted as $D_s$) is the MASTER-DEC dataset (based on refined physical models and a strong revision process). The target domain (denoted as $D_t$) is a new given dataset (whose available seismic catalog will not be considered). The goal is to address a domain adaptation task (Kouw and Loog, 2019; Farahani et al., 2021) to reduce the cost of developing a reliable seismic catalog and database for a new given dataset with minimal initial human supervision. That is, automatically provide supervision for labelling large amounts of data from $D_t$ using labelling functions derived from domain knowledge $D_s$.

In a domain adaptation framework, typically $D_s$ and $D_t$ have the same feature space but different distributions. However, in this study, for the pseudo-labeling task we assumed that:

- The marginal distributions of $D_s$ and $D_t$ are the same: $P_s(X_s) = P_t(X_t)$, where $X_s$ and $X_t$ are the input feature vectors associated with different seismic windows or frames in both domains. As such, the pseudo-labeled samples do not need to contain any domain information, and the occurrence of different seismic events is equally likely in both domains.

- The conditional distributions of $D_s$ and $D_t$ are the same: $Q_s(Y_s|X_s) = Q_t(Y_t|X_t)$. As such, the pseudo-labeled samples are valid in both domains.

Such assumptions have important implications since in the target domain, while the marginal distributions of $D_s$ and $D_t$ are the same $[P_s(X_s) = P_t(X_t)]$, the conditional distributions could be different $[Q_s(Y_s|X_s) \neq Q_t(Y_t|X_t)]$. This shows how similar feature vectors taken as the input could output different probabilistic event detection matrices. That is, the description or characterization of seismic categories could change between domains, or $D_t$ could contain seismic categories unforeseen in $D_s$.

Therefore, leveraging the probabilistic detection matrices output by the system trained in $D_s$, we can apply a weakly supervised learning technique as a pseudo-labeller in $D_t$ to construct a new catalog from which to train a new system in a supervised way. Those subset of the unlabelled dataset with high per-class probability, and then high confidence, are added to the new catalog. Although imperfect, this method guarantees that, at least, events showing characteristics similar to those annotated in the master catalog will be included in the new training dataset. As a result, after the re-training phase, the target catalog could be enlarged and updated. It is important to note that this experiment does not aim to correct the catalog created by our colleagues with utmost dedication and effort; it simply seeks to highlight that a pseudo-labeler can be a valuable tool in constructing and reviewing it with success and low time-consuming effort.

Taking these factors into account, our proposed approach is outlined as follows and depicted in Figure 3:

1. **Recognition:** According to Figure 3 a, the recognition block analyzes a subset of data from the new dataset using a pre-trained system (RNN-LSTM, Dilated-RNN, TCN) and gets a probabilistic event detection matrix with per-class membership outputs. The data stream illustrates continuous or streaming analysis (allowing near real-time processing). To carry out the recognition step using the network seed (trained with the MASTER-DEC dataset), streaming or continuous signals are filtered between 1 and 20 Hz and split into frames or windows; the same algorithm of feature extraction used the MASTER-DEC is applied. For each window, a feature engineering pipeline based on a logarithmic scale filter bank is applied. This pipeline reduces the dimensionality of the input vector associated with each analysis window

(compared to raw signals), which facilitates the training and convergence of the systems, as it increases the separability of the data based on well-studied features in the literature (review Titos et al. 2024 for a detailed understanding of the parameterization pipeline).

2. **Event Detection and Confidence Analysis (Concept drift detection):** Ignoring the information contained within the available seismic catalog, the concept drift detection block analyzes the confidence of each detected event using the previously obtained probabilistic event detection matrix with per-class membership output. This step allows us to quantify the severity of drift between datasets (usually knows as "concept drift") (Lu et al. 2018). High or extremely high per-class recognition probabilities for each event type indicate that the systems are well-fitted to the master database. Low per-class probabilities indicate a change in the description of the analyzed information. Accurate and robust dissimilarity measurement and statistical hypothesis evaluation are not strictly necessary given the well-known dissimilarity between volcanic environments.

3. **Concept Drift Adaptation:** An adaptive threshold mechanism where a probability threshold is defined to select the events that will be included in the new database is employed. Events with an average per-class probability exceeding this threshold are selected and incorporated as training instances in the training set.

4. **Re-training process:** Finally, the ML systems trained with the MASTER-DEC used in step 1 are re-trained using the selected instances and labels obtained in step 3.

5. **Iterative Refinement:** Repeat steps 2 to 4 iteratively until the desired result is achieved.

## 3.2 Experimental framework

While the literature offers a variety of accurate machine learning architectures used to uncover descriptive patterns in seismic signals (Malfante et al., 2018; Lara et al., 2021; Hibert et al., 2017; Titos et al., 2018; Bueno et al., 2021; Titos et al., 2019; Canario et al., 2020; Bicego et al., 2022; Alasonati et al., 2006; Benítez et al., 2006; Köhler et al., 2010; Bhatti et al., 2016; Titos et al., 2018; Bueno et al., 2021; Titos et al., 2022), some of these methods may not be as effective for the specific challenges posed by continuous or streaming data (such as POPO2002 and LAPALMA2021). Given the inherent variability and complexity of these data (consisting of seismic signal sequences containing multiple events, where the goal is to detect and classify each individual event), specialized approaches capable of adapting to these conditions are required. More specifically, we will base our experimental framework on the pre-trained systems previously published in Titos et al. (2018, 2022 and 2024). These systems correspond to the Recurrent and Dilated Recurrent Neural Networks (Hochreiter, S., Schmidhuber, J., 1997; (Schmidhuber, J., 2015); (Chang et al. 2017)), both with LSTM cells, along with Temporal Convolutional Networks (Lea et al., 2017) (henceforth referred to as RNN-LSTM, Dilated-LSTM, and TCN, respectively).

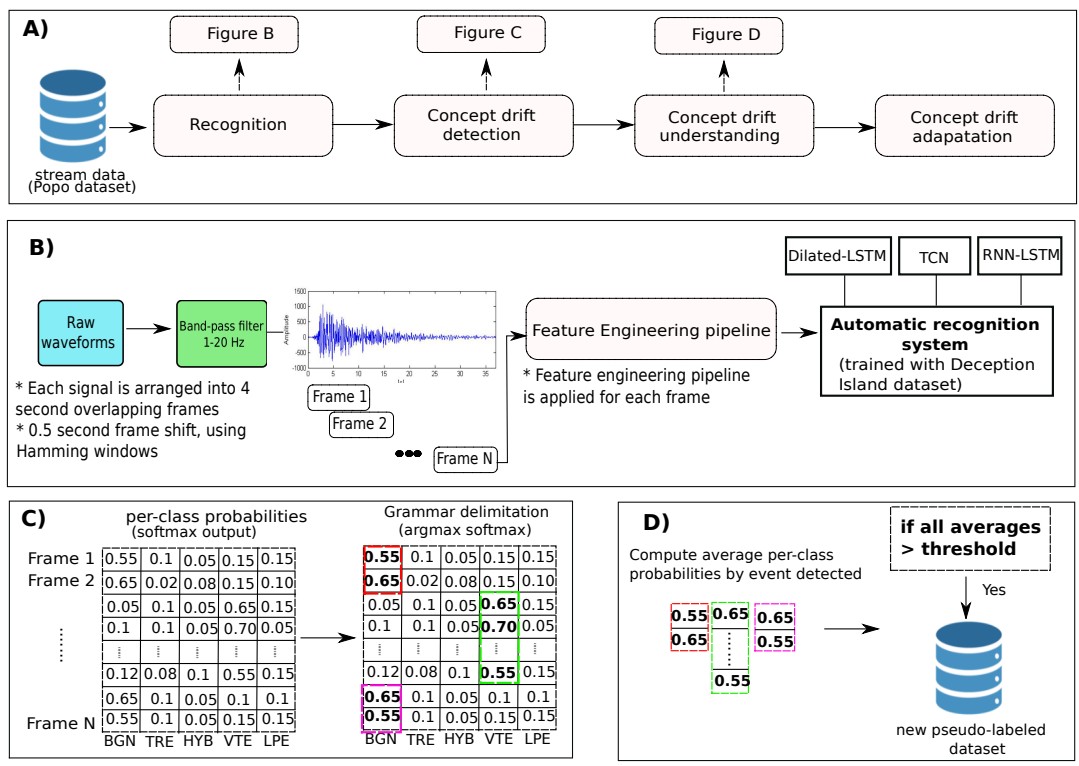

**Figure 3.** a) Overview of the weakly supervised event selection algorithm developed. A subset of the dataset (in our case 40% of the total) is used as a training set by the reference pre-trained systems. The rest of the data is used as a test set. Only high per-class probability recognized events are selected as new training instances. b) Workflow structure and the specific preprocessing steps employed, which relies on frequency analysis within the logarithmic filter bank domain (Titos et al. 2022). This processed information serves as the input for the different volcano-seismic recognition systems. c) For each detected event, the confidence of the detection is analysed using a probabilistic event detection matrix with per-class probabilities output by the systems. d) Drift adaptation mechanism based on an adaptive threshold is then adopted. Those events whose average number of per-class probability is greater than a given threshold are selected and included as new training instances.

### 3.2.1 Developing automatic recognition systems from available catalogs

The standard procedure for developing an automatic volcano-seismic monitoring system from scratch using supervised machine learning techniques involves having a sufficiently representative seismic catalog (selecting and segmenting a large, reliable set of well-labeled seismic events that cover the maximum possible range of events occurring in the studied volcanic area). These events serve as the initial seed for the training procedure. This training can be carried out using different approaches, ranging from training the system from scratch to using transfer learning techniques (Weiss et al. (2016)). Transfer Learning offers significant advantages, especially when the available data for a particular volcano is limited. This technique allows for reusing a model pre-trained on data from another volcanic region (for example, a previously studied volcano) and adapting it

to new data with considerably less computational and labeling effort. By transferring knowledge acquired from one volcano to another, the system's ability to recognize seismic patterns and adapt to different volcanic characteristics could be enhanced, leading to improved accuracy and generalization.

Thus, in the first experiment, to demonstrate that state-of-the-art machine learning model can effectively learn the information contained in a seismic catalog (assuming catalog-induced learning biases), a recognition system based on transfer learning approaches will be developed from scratch utilizing three different architectures. To achieve this, three systems pre-trained on MASTER-DEC will serve as the foundation for adapting recognition systems to the Popocatépetl volcano. Specifically, these systems will be re-trained with the available data and catalog from POPO2002 dataset. Given that the POPO2002 catalog contains 7 seismic categories, a recognition system based on transfer learning can be constructed in different ways. One approach is to consider only the categories present in MASTER-DEC, while another includes all the categories (i.e., incorporating Explosions and Garbage events in the training set, thus expanding the number of seismic categories by two). From a machine learning perspective, these two approaches have no major implications. In the first case, where only 5 seismic categories are used, the systems would undergo retraining with the new catalog. In the second case, when using 7 categories, systems are adjusted to accommodate 2 additional categories, leveraging the pre-existing parameters while updating only the output layer. After that, the models are trained as usual.

### 3.2.2 Developing automatic recognition systems with weakly supervised pseudo-labeling

To test our initial hypothesis and following the workflow outlined in Figure 3, this second experiment highlights the use of weakly supervised approaches to enhance seismic-volcanic catalogs. By leveraging an existing unbiased master catalog, we can incorporate prior knowledge into the new dataset under review. This process involves using each of the three reference systems (RNN-LSTM, Dilated-LSTM, TCN), considered well-trained, to reassess and label the seismic categories in the new dataset, then retraining themselves based on these pseudo-labels. Therefore, Each system will analyze a subset of the total POPO2002 database to create a training set for the retraining process. Once retrained, the systems will generate a new seismic catalog, which will then be compared and analyzed against the original POPO2002 catalog to assess the results.

Since MASTER-DEC is composed of five seismic categories, and the weakly supervised approach relies on pre-trained models, the experiments presented here are based solely on these five categories. This limitation is a consequence of the methodology and must be properly understood in order to ensure a correct interpretation and discussion of the results, as it directly influences the way the data is analyzed and compared with the original catalog. An important consideration in this experiment is that the recognition percentage obtained by the systems before and after retraining, using the original catalog annotations as a reference, can provide valuable insights into the algorithm's behavior. Therefore, both results will be taken into account in this experiment, with the aim of analyzing in detail how the retraining process with the new pseudo-catalog affects the system's performance.

### 3.3 Building a new catalog during an eruptive crisis: The Tajogaite volcano use case, 2021

The third and final experiment aims to analyze the robustness of the proposed methodology by building a seismic catalog from scratch in a highly demanding use case, such as during an eruptive crisis. Since we have not had the opportunity to test it in an actual eruptive scenario, we propose using data from the Tajogaite volcano during the 2021 eruptive episode. We also suggest abstracting this offline test to simulate a real-time episode, as if data were being analyzed in real-time, since the functionality would be exactly the same. As previously mentioned, the selected pre-trained systems are capable of operate in near real-time, with particularly short latency times, analyzing (not re-training) 24 hours of data in few seconds.

Therefore, for this experiment, a pair of 24-hour seismic records from the PPMA and PLPI seismic stations, corresponding to September 12, 2021, just a few days before the eruption began when an increase in activity was detected, have been used. To conduct an analysis and comparison of the results, we have a seismic catalog created by geophysical experts from that volcano during the eruption crisis itself. Given the large number of recorded events, the significance, and the urgency of the moment, we believe that this catalog meets the human requirements of the time. Again, just as we argued in the case of the POPO2002 catalog, this experiment does not aim to correct the catalog created by our colleagues with utmost dedication and effort; it simply seeks to highlight that a pseudo-labeler can be a valuable tool in constructing and reviewing it.

## 4 Results

This section presents the results supporting the experiments outlined in the previous section. For each experiment, tables describing the system performances in terms of accuracy, along with detailed confusion matrices are presented. For Experiments 1 and 2, accuracy (%) metric evaluates the capability of the developed systems to accurately recognize (detect and classify) the events annotated in the POPO2002 seismic catalog. The normalised confusion matrices provide a breakdown of true positives, false positives, false negatives, and true negatives, allowing a thorough analysis of each system's robustness in recognizing each event type. All results were obtained using a Leave-One-Out cross-validation process with 4 random partitions. Each time, we select T% of the entire database as training set, and the remaining (100-T) % as test set to check the performance of the systems. This analysis helps to identify specific areas where the model may struggle, such as mis-classification between event types with similar features. Finally, in experiment 3, where only partial knowledge of the earthquakes recorded during the crisis is available, results evaluate the model's ability to generate a more comprehensive and reliable catalog. This catalog will serve as a basis for inferring potential volcanic dynamics, with confusion matrices helping to assess how well the model distinguishes between known and newly identified event patterns, which is critical in real-world eruptive crisis scenarios.

The optimal RNN-LSTM configuration consists of a single hidden layer with 210 units and no dilations. For the Dilated-LSTM model, the configuration that yielded the best performance included three hidden layers, each with 50 units and 2–4 dilated recurrent skip connections per layer. The TCN model, achieved optimal performance with 50 filters, a kernel size of 2, and dilation values of 8, 16, and 32. Only one residual block was used, as additional blocks are more suitable for longer sequences, such as waveforms with extensive time samples. Data normalization was performed using standard deviation normalization, where each feature was normalized by subtracting its mean and dividing by its standard deviation, calculated from

the training set. The model was optimized using Stochastic Gradient Descent (SGD) with a fixed learning rate, ranging from
0.004 to 0.01, with the optimal learning rate found to be 0.001. To prevent overfitting, early stopping and L2 regularization
were applied during training.

## 4.1 Developing automatic recognition systems from available catalogs

Table 4 presents the recognition results obtained by the pre-trained systems after being trained on POPO2002 catalog. Since
using a transfer learning approach allows for more efficient use of computational resources, and the fine-tuning phase typically
requires fewer resources than training a system from scratch, two experiments were conducted. These experiments considered
5 and 7 seismic categories, each using 20% and 40% of the total data for the training set (T = 20 and T = 40). This means that
the results were obtained using 80% and 60% of the data in the test partition, respectively. Table 5 summarizes the averaged
normalised confusion matrices belonging to the test using 5 seismic categories and 40% of the total data for the training set.

|  | 5 seismic categories | | 7 seismic categories | |
| --- | --- | --- | --- | --- |
|  | Training percentage | | Training percentage | |
|  | 20% | 40% | 20% | 40% |
| RNN-LSTM | 77.38 | **88.99** | 84.01 | **84.39** |
| Dilated-LSTM | 82.88 | **84.70** | 84.05 | **85.21** |
| TCN | 82.46 | **88.30** | **85.77** | 83.27 |

**Table 4.** Self-consistency results using 5 and 7 seismic categories, with 20% and 40% of the data for training and 80% and 60% for testing, respectively. The results correspond to the average accuracy over the four partitions.

|  | RNN-LSTM | | | | | Dilated-LSTM | | | | | TCN | | | | |
| --- | --- | --- | --- | --- | --- | --- | --- | --- | --- | --- | --- | --- | --- | --- | --- |
|  | BGN | TRE | HYB | VTE | LPE | BGN | TRE | HYB | VTE | LPE | BGN | TRE | HYB | VTE | LPE |
| BGN | **0.97** | 0.02 | 0 | 0 | 0.01 | **0.96** | 0.02 | 0 | 0 | 0.02 | **0.98** | 0.01 | 0 | 0 | 0.01 |
| TRE | 0.06 | **0.78** | 0 | 0.05 | 0.11 | 0.13 | **0.69** | 0 | 0 | 0.18 | 0.11 | **0.68** | 0 | 0.09 | 0.12 |
| VTE | 0.08 | 0.13 | 0 | **0.51** | 0.28 | 0.12 | 0.17 | 0 | **0.31** | 0.4 | 0.14 | 0.09 | 0 | **0.59** | 0.18 |
| LPE | 0.05 | 0.07 | 0 | 0.03 | **0.85** | 0.04 | 0.18 | 0 | 0 | **0.78** | 0.05 | 0.05 | 0 | 0.04 | **0.86** |

**Table 5.** Averaged normalized confusion matrices associated with the Leave One Out cross validation process for the Popo2002 dataset. These results belong to the test using 5 seismic categories.

## 4.2 Developing automatic recognition systems with weakly supervised pseudo-labeling

Table 6 presents the recognition accuracy achieved by the pre-trained systems, which were retrained using the proposed weakly supervised approach with the training partition set to 40% of the total POPO2002 dataset. As previously stated, since MASTER-DEC consists of five seismic categories and the weakly supervised approach builds on pre-trained models, the results presented here include only these 5 seismic categories. The first column of the table represents the results obtained by directly applying recognition with the pre-trained models. This column shows the degree of similarity between the original POPO2002 catalog and the pseudo-catalog constructed using the pre-trained systems as pseudo-labelers. The second column reflects recognition results compared to the original POPO2002 catalog after the systems have been retrained using the previously constructed pseudo-catalog. Table 7 summarizes the averaged normalized confusion matrices of the new systems based on the weakly supervised approach, with the POPO2002 catalog as the reference. The results are over the whole test set using 40% of the whole set for training and five seismic categories. The y-axis corresponds to the real label or ground-truth and the x-axis corresponds to predicted labels. Finally, Table 8 summarizes the comparison between the events initially annotated in the POPO2002 catalog and the events detected by the new automatic systems following the weakly supervised approach.

|  | Five seismic categories blind test | 'Weakly supervised TL' using five seismic categories TL |
|---|---|---|
| RNN-LSTM | 55.95 | 64.89 |
| Dilated-RNN | 50.13 | 55.72 |
| TCN | 58.27 | 66.16 |

**Table 6.** Classification accuracy (acc. %) on the test set achieved by the pre-trained systems, which were retrained using the proposed weakly supervised approach with the training partition set to 40% of the total POPO2002 dataset and only 5 seismic categories.

|  | RNN-LSTM | | | | | Dilated-LSTM | | | | | TCN | | | | |
|---|---|---|---|---|---|---|---|---|---|---|---|---|---|---|---|
|  | BGN | TRE | HYB | VTE | LPE | BGN | TRE | HYB | VTE | LPE | BGN | TRE | HYB | VTE | LPE |
| BGN | **0.88** | 0.09 | 0 | 0 | 0.03 | **0.67** | 0.32 | 0 | 0 | 0.01 | **0.8** | 0.19 | 0 | 0 | 0.01 |
| TRE | 0.29 | **0.36** | 0.03 | 0.02 | 0.03 | 0.29 | **0.5** | 0 | 0 | 0.21 | 0.19 | **0.7** | 0 | 0 | 0.11 |
| VTE | 0.27 | 0.41 | 0.08 | **0.03** | 0.21 | 0.46 | 0.28 | 0 | **0.03** | 0.23 | 0.36 | 0.46 | 0.03 | **0.06** | 0.09 |
| LPE | 0.36 | 0.19 | 0.06 | 0.06 | **0.33** | 0.47 | 0.18 | 0 | 0.01 | **0.34** | 0.41 | 0.33 | 0.01 | 0.01 | **0.24** |

**Table 7.** Normalized confusion matrices for the new retrained system using a weakly supervised approach, with the POPO2002 catalog as reference. The results are over the whole test set using 40% of the whole set for training and five seismic categories. The y-axis corresponds to the real label or ground-truth and the x-axis corresponds to predicted labels.

## 4.3 Building a new catalog during an eruptive crisis: The Tajogaite volcano use case, 2021

Table 9 shows the recognition results obtained in this experiment using 24-hour seismic traces from the PLPI and PPMA stations on 9/12/2021 at Tajogaite volcano. The number of events manually annotated by experts during the volcanic crisis for

|  | Popo2002 catalog | RNN-LSTM | Dilated-LSTM | TCN |
|---|---|---|---|---|
| BGN | 340 | >20,000 | >20,000 | 17,206 |
| TRE | 273 | 3,291 | 2,538 | 3,204 |
| VTE | 371 | 1,741 | 1,032 | 94 |
| LPE | 1,155 | 2,230 | 2,250 | 2,159 |

**Table 8.** Comparison between the events initially annotated in the catalog and the events detected by the new automatic systems following the implementation of a weakly supervised approach.

the analyzed day, serving as a guide for the subsequent analysis is 247 earthquakes, both tectonic and volcanic in origin. As mentioned earlier, it is important to highlight that these results correspond to an experiment where only a tentative earthquake
catalog (constructed during the eruptive crisis under the urgency and challenges that such situations entail) is available. For this reason, to conduct a rigorous comparative analysis, we have included the recognition results from a widely-used tool like PhaseNet (Zhu and Beroza (2019)). PhaseNet is a neural network-based system designed for automatic phase picking of seismic events. It detects and labels seismic phases and estimates the probability of each phase type (P and S) across the trace. After analyzing the two seismic stations, PLPI and PPMA, for September 12, 2021, 1173 P-phases and 1518 S-phases were
obtained for PLPI, and 390 P-phases and 522 S-phases for PPMA.

|  | RNN-LSTM | | Dilated-LSTM | | TCN | |
|---|---|---|---|---|---|---|
|  | PLPI | PPMA | PLPI | PPMA | PLPI | PPMA |
| BGN | 4344 | 4641 | 1800 | 3005 | 6409 | 8642 |
| TRE | 109 | 64 | 229 | 241 | 152 | 139 |
| HYB | 12 | 14 | 5 | 8 | - | - |
| VTE | 187 | 131 | 194 | 161 | 333 | 403 |
| LPE | 1008 | 1032 | 564 | 711 | 516 | 761 |

**Table 9.** Recognition results obtained by the pre-trained reference models on the seismic traces recorded on 12/9/2021 at the PLPI and PPMA stations. Results are without considering re-training process.

## 5 Discussion

### 5.1 Developing automatic recognition systems from available catalogs

The classical way to assess the robustness of an automatic recognition system is by evaluating its recognition accuracy across
all events included in the catalog. Typically, a system with an average performance below 75% is considered unreliable. However, this lack of reliability is often not due to the system's ability to learn to distinguish between different events, but rather results from how the catalog is constructed. Specifically, if the seismic categories are not homogeneous and events of different natures are assigned to the same type, the system's performance will drop. If events classified as part of the same category are

not consistent, the system will struggle to make accurate predictions, as the inherent variability within each type undermines the learning process. In such cases, the recognition accuracy typically falls below 70%. Therefore, Tables 4 and 5 not only provide information about the reliability of the developed systems but also about the consistency of the catalog itself.

According to such results, the three proposed systems are shown to achieve a high degree of recognition in both experiments (including 5 and 7 seismic categories), allowing us to conclude that the systems effectively learn to recognize the events annotated in the catalog. It is worth noting, however, that in the second experiment, when the number of seismic categories increases from 5 to 7, the recognition rate of the 3 systems is slightly affected. This result is clearly influenced by the imbalance in the dataset. The seismic category Explosion (EXP), with only 4 events, has no impact on the outcome. In contrast, the inclusion of the Garbage (GAR), with 2,739 events of varying durations, significantly changes system performance. Firstly, because it is the predominant category in terms of both number and duration, performing an analysis by windows results in a considerable increase in labels of this type, biasing system learning. Secondly, the spectral characteristics describing GAR events are very similar to those of BGN events. The former represents a set of events without a clear definition, while the latter represents seismic noise. Therefore, including both in the training process leads the systems to confuse the two, with GAR emerging as the more dominant qualitative category due to its imbalance.

Regarding the confusion matrices across the 3 systems, the analysis suggests that, the POPO2002 catalog is consistent, within each seismic category, there is coherence among the elements classified within the same category. However, propagation and source effects can influence seismic event characterization. For instance, VTE events are not well-identified, with confusion rates exceeding 60% in some cases, meaning only 40% of VT events are accurately classified. The highest confusion levels are observed between the VTE and LPE categories, possibly due to shared characteristics, as LPE events may resemble highly attenuated VTs, causing potential biases in event categorization. This overlap suggests that some seismic categories have elements positioned in overlapping areas of the representation space (mathematical space where data points are mapped according to learned features), where they share similar projected features, and events, despite being assigned to a specific cluster, could transition between categories (similar to MASTER-DEC and described in Figure 3). Thus, although system performances range between 85% and 90%, this does not always reflect a complete or unbiased seismic catalog. Rather than solely learning to characterize volcano dynamics based on an underlying physical model, the systems may be learning the information contained within the catalog itself. Consequently, catalog-induced learning could limit a system's ability to generalize, potentially obscuring information relevant to advancing our understanding of volcanic behavior.

## 5.2 Developing automatic recognition systems with weakly supervised pseudo-labeling

Once the construction of catalogs through transfer learning has been discussed, we are now ready to discuss the use of weakly supervised pseudo-labeling approaches. Results demonstrate that, when applied effectively, these methods can significantly improve the detection and identification of diverse earthquake-volcanic signals. According to Table 6, using pre-trained systems as pseudo-labelers results in a substantial decrease in overall performance compared to building automatic monitoring systems from available catalogs (Table 4). However, a closer inspection of the Table 8 shows other aspects of the performance being very encouraging.

First, the new systems recognized events that were originally not annotated in the preliminary catalog during data-labeling. The vast majority of such recognized events, were discovered within long segments labeled as GAR or TRE. An example of this behavior can be seen in Fig. 4, which shows LP events (red boxes) that were not initially annotated during labeling within a trace labeled as TRE, along with the correction of an event originally labeled as LP, now relabeled by the system as VT. This scenario occurs many times throughout the dataset, and these additional labels reduce overall recognition accuracy relative to the original labeling, although they do not necessarily represent errors.

Second, among the seismological community, there is a marked interest in associating different types of seismo-volcanic

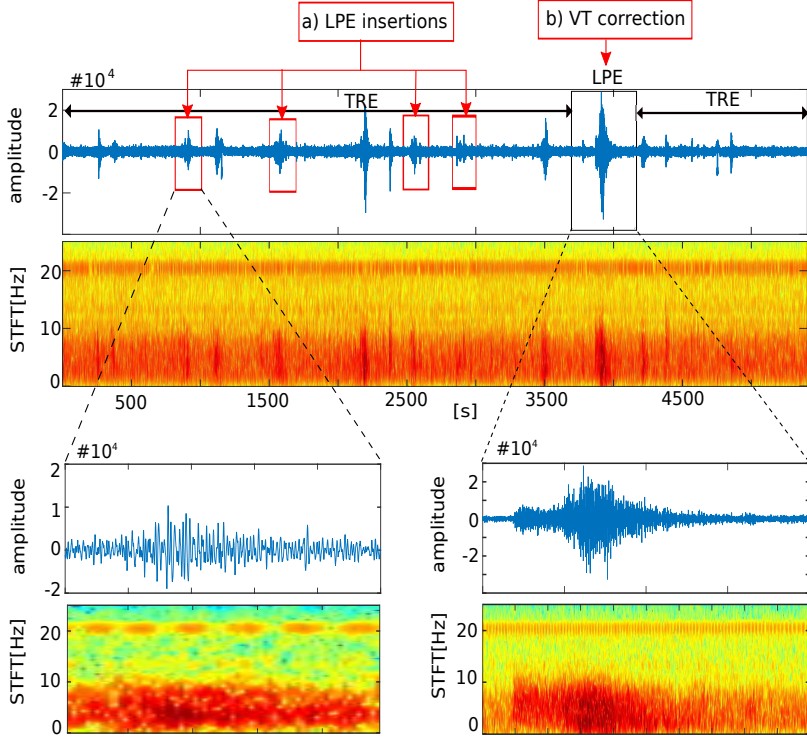

**Figure 4.** Insertion-based errors when retraining systems using a weakly supervised approach. Detection of LP events (red boxes) that were initially overlooked during the labeling process within a trace labeled as TRE–LPE–TRE. Correction of an event originally labeled as LP, which the system now re-labels as VT. This scenario occurs frequently throughout the dataset, and these additions reduce per-frame recognition accuracy compared to the original labeling; however, they do not always indicate errors.

signals with models of seismic sources in order to better understand the physics of the underlying processes. At present, there are two main complementary lines of research within volcano seismology: a) the detection and identification of different types of volcanic events and b) the investigation of physical source models that explain the origin of these signals. As scientific knowledge has advanced, a paradoxical situation has developed: there is a lack of uniformity in the naming of observed seismic signals. Therefore, the subjectivity of human operators during the labeling process can lead to discrepancies in catalog construction. As a result, catalogs and automatic recognition outcomes often vary across different volcanoes and researchers,

which ultimately reduces the system's ability to be universally applied and impacts its performance. A clear example of this discrepancy can be seen in Table 7. According to such table, on average, only 5% of the analysis windows labeled as VTE in the original catalog were recognized by the retrained systems. On initial inspection, these results might suggest poor systems recognition for this seismic category, but interestingly, it is one of the most distinctive events due to its high-frequency content and exponential energy decay. So, what accounts for the low recognition rate? A detailed analysis shows that it is mainly due to labeling discrepancies between the MASTER-DEC event prototypes and POPO2002 catalog annotations. On the one hand, the start and end points of some events are often marked in positions that differ significantly from those annotated by the automatic systems. Instead of recognizing entire seismic traces such as volcano-tectonic earthquakes (VTE) as annotated in the original catalog, the systems detect background noise (BGN) segments before and after the earthquakes. While segments with high spectral content were detected and classified as VTE, those with low spectral content were classified as BGN or TRE. These additional detections reduce per-frame recognition accuracy. This can be clearly seen in Fig. 5 during earthquakes recognition. On the other hand, the VTE prototype events used in MASTER-DEC have very specific characteristics. However, some of

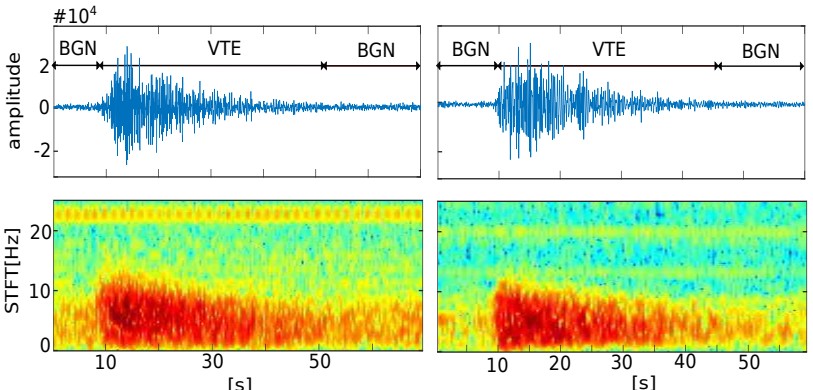

**Figure 5.** Insertion-based errors when retraining systems using a weakly supervised approach. Event delimitation: examples of the labeling process for the systems. Instead of recognizing entire seismic traces such as volcano-tectonic earthquakes (VTE) as annotated in the original catalog, the systems detect background noise (BGN) segments before and after the earthquakes. These additional detections reduce per-frame recognition accuracy; however, after a posterior revision, they should not be considered errors. The current colormap in the spectrogram represents the energy levels. The blue color corresponds to the minimum energy, while the red color corresponds to the maximum energy.

the VTE events labeled in POPO2002 do not reliably share these characteristics. This may be due to the fact that catalogs are often constructed using data from multiple seismic stations, with strong attenuation and source effects, while imposing rules or conditions for identifying signals. Therefore, the original labeling of an event does not always align with the waveform and spectral content of the analyzed signal, as it may vary depending on the station being analyzed. As a result, if the signal being analyzed does not align with the characteristics of the prototype event used to construct the system, such signal will be labeled or associated with the event prototype that most probabilistically resembles it. This behavior reduces the recognition rate for this seismic category. Figure 6 illustrates this behavior, showing two examples of events annotated as VTE in the POPO2002

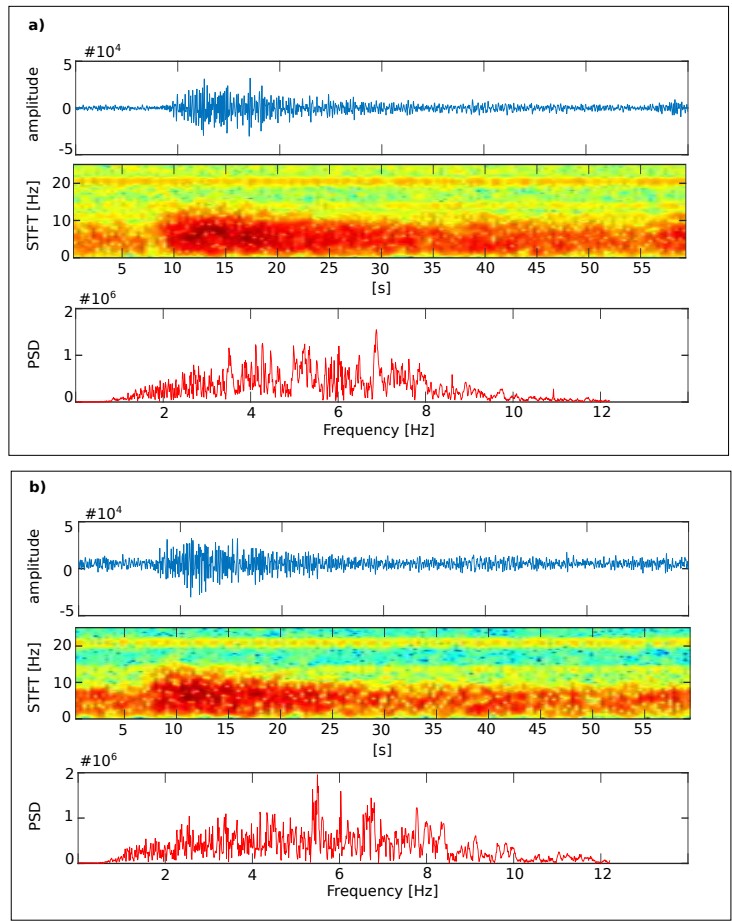

**Figure 6.** Two examples of event annotated as VTE in the POPO2002 catalogue being recognized as TRE for the systems. The current colormap in the spectrogram represents the energy levels. The blue color corresponds to the minimum energy, while the red color corresponds to the maximum energy. The PSD reflects the distribution of a signal's energy among the frequencies.

catalog that are recognized as TRE by the systems. The Power Spectral Density (PSD) of both events shows a clear content in low and intermediate frequencies (1-12 Hz), perfectly aligning with the source model proposed by Ibañez et al. (2000) in Table 1, which is also followed by the MASTER-DEC. Similar to the previous analysis, this behavior is repeated throughout the database, not only with TRE but also with LPE events, which explains the high degree of confusion addressed. A potential solution to this situation would be to apply the algorithm to different stations.

Third, intra-category variability can also affect the overall recognition of the systems. The new dataset contains high variability in some categories (categories composed of distinct events with shared characteristics are grouped into a single category, such as various LPs, TRE events, or regional and volcano-tectonic earthquakes all labeled collectively as earthquakes). Again, the nature of the seismic data played an essential role. Within the feature space, the representation of events belonging to a given subcategory in the new domain (POPO2002) was closely related to the representation of events belonging to a different

category in the source domain (MASTER-DEC). For example, similar to what occurs with some events in Fig. 3, the representations of some LPEs in POPO2002 are very close to the representation of TRE in MASTER-DEC (Fig. 7)a). As such, the algorithm assigns the TRE label during the training phase. This decreased the overall systems performance since many frames (33%, 19%, and 18% for TCN, RNN–LSTM, and Dilated–LSTM, respectively) were detected as TRE. The same issue arose for some attenuated earthquakes, which were labelled as LPE in the original seismic catalog but classified as VTE or TRE since, even when attenuated, they align with the feature space representation of an earthquake event in MASTER-DEC (Fig. 7b). Finally, low-energy TRE events were clearly mis-classified as BGN because the peak-to-peak amplitude degradation of the signals was related to attenuation effects. This complex scenario was widely discussed by Titos et al. (2018); therefore, to correctly deal with these errors, further information from several seismic stations is needed. The results suggest that overall recognition can be strongly biased by the intrinsic limitations addressed when developing the seismic catalog and from which the comparative metrics were obtained. Therefore, if labelling criteria between datasets differ, per-frame recognition results will vary widely. Hitherto, the development of new monitoring systems has focused primarily on improving existing recognition rates. However, our findings confirm that by leveraging an existing unbiased master catalog, we can incorporate prior knowledge into the new dataset under review. Using automatic pseudo-labelers have the remarkable capability of simultaneously identifying unannotated seismic traces in the catalog and help to correct the labels of mis-annotated seismic traces. Although the general performance of the system seems to decrease relative to the original catalog, previously hidden information that can improve knowledge of the volcanic dynamic background can be obtained.

### 5.3 Building a new catalog during an eruptive crisis: The Tajogaite volcano use case, 2021

To conduct a detailed analysis of the results obtained in this experiment, it is essential to know the reference data. As mentioned earlier, this experiment considered the seismic traces from two stations, PLPI and PPMA, for September 12, 2021, a few days before the eruption of Tajogaite volcano began. On this day, given the volcanic activity and monitoring conditions only 247 earthquakes, both tectonic and volcanic, were annotated in the catalog.

Considering this information, we now proceed to discuss the results. For the sake of the comparison, we will start analyzing the outcomes obtained by PhaseNet. PhaseNet detected several hundreds of P and S phases, with the number of S phases being higher at both stations. It is due to the greater energy associated with these waves. However, as it can be seen in Figure 8a, when fixing a phase score threshold highlighting the reliability of the detections, the number of detections decreases rapidly with high values. For example, for values close to 80%, only approximately 722 P-phases–503 S-phases at PLPI and 282 P-phases–216 S-phases at PPMA are detected. This significantly reduces the number of potential events that could be included in the catalog. Fig. 8b shows the match between detections and the cataloged events. Of these 247 annotated events, Phasenet detects 206 P-phases and 199 S-phases at PLPI; and 157 P-phases and 28 S-phases at PPMA, all without applying any probability threshold. Again, when setting the phase score threshold greater than or equal to 80%, the detections decrease to 163 P-phases and 164 S-phases at PLPI, and 116 P-phases and 21 S-phases at PPMA. This behavior underscores the complexity of constructing seismic catalogs, as even when focusing solely on seismic phase detections, there is no consistent criterion between a human

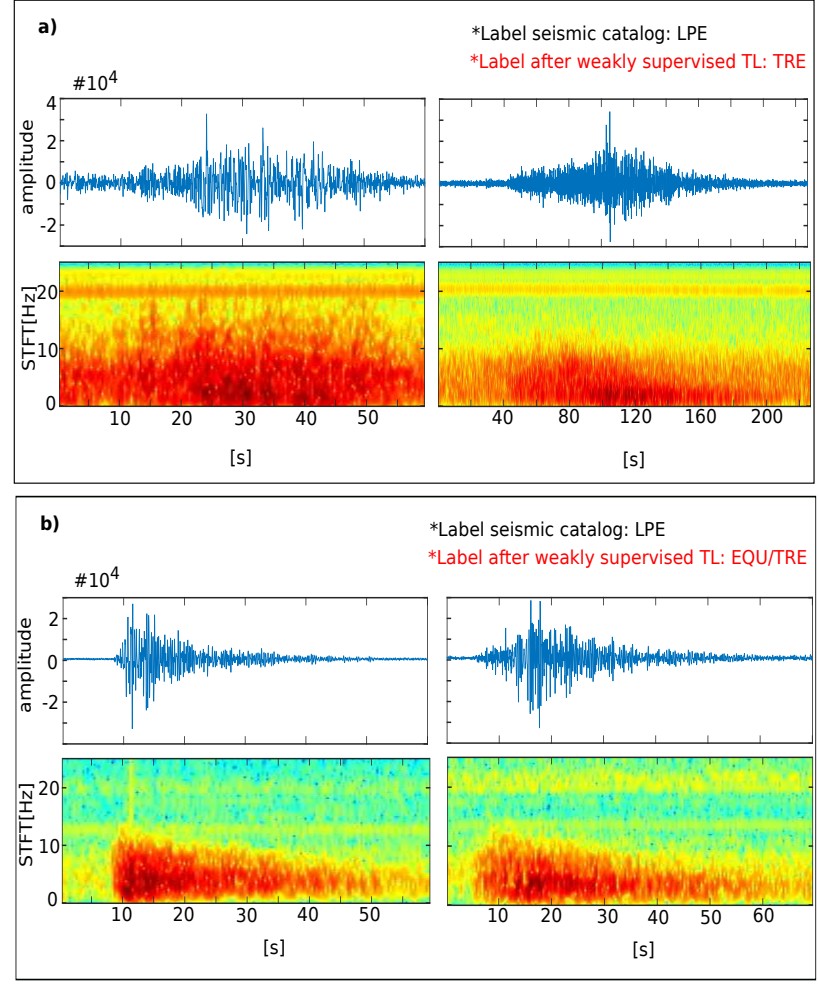

**Figure 7.** Detailed analysis of intra-class variability and attenuation-based errors when applying a weakly supervised approach. (a) Intra-class variability-based errors: some long period event (LPE) subcategories in POPO2002 are very close to the representation of tremor (TRE) in MASTER-DEC. (b) Two attenuated earthquakes labelled as LPE in the seismic catalog, but classified as volcano-tectonic earthquake (VTE) or TRE. The current colormap in the spectrogram represents the energy levels. The blue color corresponds to the minimum energy, while the red color corresponds to the maximum energy.

operator and advanced automatic systems for choosing events. More importantly, even when considering the inclusion of these potential events, extensive human supervision would be required to validate and categorize them.

Looking at the recognition results obtained by the pre-trained reference systems (see Table 9), it can be observed that a big amount of events are being detected. However, similar to Phasenet, some of such events should be discarded because the reliability of the recognitions. Figure 9 depicts such reliability based on the belonging probabilities outputted by the systems. To dive into these results: 1) we will analyze how the number of detections changes as the reliability changes (we focus on

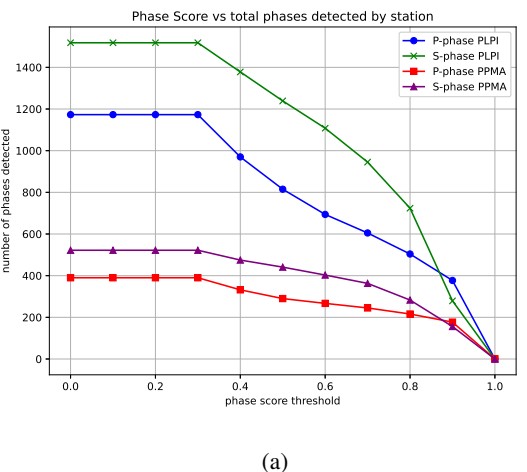
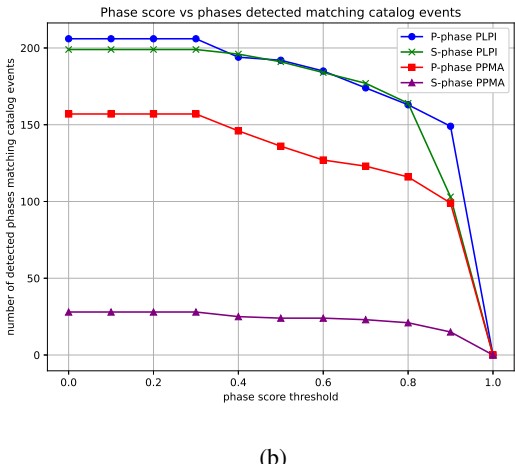

|         (a)         |         (b)         |

**Figure 8.** Evolution of the number of detected phases at the seismic stations as the phase score threshold varies. A) Total number of phases detected at both stations. B) Number of phases matching the 247 events recorded in the catalog on 12/9/2021.

more specific or sensitive systems); 2) we will examine how the systems perform using as reference the 247-events annotated in the catalog; and 3) we will assess the reliability of the remaining detected events in order to evaluate the reliability of the new pseudo-catalogs.

Across all systems and at both stations, the number of detected events decreases significantly as the probability threshold increases, particularly for values above 80%. At higher thresholds, the detections are predominantly limited to events closely correlated to the prototype events on which the systems were trained. Figure 9c shows that for thresholds above 80%, the number of detected earthquakes by both RNN-LSTM and Dilated-LSTM averages between 120 and 150 events at both stations. For TCN, the number of detected earthquakes is significantly higher, highlighting that its specificity could be set at slightly higher thresholds, around 85-90%. The main reason for the non-detection of certain catalog-annotated events was their differing spectral content compared to the average spectral content of the earthquakes annotated in the catalog. Specifically, by comparing the spectral content of the undetected events with the average spectral content of all the annotated events, a clear attenuation of energy is observed at higher frequencies (>15 Hz). This characteristic is crucial, as the systems were trained with prototype events that had a clear energy component at high frequencies. Figure 11 illustrates a couple of examples of this behavior. The first row corresponds to the seismogram of the event being analyzed (annotated in the catalog but not detected by any of the systems). The second row corresponds to their spectrograms. The third and fourth rows show the average power spectral density (PSD) of all events annotated in the catalog for that day and the PSD of the event under analysis. The fourth row of both Figure 10a and 10b show a clear attenuation of energy at high frequencies and a higher level of energy at lower and intermediate frequencies, respectively. In general, these events reflect belonging probabilities ranging between 50% and 80%. It highlights the importance of adjusting the specificity or sensitivity threshold when creating new pseudo-catalogs.

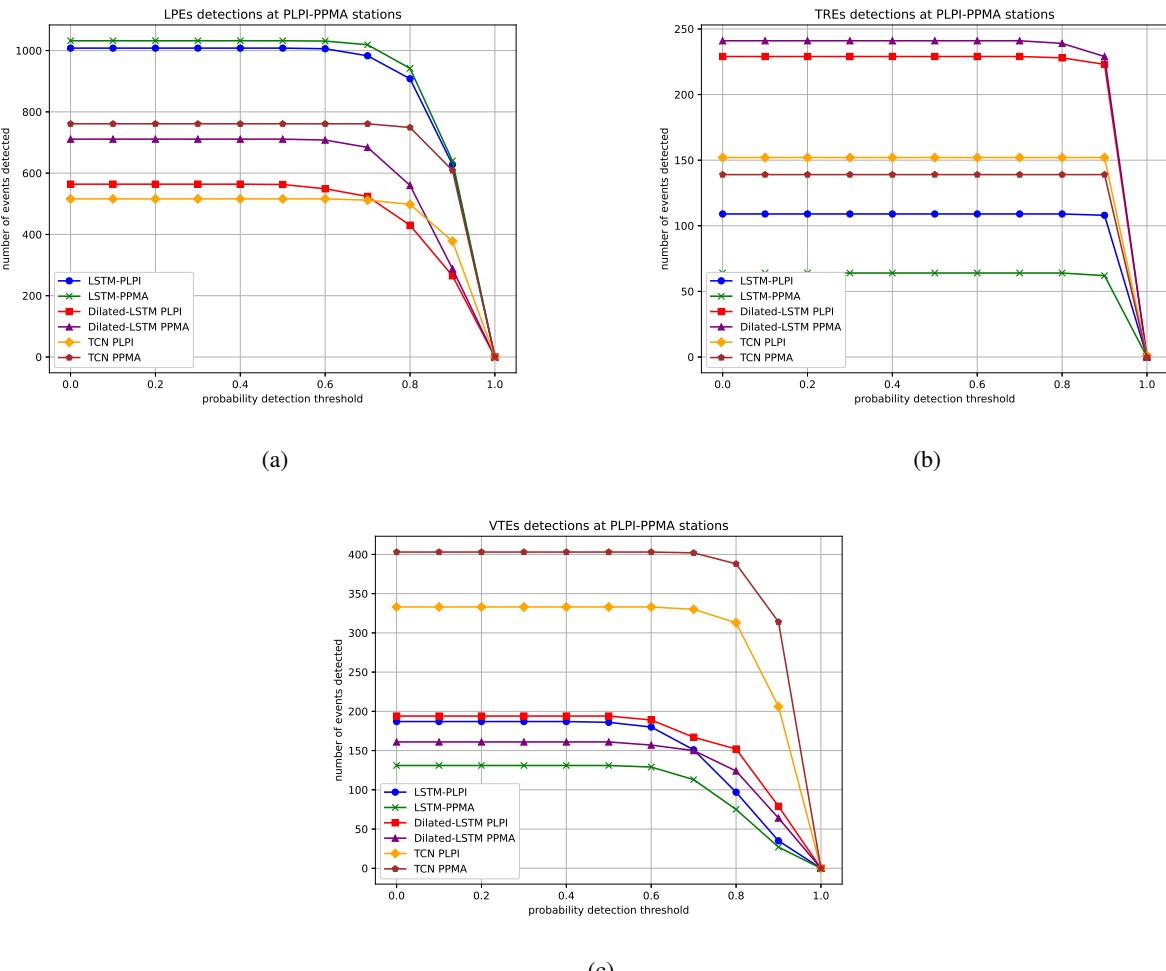

**Figure 9.** Evolution of the number of detected event at the seismic stations as the belonging probabilities threshold varies. A) Total number of LPEs detected at both stations. B) Total number of TREs detected at both stations. C) Total number of VTEss detected at both stations.

Regarding the detection of events identified by the systems but not annotated in the catalog, on average, RNN-LSTM and Dilated-LSTM detected approximately 60 earthquake-type events, while TCN identified over 150. Figure 11 presents a couple of examples of such earthquakes. The PSDs reveals that they share characteristics consistent with those of earthquakes. However, as indicated by the probabilities shown at the top of the figure, their partial similarity in spectral content prevented them from being classified with higher confidence.

Finally, it is important to discuss the recognition of events different to earthquakes, for which there is no available information to contrast the results. Figures 9a and 9b show the number of LPE and TRE events recognized by the systems, along with their corresponding membership probabilities. From these figures, it can be concluded that the number of detected events is high

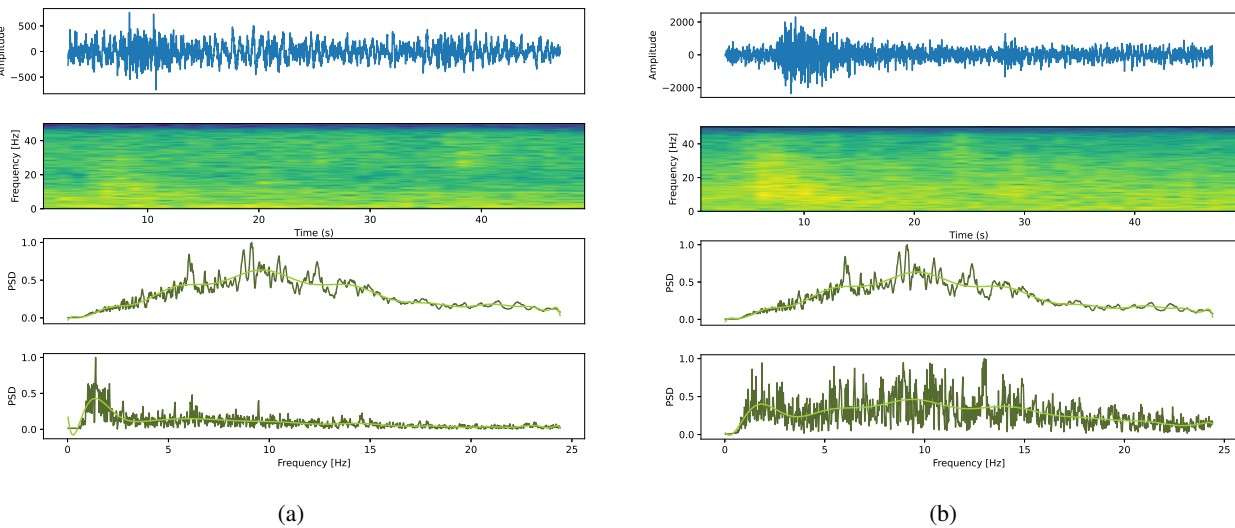

**Figure 10.** Example of two earthquakes annotated in the catalog that were not detected by any of the 3 reference systems. a) Spectral analysis of an undetected earthquake, where a clear attenuation of energy at high frequencies is observed. b) Spectral analysis of an undetected earthquake, where an energy distribution in intermediate frequencies and attenuation at high frequencies are observed.

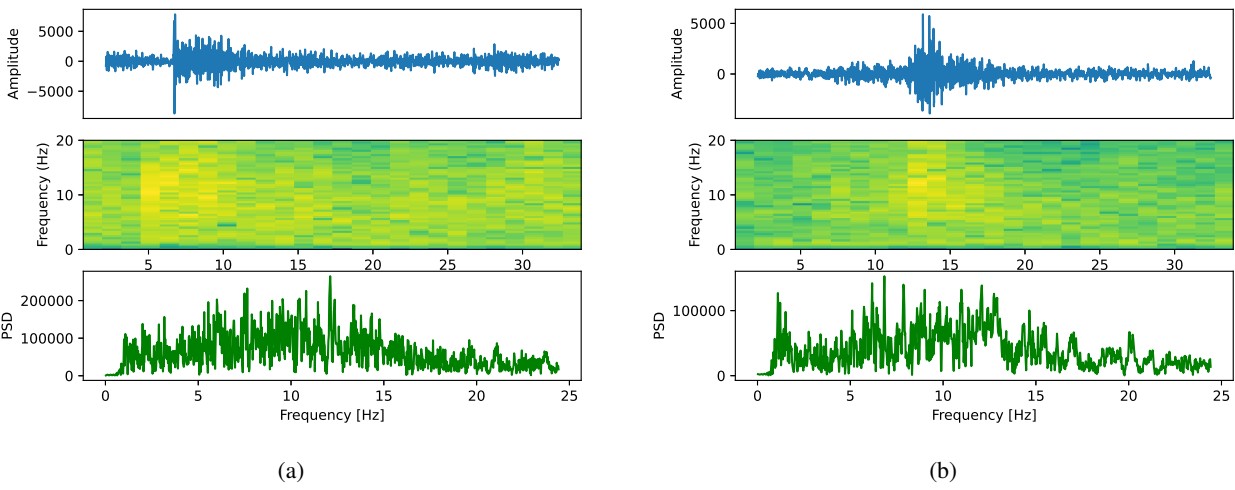

**Figure 11.** Example of two earthquakes not annotated in the catalog that were detected by the 3 reference systems with probabilities ranging from 63% to 78%.

for both categories, and the assigned membership probabilities are also relatively high, ranging from 80% to 95%. Unlike earthquakes, where high-frequency energy from external factors can lead to errors, TRE and LPE events are highly distinctive and well-defined at low frequencies. Since the systems were trained using parameter vectors based on logarithmic scale filter banks, which provide higher resolution at low frequencies than at high frequencies, the analysis of energy distribution across

low frequencies is highly reliable. Figure 12 shows an example of the LPE and TRE detections. As shown, these events were recognized with very high probabilities. Analyzing their spectral content, waveform, and energy reveals a perfect correlation with the characteristics of the prototype events on which the systems were trained, as illustrated in Figure **??**. Therefore, we can conclude that a large percentage of the detected TRE and LPE events correspond to prototype events from MASTER-DEC, which indicate the associated source mechanism of their label. It will be the responsibility of the volcano experts to analyze
whether these detected events share the same source mechanism or whether they should be re-labeled before pre-training the systems to adjust to the volcanic environment under analysis.

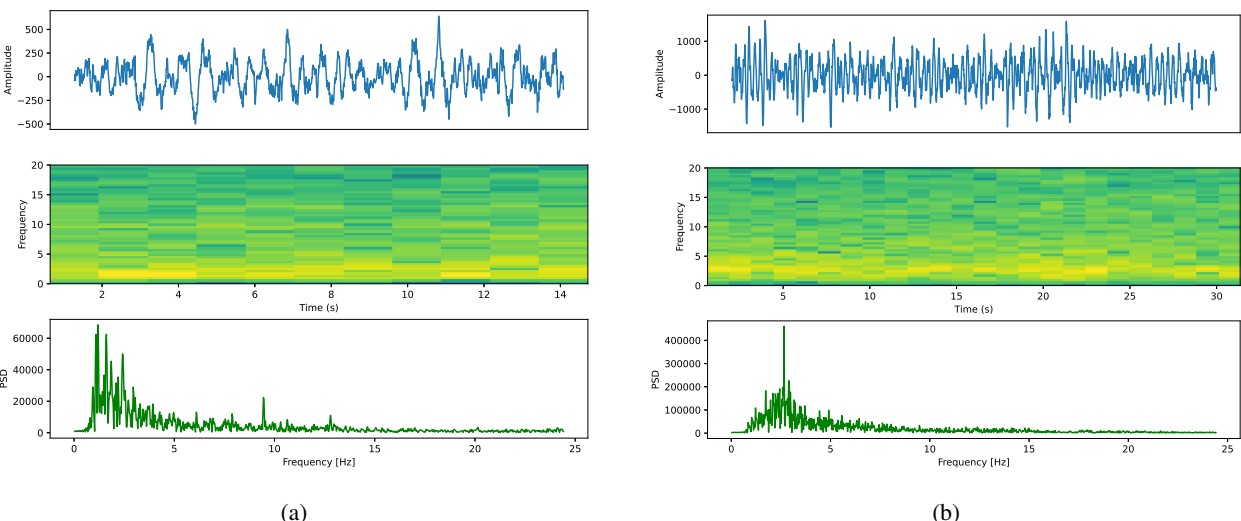

**Figure 12.** Example of a) LPE detected but not annotated and b) TRE detected but not annotated in the catalog.

### 5.4 Summary of Findings

The results presented in each experiment provide valuable insights into the development of automatic recognition systems with weakly supervised pseudo-labeling, highlighting both the strengths and limitations of the proposed methods. By synthesizing
the outcomes, we aim to offer a comprehensive understanding of how leveraging an existing automatic pseudo-labeler based on a master catalog can incorporate prior knowledge into the new dataset under review, which can inform future research and applications in the field.

Among the main strengths identified, the systems's ability to recognize previously learned prototype events, even in scenarios quite different from those analyzed during the learning process. This feature enhances its usefulness in reducing biases
when creating or improving catalogs. The results demonstrate that if systems would be trained across diverse volcanic environments with varied distributions of prototype events, recognition results could improve, suggesting good adaptability and, consequently, the construction of less biased catalogs in new scenarios and volcanic settings. However, the systems shows certain limitations, such as the detection of events that do not match any prototype, which could impact the final performance

of the re-trained systems. This primarily occurs because the pre-trained reference systems from which the pseudo-catalogs are built must assign a category to each analyzed window. Therefore, the systems will always assign a seismic category, even when the prototype is far from the signal under analysis. Once again, this challenge can be addressed by creating more comprehensive training datasets that describe different event distributions. Finally, another major challenge identified is the decision of the membership threshold from which events are included in the new pseudo-catalogs, indicating a need for post-analysis to assess the confidence of the detections, which would help distinguish between very sensitive or very specific pseudo-catalogs. Adjusting low probability thresholds will allow the creation of highly sensitive catalogs, which may result in many false positives—events that do not match the prototype. Retraining the systems with these catalogs could drop the performance and detection skills. On the other hand, a high probability threshold might not be sufficient to adapt the systems to the new volcanic environment.

## 6 Conclusions

This study provides the first comprehensive analysis of seismic catalog-induced bias when developing automatic recognition systems. We evaluated the ability of several monitoring systems trained using a master seismic catalog from Deception Island volcano to adapt to a new seismic catalog from Popocatépetl volcano through our novel, proposed weakly supervised framework. Our results confirm the robustness of data-driven approaches as a basis for the construction of short-term early-warning systems. However, quantitative and qualitative analysis confirmed that the reliability of a system is strongly biased by the undetailed coverage of the seismic catalog. While systems performance reached almost 90% per-frame recognition accuracy, intrinsic limitations when developing seismic catalogs led to extremely useful information describing the volcanic behaviour being ignored. Instead of simply learning to characterise volcanic dynamics by describing the latent physical model, catalog-induced learning can bias the system by discarding useful data describing volcanic dynamics. However, when a weakly supervised learning approach based on a master seismic catalog is applied, an unknown amount of information related to volcano dynamics is revealed.

This study raises important questions about the relevance of catalog-induced learning when developing new monitoring systems. Our results demonstrate that systems based on iterative weakly supervised or even unsupervised learning techniques could offer a more successful approach than supervised techniques under crude seismic catalogs. Therefore, we conclude that ensuring appropriate seismic catalogs and support for developing monitoring tools should be a priority to the same extent as applying new and more effective AI techniques. The use of more sophisticated pseudo-labelling techniques involving data from several catalogs could help to develop universal monitoring tools able to work accurately across different volcanic systems, even when faced with unforeseen temporal changes in monitored signals.

*Code availability.* Correspondence and requests for materials should be addressed to M.T.

*Data availability.* Correspondence and requests for materials should be addressed to M.T.

*Code and data availability.* Correspondence and requests for materials should be addressed to M.T.

*Author contributions.* All authors contributed to the conception of the study. M.T., C.B., conceived and conducted the experiments. All authors analysed and interpreted the results. All authors reviewed the manuscript.

*Competing interests.* The authors declare no competing interests

*Disclaimer.* TEXT

*Acknowledgements.* This work has been partially funded by the Spanish Project PID2022-143083NB-I00, "LEARNING", funded by MCIN/AEI /10.13039/501100011033 and by FEDER (EU), DIGIVOLCAN (PLEC2022-009271), SDAS (B-TIC-542-UGR20), and PROOF-FOREVER (EUR2022-134044). The authors would also like to acknowledge Instituto Andaluz de Geofísica, Instituto Volcanológico de Canarias (IN-VOLCAN), Raúl Arámbula-Mendoza and CENAPRED for providing the data from Deception Island, Popo and Tajogaite volcanoes supporting this work. Any opinions, findings, conclusions, or recommendations expressed in this research are those of the authors and do not 590 necessarily reflect the view of the funding agencies.

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
