# Peer review of "Could seismo-volcanic catalogus be improved or created using weakly supervised approaches with pre-trained systems?"

_Natural Hazards and Earth System Sciences, 2024_

## Author Comment (AC1)

**Reviewer#1:**

Dear Dr. Gordon Woo,

**We appreciate your response and are pleased to reply with the same sincerity and commitment.**

1. **Willingness to cooperate and the need for open data**: We would like to emphasize that we fully agree with the reviewer  that volcano monitoring should be, and indeed is being conducted from a multiparametric perspective (seismology, geodesy, gas, modeling...). Our experience in the field and our collaboration with various volcanological observatories around the world, as well as our experience in monitoring volcanoes during Antarctic campaigns, are evidence that we share this same approach. **However, from a purely scientific perspective, we believe that the field of volcanology must finally address the issue of Open Science**. Data acquisition is currently an expensive and intensive process, both in economic and temporal terms, primarily funded through national and supranational projects. However, the use of data is often closely guarded by the owners or managers of different observatories, which imposes significant limitations on data usage unless there is a strong connection with the observatories or data owners.Indeed, European data ownership and management policies hinder open access, limiting the natural development of disciplines like volcanic signal recognition. Unlike automatic speech recognition, which has seen significant growth over the past two decades, this field remains stagnant due to data access barriers.

2. **Need for robust catalogs:** Even assuming that monitoring data are publicly available, we still face the limitation of the scarce availability of robust catalogs that are exhaustively prepared and analyzed by specialists in the respective volcanoes, which serve as a comparative basis for our work. For example, IRIS has seismic records available for many volcanoes, but there are no robust labeled catalog associated with them, so we cannot validate the results of our approach without expert supervision. In addition, It is important to note that while there are many available and public seismic catalogs, many of them only correspond to very specific events like earthquakes. Although these catalogs are public, they do not offer the possibility to download and analyze continuous seismic waveforms. They simply report the occurrence of an event at a specific time and location. As we mentioned earlier, even in the case of public repositories managed by national and supranational entities, it is practically impossible to access the recorded seismic data. This issue is even more pronounced in Europe (in the United States, there is a clear trend towards sharing and making data available in public, durable, and high-quality repositories). Many volcanological observatories, despite being funded by public money, still have restricted access to their data. In some cases, this restriction lasts a couple of years, but it is not always enforced. **Therefore, we conclude that we are completely open to testing our proposal with as much data as the reviewer deems necessary. However, to do so, we need to know where to obtain such data and, of course, robust catalogs that can serve as a master reference for comparison**. In our work, we have used the Deception Island database and cataloged it as a master reference because its

development was carried out under the supervision of several experienced volcanic seismologists who specialize in the seismic monitoring of that volcano. Therefore, this database and catalog have been extensively reviewed and analyzed, serving as the basis for many other published works.

3. **Scope and limitations of our work:** our work does not aim to be a universal tool for creating catalogs for any volcano. Instead, it seeks to highlight a significant issue in volcanic monitoring from a seismic perspective and to provide a methodology through which, using techniques purely based on seismic observations, we can develop a robust tool that can easily adapt to different volcanic environments, creating effective and reliable catalogs that enhance our understanding of volcanic dynamics. To achieve this, each observatory will need to set up its system based on the available data or the similarities of its volcano with others that have public data and catalogs that support the development of the tool.

   a. To facilitate understanding of the proposed methodology, we suggest the following analogy based on speech recognition systems: Imagine we have a database and various semantic fields in a language with Latin roots. Now, suppose we train a speech recognition system using this database. If that system were used to recognize these semantic fields in other languages with Latin roots, we would achieve robust recognition because many of the words share roots and meanings. However, if the same system were used to recognize semantic fields in Icelandic, which has Old Nordic roots, our recognition would likely be very poor. To achieve proper recognition, we would need to apply a system trained in a related language, but even then, the methodology could remain the same for Latin cases.

   **This is the foundation of our proposal: provide a methodology that allows for the creation of catalogs and the inference of dynamics in a flexible and scalable manner**. The goal is not to create a universal tool that can recognize all volcanoes. This is impossible, as a volcano—continuing with the language analogy—could exhibit various "languages" during different eruptive stages and under different volcanic dynamics. Our work, therefore, simply offers a concise and reliable methodological framework from which knowledge can be created.

4. **Transferability and robustness**: Regarding transferability to other volcanic environments, a preliminary approach that served as a foundation for this work has been applied to the Bezymianny[1] and Peteroa[2] volcanoes, and is currently being applied to the La Palma volcano(in construction). In all scenarios, the results have been very satisfactory, and the outcomes are sufficiently robust and competitive compared to classical techniques. As mentioned earlier, this work improves such cited references by incorporating a self-adaptation mechanism that enables the creation of robust catalogs with less human intervention.

---

## Author Comment (AC2)

**Reviewer#2:**

**Dear reviewer#2, thank you for your valuable revision. In the following paragraphs, we respond to each of your suggestions.**

This manuscript can be better written, and its science better executed. As it stands, the manuscript appears too ambitious in scope. The results being presented are insufficient to deliver the intended scientific message, and the methods described lack sufficient details for reproducibility and related discourse.

In this work, the authors put a strong emphasis on using weakly supervised frameworks to improve seismo-volcanic catalogs for eruption forecasting or early warning. While there is novelty in the application of weakly supervised approaches in the context of volcano seismology, far too many details are left out on the seismology front, and the discussion in relating (improved) seismic catalogs to eruption onset or characterization is clearly absent.

Consider the following issues:

1. Throughout the manuscript, the authors utilize catalogues derived from Deception Island and Popocatepetl. Strong words are used to assert their robustness and quality, yet readers lack information on the related monitoring network, duration of observation (Deception Island), and contemporaneous volcanic activity for which each catalog was constructed. There is some passing discussion on "seismic attenuation processes" and "source radiation patterns", as well as the proposed underlying mechanisms behind each signal type from literature, but how do we know for sure if there is no information on the seismic network geometry, source-receiver distance, or eruption style being recorded?

**This work comes from a database that has been widely used in previously published studies. Throughout the paper, references describing the monitoring periods, the network's geometry, and the instrumentation itself have been provided. Additionally, these studies already contain clear references and figures depicting the waveforms and spectrograms of the signals used. We believe that including this information again in this work would be redundant. However, if the reviewer thinks it would be helpful, we will provide a detailed description of this information in the next version of the manuscript.**

2. Even if we were to assume that the catalogs are 100% accurate in their labels, this does not mean that they are necessarily suited for machine learning applications. When building a classification model, at least some care must be taken to balance the labeled dataset, especially if accuracies are being used as a metric. A perfectly labeled catalog could still be deficient in certain classes which the model hopes to classify. In such cases, the resultant biases need to be more thoroughly discussed.

**We agree with the reviewer's comment; however, the collection of events on a volcano is subject to its activity and the observation period during which the**

database was created. In this regard, we have attached Cumulative Distribution Functions (CDF) for the reviewer's reference, which represent the recognition level achieved by the classifier, once trained, on this database within the test set. **These probabilities can be used as an indicator of the confidence level in the recognition**. As can be observed, noise, tremor, and long-period events each achieve recognition rates above 90% probability of belonging in 90% of the evaluated events during the test period. This demonstrates that both the database design and the training process are adequate. It is important to highlight that the recognition of earthquakes shows a decline, but this is due to the inherent nature of the database, its imbalance typical of a real-world scenario, and the specific characteristics of the event. Given the results obtained here, we believe that transferring this knowledge to other volcanoes or to different observation periods of this same volcano is justified. We could say that our model is capable of robustly recognizing different types of events.

[Figure]

**Figure 1:** Cumulative distribution functions (CDFs) of probabilities for the predicted seismic categories. It is important to note that these CDFs are the classification

probabilities for the four types of volcano-seismic events—long period event (LPE), volcano-tectonic earthquake (VTE), background noise (BGN), and tremor (TRE)—obtained with different Recurrent Neural Networks using Long Short Term Memory cells (RNN–LSTM), Temporal Convolutional Networks (TCN), and Dilated Recurrent Neural Network using Long Short Term Memory cells (Dilated–LSTM) architectures with 2,3 and 4 hidden layers.

3. The authors introduce a set of (typical) labels used in volcano seismology, but fail to show clear examples from each dataset until late in the paper. An early figure showing the different classes from each volcanic setting (waveforms and spectrograms) would have been really informative on how the human experts had distinguished the different signal types, and what classifications they are hoping to achieve with their models.

**As we previously mentioned, we did not include the waveforms and spectrograms of the events because this database has been widely cited and referenced in previous works, which form the foundation of this study. Therefore, we considered that information redundant. However, if the reviewer deems it necessary, we will include and thoroughly describe this information in the next version.**

4. Although the algorithm framework is shown in Figure 1, there is data pre-processing and feature engineering step is too opaque. What does the "stream data" entail exactly? Why was a bandpass filter of 1-20 Hz chosen? How many stations are being used to constrain each label? Are we looking at the vertical component only? Is the instrumentation the same at Deception Island and Popocatepetl? What is the feature space here? If the features were indeed learned in the "deep learning" sense, it was not entirely clear to me how they were computed.

**The pipeline presented in Figure 1 has been widely used for the automatic recognition of continuous seismo-volcanic signals and has served as the baseline for several previously published works. In this context, stream data refers to the continuous flow of data that is generated and transmitted in real time. This data is produced constantly and is processed or analyzed as it is received. Consequently, it is connected to the recognition block (detailed in Figure B). Essentially, this illustrates that the data can be processed in quasi-real-time and continuously, meaning it is not composed of isolated and segmented events.**

**Regarding the bandpass filter, we chose the range of 1-20 Hz because the frequency content associated with the source models (proposed by Ibáñez, J.M. et al. (2000)) describing seismo-volcanic events is primarily located within this frequency band. Furthermore, since we are using a filter bank on a logarithmic scale, we enhance the resolution of the analysis in the lower frequency ranges.**

**Given that we are working with established databases, multiple stations are not being used in this work to establish or constrain the label of each recognized event at test time. This is something that could easily be incorporated and has been done in other studies. In summary, this work utilizes information from a**

**pre-established catalog (composed by labels obtained for different seismic stations), which serves as the basis for conducting comparative statistics.**

**With respect to instrumentation and data components, this study only considers the vertical components of the recorded seismic data. The instrumentation details related to the data from Deception Island, which form the foundation of this work, are referenced throughout the manuscript. As for the instrumentation details of the POPO database, we did not have access to them. We only have access to the catalog and preprocessed seismic data, where the instrumental response has been corrected. Both databases are recorded at a sampling frequency of 50 Hz.**

**Lastly, we address the feature space. As previously mentioned, each signal is windowed into 4-second segments with a 3.5-second overlap. For each window, a bank of 16 filters on a logarithmic scale is applied, and the first and second derivatives of each component of the 16-feature vector are calculated to enrich the contextual information. This results in a 48-feature vector per window. Thus, the feature space referred to in the text corresponds to the space of the parameterized features. The models use these features to extract hidden information in the form of nonlinear relationships and to recognize the events. In conclusion, this work applies a feature engineering process to help the models more effectively learn to characterize each event type and its temporal evolution. Therefore, while the model learns autonomously, the features have been previously extracted based on expert knowledge of the problem at hand.**

5. The authors mention that volcano monitoring and eruption forecasting involves a multidisciplinary approach. However, much of the manuscript is aimed at improving a catalog using machine learning techniques, which only involves the discipline of volcano seismology. The translation of this information into understanding unrest and hazards is absent. If the authors were to show that rapid catalog improvement could result in near-real-time characterization of real volcanic unrest, it would have made a far more convincing case. Unfortunately, this was not done or shown.

**This work focuses on improving seismic catalogs, which can assist in the multiparametric monitoring process supported by volcanological observatories. At no point has this work aimed to eliminate or downplay the importance of the multiparametric approach. On the contrary, this work is centered on the improvement or real-time construction of seismic catalogs based on the knowledge acquired from a master database. Even so, our work does not aim to be a universal tool for creating catalogs for any volcano. Instead, it seeks to highlight a significant issue in volcanic monitoring from a seismic perspective and provide a methodology through which, using techniques purely based on seismic observations, we can develop a robust tool that can easily adapt to different volcanic environments, creating effective and reliable catalogs that enhance our understanding of volcanic dynamics. To achieve this, each observatory will need to set up its system based on the available data or the similarities of its volcano with others that have public data and catalogs that**

**support the development of the tool. This was the intended goal of the work, but in light of the feedback received, we realize that we may not have been clear enough for our idea to be interpreted correctly.**

6. A key issue in volcano seismology machine learning literature is that volcanoes do not behave uniformly over time. Unrest signatures can vary from eruption to eruption, and from volcano to volcano. One way to make the applicability of this work more convincing could be to show its "temporal transferability" for one volcano in between different eruptive periods, before showing its applicability at a completely different volcano (i.e. "volcano transferability") as the authors have attempted in this work.

**As we mentioned earlier, our work does not aim to be a universal tool for creating catalogs for any volcano. Instead, it seeks to highlight a significant issue in volcanic monitoring from a seismic perspective and provide a methodology through which, using techniques purely based on seismic observations, we can develop a robust tool that can easily adapt to different volcanic environments, creating effective and reliable catalogs that enhance our understanding of volcanic dynamics. To achieve this, each observatory will need to set up its system based on the available data or the similarities of its volcano with others that have public data and catalogs that support the development of the tool.**

As the manuscript stands, it seems more suited for a journal like IEEE, where novel applications of machine learning techniques are discussed. In the context of NHESS or any other Earth Science journal, I would hope to see a more rigorous discussion of (1) the labeled dataset, (2) the different ML architectures, (3) the contextual volcanic unrest for which the seismic signals are observed, and (4) the relation between seismic catalogs and eruption forecasting.

**We believe that the novelty of this work lies not in improving an AI algorithm, but in highlighting the problem of training and creating automatic recognition models with biased catalogs, and how this issue can potentially be addressed using weakly supervised techniques. This is why we chose to submit this work to this journal. We are not attempting to create an AI model that universally generates catalogs. Our goal is to provide the reader with the perspective that a model with a high recognition rate in a given database may be biased by the information it has learned from that database. What might initially seem like a very robust model may not be as strong as we think. To illustrate this, we present a weakly supervised approach based on models trained with a master database, demonstrating that a model trained from scratch can learn to recognize the labeled information in the catalog while leaving useful information hidden. As we've mentioned, this approach is not intended to be universal, as there are many forms of semi supervised learning that can be employed. The idea is to show the volcanological community that the development of automatic seismic-volcanic signal recognizers can be biased, even when they have robust recognition rates. Consequently, the use of semi-supervised techniques could be a potential solution.**

To conclude, I would like to inform the reviewer that this work is a continuation of all our previous research works. Most of the questions you may have can be addressed by consulting these references. Additionally, many of our publications have appeared in IEEE journals, which tend to attract a more technically-focused audience, particularly in machine learning. This is why we decided to submit this work to NHESS, aiming to increase its visibility within the volcanology community.

---

## Author Comment (AC3)

**Reviewer#7:**

**Dear reviewer#7, We are very thankful for your thoughtful suggestions. Below, we present how we have addressed them.**

Summary

I have now had the opportunity to read and review the manuscript "How can seismo-volcanic catalogues be improved or created using robust neural networks through weakly supervised approaches?". Where the authors use machine learning techniques and a dataset from Deception Island as the master catalog to create and compare a new catalog for Popocatepetl in Mexico. While there are a lot of caveats and author interpretations in this research, the science, information and methods are interesting. The manuscript shows a small progress in ML techniques that can be used as the basis for future research. Below I list a few major comments for the review along with some line-by-line comments. Additionally, I would like to make a note about the subject matter. I feel this research would be more suited for a different journal. I was a bit surprised when I saw this manuscript was submitted to Natural Hazards and Earth System Sciences.

Major comments:

-What about other signals when building the model? There is a lot of source noise in volcanic terrains, how do these methods work when you introduce for example mass flows, edifice collapse, rock falls, ballistics, etc. In the same train of thought, how did leaving these out affect the outcomes. Furthermore, how about teleseismic earthquakes, how does the classification work on these?

**This question is very interesting, and we thank the reviewer for bringing it up. As we have mentioned in our responses to other reviewers and in the text itself, our proposal serves as a use case and an example of operation applied to the databases we have available. If, during the training of the master system, events that may appear in other volcanic environments are not considered, those events will not be recognized, as the master system will only recognize those events it has been trained on. This highlights the importance of sharing databases and catalogs from which master models can be built that can be universally used. In this work, we aim to illustrate how the system functions and the implications of catalog bias, as well as how it could be improved.**

**Our work is not a universal system and therefore cannot be applied to volcanoes with very different dynamics. This is why we have compared it to a database of similar events. However, the proposed methodology could indeed be used among volcanoes of similar nature, provided there are reliably labeled databases available. We believe this is the important aspect of the work; we are not offering a universal tool, but rather a universal methodology to address the significant issue of incompleteness in seismic catalogs.**

-"Early warning" is capitalized on line 24, but is not anywhere else, stay consistent throughout.

-Have you looked at the source depth of the signal, differing characteristics can occur depending on depth, you may have a problem similar to the attenuation issue.

**In this work, we do not focus on the analysis of the events; instead, we rely on the information contained in the catalogs, assuming that this work has been carried out previously. It is from these catalogs, which we consider reliable, that we conduct our experiments to validate our initial hypothesis.**

-I think the length of the training dataset is too short, how can we get a sense of what goes on at a volcano in just two months of data. Similarly, please explain why you are using a pre-eruptive model on a volcano that is in a phase of unrest. The difference in signal characteristics are going to be different, also the types of signals as I mentioned before.

-Some acronyms do not match, I tried to correct some in my Line-by-line comments, but it got too out of hand. A good example is the constant change between VT and VTE.

-While the frequency band of 1-20 Hz is fine, I am wondering about the difference between sensors. This paper does not mention any details about the sensors. What is the sampling rate of each sensor, are they all the same, are they different at different volcanoes. The details about each sensor are very important in knowing which frequency range can be used. Furthermore, how about is the sensor broadband or not. Is every signal from the vertical competent? If so how about using horizontal components?

**In the new version of the manuscript, we will attempt to include more detailed information about the instrumentation. However, we would like to emphasize that both databases have been sampled at 50 Hz and that the response of the systems has been removed, making it possible to analyze the signals effectively from the outset.**

-How did you choose which time window to use? What if there is a signal longer than 4 seconds, e.g. tremor, mass flow?

**The choice of the analysis window is inherited from previous studies. In those works, tests were conducted with different window sizes, and the best results were obtained using 4-second windows. While the results might vary with different window sizes, in this work, this parameter does not hold significant importance. We simply used the window size that performed best for Deception Island.**

**Addressing the reviewer's second question, it appears that the methodology has not been fully understood. The reviewer asks what happens when a signal longer than 4 seconds (the proposed analysis window) is received. The answer is:**

nothing changes. That signal will be segmented into 4-second windows with a 3.5-second overlap, and each window will be analyzed separately. For each window, we will obtain a label, and the combination of those labels will give us the trace analysis and event recognition.

The potential issue could arise in the opposite case, when an event lasts less than 4 seconds. In that case, we could have two events within the same window, and we must choose which label to assign. In our approach, during training we assign the label of the event with the longest duration within the window.

-You only train on one volcanic environment or master. I would like to see what the results would be if you used multiple environments from different volcanoes to make the master.

We agree with the reviewer that this test would be a great contribution both to the article and to the field. However, we only have access to the databases and catalogs analyzed here. In this regard, we are fully open to collaborating and testing our proposal on as many volcanoes as necessary, provided that we are given access to reliable data and catalogs from which conclusions can be drawn. This is an important issue in the field, as data is rarely made public, and access is often restricted.

-Most of the text in the methods section should be in the introduction. I suggest making a section in the introduction describing different kinds of methods people used in the past and then in the methods, explain the techniques you used for this research. Most of everything before section 3.1 should be in the introduction.

We will aim to implement these suggestions in the revised version of the manuscript if it is accepted for publication.

-I would like to see some comments about computing power and time. Some ML models and processes take lots of commuting resources as well as extended processing lengths. I would like to see a paragraph discussing these stats in the manuscript. What would I need to reproduce or do a similar computation at my observatory?

This is also an interesting point, and we appreciate the reviewer's input. From a computational perspective, since the base of the weakly supervised algorithm is already trained, its recognition process would be immediate, providing results within seconds. The only computationally intensive aspect could be the retraining using the pseudo-labeled database. However, considering that the models described here consist of several hundred thousand parameters, the training time, as observed in other referenced works along the manuscript conducting similar analyses, would range from several minutes to an hour. Therefore, it could be applied daily, weekly, or monthly, keeping the system updated in just a matter of minutes.

-I would like to know how much human work or time goes into creating this new catalog. Since it is a supervised learning technique, you still need human input and review, so how much time/effort are we gaining?

**Creating a catalog from scratch is a very time-consuming task. It requires reviewing and analyzing signals using various types of analysis. By applying an algorithm like the one we propose, the human operator can obtain a tentative catalog that includes the event type, start, and end times. Therefore, the operator's task becomes validating whether what the system produces is valid, which is much faster and simpler than manually isolating, cutting, and analyzing the signals.**

-In Lines 400-408: The training missed labeled tremor events, and you say this error was not actually an error, how can this be? The algorithm mislabeled, which means it did not work. Furthermore, reading your explanation further signals that this technique cannot be completed universally across different volcanoes. The attenuation affects you mention, points to the fact this would be difficult to do universally. A human had to go back in and review every event to make sure the event was labeled correctly, so how does this save time or is a better option?

**Lines 400-408 detail the reason behind the observed performance drop of all tested systems, ranging from 20% to 33%, due to discrepancies between manual and automatic labeling concerning tremor and LPE-type events. In Figure 4A, two high-energy, low-frequency events are labeled as LPE in the POPO2002 catalog. However, these events resemble TRE-type events from Deception rather than LPEs from Deception. Consequently, during recognition and pseudo-labeling, the system assigns TRE labels to both events. Statistically, this would be considered an error when compared to the catalog, but after consulting with experts, these events can indeed be labeled as TRE, since both their duration and waveform match a TRE-type event. Therefore, what is computed as an error (detection) is not truly an error upon evaluation.**

**Regarding the system's universality, as previously mentioned, the aim is not to provide a universal system but rather a universal methodology applicable to any volcano. In terms of review efficiency, a human operator would typically need to detect, isolate, analyze, and classify events. With our approach, the operator simply needs to validate the classification offered by the system, as it already provides classification, isolation, and event segmentation.**

-I would like to see more one-on-one comparison statistics in reference to Table 6. It is great the algorithm found more events but how many of the catalog events did it find and how many of the "human" events did it miss? Also, how many of these "new" events are real? Do the humans perform better for certain signal types and vice versa? How does each signal classification compare to one another.

**This analysis is highly interesting and could greatly enrich the work. However, such validation should be performed by one or more experts, and given the large**

number of recognized events, it would take some time to obtain reliable statistics. We are considering the idea of randomly sampling different types of recognized events, analyze them and gather statistics that could be extrapolated to the entire set of recognized events.

-There is a lot of repetitive nature of some paragraphs, try to go over the manuscript and cut some of this out.

We will make an effort to review the text and reduce the existing redundancies.

-A point on universality, every volcano is different even in the pre-eruption context of this manuscript. Some volcanoes do not even display signs of activity before erupting, so how can these ML techniques be considered universal at this point?

Authors did not intend to convey that our system is universal. In fact, throughout the text, we propose the premise that this methodology could become universal if multiple catalogs and volcanoes of different nature are considered in the training process with a master database: "The use of more sophisticated pseudo-labelling techniques involving data from several catalogues could help to develop universal monitoring tools able to work accurately across different volcanic systems, even when faced with unforeseen temporal changes in monitored signals." This sentence was included precisely because we understand that every volcano is different, even in the pre-eruption context. Therefore, by incorporating information from different volcanoes, the system could have more universal applicability and could be used in observatories where reliable information or catalogs are lacking. The only challenge, as we have already mentioned, is the availability and access to such data. Once again, we are open to collaborating on this effort, considering different volcanoes and catalogs. We invite the reviewers to join this initiative and work together on a project that leads to a universally applicable system.

---

## Author Comment (AC4)

**Reviewer#4:**

**Dear reviewer#4, many thanks for taking the time to review our work and provide such insightful advice. Below, we respond to your observations.**

The authors have taken on an interesting and challenging topic, using machine learning to classify volcanic seismic signals. However, there are several important areas where the paper needs improvement to better explain the methods and show how this research can be useful for volcano monitoring and eruption prediction.

1. The paper focuses mainly on classifying seismic signals but does not clearly explain how this helps us understand volcano activity or predict eruptions. The authors should add more detail about how these results can be used in real-world volcano monitoring systems, including other important data types like geodetic (ground movement) and geochemical data. This would make the study more useful for predicting volcanic hazards.

   **The aim of our work is to highlight an issue we have been observing for some time, which we believe needs to be addressed with great depth. There are many automatic seismic signal recognition systems in the literature, and almost all of them perform very well. However, the question is: Are the results of these models genuine, or do they simply reflect how well the systems have learned the information contained in the training catalogs? To explore this, we conducted a series of experiments that support our hypothesis. According to our results, when applying a system trained on a master database to a new database from a different volcanic environment, we find a significant discrepancy between the detected events and those annotated in the original catalog. Based on the approach followed by volcanological observatories, seismic catalogs describe the volcanic behavior or dynamics from a seismic perspective. A biased or incomplete catalog can lead to incorrect conclusions, and when comparing behaviors during future crises, these conclusions could also be biased. Therefore, obtaining more complete catalogs will help in understanding volcanic dynamics. As for the use of other types of data, this falls outside the scope of this study. It will be the task of the observatory to correlate information from different data sources and draw valuable conclusions. Without a doubt, this is an interesting idea for future work, but again we believe it is beyond the scope of this research. Additionally, we do not have access to the data to carry out such a data fusion approach. If the reviewer has access to these data, we sincerely express our openness to initiate the development of this idea through a scientific collaboration.**

2. The paper does not provide enough information about the seismic data used. The authors should explain more clearly how they collected the data, what each type of seismic event means, and how the events were labeled. A table or figure

showing how many events of each type were found would help the reader understand the data better. The authors should also show examples of different signal types earlier in the paper to make it clearer how the classification works.

**As we have previously mentioned to other reviewers, if this article is ultimately accepted for publication, in the next version, we will improve the description and visualization of the events that make up the different seismic catalogs.**

3. The paper lacks detail about how the data was processed. For example, the authors mention using a bandpass filter (1-20 Hz), but they do not explain why. They should also explain which components of the signal were analyzed (e.g., vertical component) and whether the same stations and equipment were used for both volcanoes. Providing these details will make the study more transparent.

**We once again agree with the reviewer's suggestion. In this version, we did not provide a detailed explanation of the data used. If the article is accepted, we will aim to address this suggestion in more detail.**

**Regarding the use of the band-pass filter between 1 and 20 Hz, we would like to clarify that this decision was based on expert knowledge of the problem and the parametrization scheme used. According to the source models and characteristics described in Table 1, volcanic seismic signals have discriminatory spectral content between 1 and 20 Hz. Therefore, the different types of events can be characterized based on the information within these bands. To do this, we applied a logarithmic scale filter bank, which increases resolution in the lower frequencies where different events exhibit distinct characteristics, allowing for more detailed analysis. Considering both premises, we believe the band-pass filter between 1 and 20 Hz is justified.**

4. The authors suggest that their method can detect more seismic events, but they don't provide enough examples of how this would help in real-time volcano monitoring. It would strengthen the paper if they could show how these improved classifications lead to better volcano hazard assessments or warnings. Additionally, volcanoes can behave differently over time. It would be useful to see if the model works well over different eruption periods or at different volcanoes.

**We thank the reviewer for their comment, as it gives us the opportunity to openly propose collaboration with organizations or researchers who have access to different volcanoes and eruptive periods, allowing us to validate the robustness of our proposal. Regarding the improvement of real-time assistance, in addition to what was previously mentioned in question 1, we commit to including this information in the new version of the manuscript.**

---

## Author Comment (AC5)

**Reviewer#3:**

**Dear reviewer#3, We sincerely appreciate your detailed comments, which have been very helpful. Below, we address your suggestions one by one.**

This paper deals with the automatic classification of seismic signals in volcanic environment. The authors suggest that weakly supervised machine learning approaches can be used to improve the detection and classification of signals, in comparison to direct transfer learning methods. Although the subject is very interesting, I agree with previous reviewers that the work must be improved before being considered for publication.

**We appreciate the reviewer's comment as it allows us to clarify the general idea of our work, which we may not have conveyed effectively. Our work does not propose that weakly supervised machine learning approaches are superior to direct transfer learning methods for improving the detection and classification of signals. Instead, we argue that an automatic recognition system (whether trained from scratch or using a transfer learning approach) with a high recognition rate may still be biased. This means that it might have learned the information in the training catalog very well, without showing signs of overfitting, but could still be limited in scope.**

**To demonstrate this, we designed an experiment in the context of transfer learning (TF). We trained a system initially on Deception data, then retrained it using data from Popocatépetl (POPO). The recognition rate exceeded 85% at the frame level (applying a grammar would further improve the results). However, when we applied a weakly supervised algorithm, we observed that while the system accurately recognized cataloged events, it failed to detect many other events present in the seismic traces but not recorded in the catalog.**

**Therefore, our point is not that weakly supervised learning is better or worse than TF for detecting or classifying seismo-volcanic events. Rather, we suggest that a weakly supervised approach, applied to systems trained on less biased catalogs, can be a robust solution for addressing the issue of catalog completeness.**

1) As stated in RC2, the manuscript lacks a description and a discussion of the phenomena to put the results in the perspective of volcanic monitoring and eruption forecasting. In its present form, you focus on signal classification, and not on the understanding of "volcano dynamics" as stated in the conclusion (l.455). A more detailed description of the data acquisition and catalog construction methodology is also missing.

**First of all, we would like to emphasize that our work focuses on recognition (detection + classification), not just signal classification. This is crucial because classification inherently involves a process of isolating and segmenting potential events that will later be classified. This process is very expensive, which is why we operate under the premise that existing catalogs, although highly reliable, are incomplete.**

**Secondly, when we mention that a more complete catalog can improve the understanding of volcanic dynamics, we are referring to the idea that having a more detailed view of the frequency and occurrence of events during different volcanic phases (eruptive, pre-eruptive) allows observatories to enhance their knowledge of the volcanic dynamics associated with their volcano. As is well known, volcanological observatories count seismic events during risk assessments and compare behaviors with previous eruptions. By increasing the quality and completeness of seismic catalogs, this can help deepen the understanding of the volcano's behavior and apply this knowledge during future crises.**

2) In the introduction, the authors mainly rely on their own publications for transfer learning approaches (e.g. "Based on our experience" l.86), but are they really the only team working on transfer learning methods for seismic signal classification? It is not clear either the extent to which this paper is novel compared to previous works on the same subject by the authors research team (citations l.86), or to other applications / studies in the litterature. The litterature review in the introduction is also, in my opinion, lacking key elements, in particular regarding existing fully unsupervised approaches. They have proven effecetive in volcanic context (e.g. Steinmann et al. (2024). Machine learning analysis of seismograms reveals a continuous plumbing system evolution beneath the Klyuchevskoy volcano in Kamchatka, Russia. JGR: SOlid Earth, 10.1029/2023JB027167). What are the limitations of fully unsupervised machine learning? How is your approach complementary? Similarly, you only refer to recent publications on early-warning systems based on seismic monitoring (Rey-Devesa et al. (2023) : you must be more explicit on the different approaches, and use more references.

**In the field of automatic seismo-volcanic signal recognition, there are other colleagues making significant contributions. However, when we refer to "our previous experience," we are not suggesting that only our group works in this area. What we want to highlight is that our group has extensive experience in this field, and thus, the problem addressed is well-established and thoroughly tested.**

**Regarding the novelty of our work compared to others in the literature, we can structure the response into two main blocks. The first, less technical and more methodological, focuses on highlighting the issue that while AI systems can effectively learn from the information contained in catalogs, this does not necessarily mean that the system is correctly classifying all the information within the seismic trace. This issue is what we refer to as bias. The system simply adjusts to the information learned from the catalog. Thus, the more descriptive and complete the catalog is, the less bias the system will have. However, creating less biased and more complete catalogs is a very costly and complex task. Therefore, a methodology is needed to facilitate this process. This leads us to the second block, which is more technical and less methodological, where we propose using a system trained with a defined and minimally biased catalog to**

**obtain reliable preliminary catalogs, which, with minimal effort, help improve the construction of less biased systems.**

**Finally, we would like to mention completely unsupervised approaches. These approaches have been successful in various fields, and automatic volcanic signal recognition is no exception. However, it is important to highlight that completely unsupervised approaches would assist us in several ways: 1) Clustering data into clusters or categories based on similarities. This would help with the task of clustering different windows and, consequently, events. However, this clustering would inherently require subsequent expert evaluation of each cluster, which essentially equates to supervised learning. 2) Exploring data and discovering effective underlying features that aid in the detection and classification process. In our case, feature engineering based on filter banks, derived from expert knowledge and experience, helps us avoid developing a preliminary stage to extract features with unsupervised systems and then use them in a supervised training process with catalogs as labels. 3) The use of unsupervised techniques deprives us of the expert knowledge contained within the catalogs themselves, which is crucial in a complex problem where signals are influenced by various external factors. This knowledge is vital and we believe it cannot be discarded from the training process. In summary, while unsupervised techniques can be complementary, our work relies on techniques that incorporate human expert knowledge as a foundational element, which can be transferred across volcanoes.**

3) The description of the catalog must be improved. As suggested by other reviewers, you need to explain more clearly what the different classes of seismic events correspond to, both in terms of physical processes and features used for classification. Their names must be homogenized (e.g. use only GAR or BGN for background noise), and you should show in a single figure / table how many events of each class there are in the two catalogs. As this is not done, it is sometimes difficult to understand your results (e.g. what are the 5 and 7 seismic categories used in Table 2)? In the same perspective, the features used for classification must be given, as well as the methodology used to compute them. It is also not clear to me if the catalogs associate labels to successive and constant time windows on the full signal, or on time windows of various lengths, defined manually, and corresponding to specific events.

**We will aim to improve the description of the catalogs and the presentation of the results to enhance their clarity. In the preprint submitted, we included the event names (categories) and the number of events exactly as they appear in the original catalogs. Regarding the physical description of the different events, in this work, we have followed the description proposed by Ibañez et al. (2000), where events are categorized based on their waveform and spectral content, as the catalogs reference the events using this categorization. However, in Table 1, we included other approaches for comparison purposes and to aid in comprehension. As we previously mentioned, in the new version of the manuscript, we will try to include an image that shows both the waveform and the spectrogram of each signal. This was not done in the first version because the**

papers describing each event were cited, and we considered this redundant information.

Regarding the meaning of the 5 or 7 seismic categories in Table 2, this information is related to the concept of Transfer Learning, which might not have been clearly explained in the manuscript. We will now try to clarify its meaning in simple terms. The aim of the article, as mentioned, is to provide an effective and straightforward solution to the problem of obtaining robust catalogs. To explain the challenges associated with the bias in the catalogs used to develop robust models, we rely on the concept of Transfer Learning. Essentially, this concept describes the idea of using knowledge acquired by a system in one domain and applying it to another, related or unrelated, domain. Therefore, the model trained on a minimally biased catalog (master) comprising 5 seismic categories serves as a starting point to train another model in a different volcanic environment with the same or different seismic categories. In this context, we have two alternatives for training the new model: 1) keep the number of seismic categories from the source domain in the target domain or 2) change the number of seismic categories in the target domain (in our case, with neural networks, this involves changing the number of outputs in the output layer).

Since the seismic categories of the POPO volcano correspond to those of Decepción (with the exception that some are divided into subcategories, such as the TRE event, which is divided into 3 subtypes) and some additional categories are considered, we designed two types of experiments: 1) Group all event subtypes into one and maintain the number of seismic categories from Decepción in the new model, ignoring the events present in POPO but not in Decepción (5 categories), and 2) Group all event subtypes into one and change the output layer of the new model, adding the events that are present in POPO but not in Decepción (7 categories). Therefore, the results shown in Table 2 reflect the recognition and adaptation percentages of the original POPO catalog when applying a Transfer Learning approach from the model trained with Decepción, considering 5 and 7 seismic categories.

Regarding the use of labels during training, we applied a windowing approach to facilitate capturing the evolution of signal information. For this, we simply used the information contained in the catalog and associated the label of each window with that information. Suppose that in the catalog, a signal is composed of 3 different event types, all of different durations: BGN-VTE-BGN. Our approach would window the entire signal into segments of a given duration and overlap. Since the catalog contains the start and end information for each event, we can associate the label of each window with the event type annotated in the catalog, as we know the start and end times for each window. Finally, each window is parameterized using a filter bank on a logarithmic scale. This allows the models to be trained with signals of varying durations that contain very different information.

4) The methodology of the weakly supervised learning must be more clearly explained, at least in an Appendix. It is not clear how the assumptions stated l. 236 to 241are important, and how the results can be interpreted if they are not verified (l.242-243 -> are the marginal distributions indeed the same? l.247-249: I don't understand the logical link suggested by "therefore", between the assumptions and the possibility to use weakly suppervised learning). Figure 1 must also be improved. In particular, the iterative refinement process is not displayed. Following remark 3), it is also not clear in the Figure what the signal in B) corresponds to : a portion of the signal identified manually in the catalog, or continuous data? For the same reason, it is not clear to me what the "dataset" mentionned in the text and in D) corresponds to. You should also explain how you define the threshold used for the drift adaptation method (l.266), and how you choose to stop the iterative refinement (what is the "desired result" l.270?). More generally, there are many terms that are technical and could be clarified for non experts readers (e.g. "self-consistency" (equivalent to accuracy?), "softmax", "argmax softmax", "confusion", accuracy" ...).

**We will attempt to more clearly and thoroughly explain the proposed methodology, either within the manuscript itself or in an appendix. Regarding Figure 1, we will improve its description. However, we would like to clarify the reviewer's concerns by briefly explaining the figure: We start with a model trained in a given volcanic environment, Deception, and aim to obtain a pseudo-catalog of events for a volcano in a different environment, POPO. Therefore, the signals in the POPO dataset are analyzed using the model trained on Deception. The events recognized in POPO by the model trained on Deception, with a user-defined membership probability, will be included in the new database with the label suggested by the Deception-trained model. In this way, after analyzing the POPO dataset, we will have a new dataset with pseudo-labels assigned by the Deception model. This database and the new labels will be used to retrain the model. Once trained, the process is repeated from the beginning (an iterative process), so that in each iteration, events are recognized with greater probability, as the model has been trained with data from this new environment.**

**Regarding the technical terms, we will aim to provide clear and formal definitions so that any reader can understand their meaning. In the initial version, we did not do so because we believed these terms were well-known within the community and that it was not strictly necessary.**

5) The presentation of the Results can also be improved. As mentionned above, as the event classes are not clearly defined, it is not always easy to understand the results. Regarding to the cross-validation : you use it for the direct TL approach, but not the weakly supervised TL, why? Besides, isn't it interesting to look at the variations of the accuracy to see if the learning is stable or not, in addition to the mean accuracy? Table 5 must be presented and discussed in more details : it is referred to only in the discussion and after a reference to Table 6.

**We will aim to improve the readability of the results in the revised version of the manuscript. Regarding the use of cross-validation, in the direct application of Transfer Learning (TL), it is used to assess how well the systems fit the original POPO2002 catalog. When applying weakly supervised learning, we could also use cross-validation, but this would involve training 16 models (4 models for each of the 4 models that make up the cross-validation for TL). Since this work is primarily focused on highlighting the issue of bias in models trained on catalog information and how that bias can potentially be reduced, we believe that, as the models derived from the weakly supervised approach show very similar performance, applying cross-validation in this context would yield very similar results. By observing the difference between the events detected using the weakly supervised approach and those originally annotated, we conclude that the use of any of the models obtained through cross-validation would result in similar outcomes.**

**Regarding Table 5, we included it in the manuscript to give the reader an idea of how the event labels differ between those assigned by the weakly supervised approach and the original POPO2002 catalog annotations. Essentially, what is observed is that all tested systems detect many events in traces that were labeled as BGN in the original catalog. Many of these events, if they meet the user-defined threshold criteria, will be included in the new training dataset, and the model will thus be able to find many of these events in the test partition. This is why Table 6 shows a significantly different number of initially annotated events compared to those recognized by the weakly supervised approach.**

6)A major argument of your work is that catalogs can be biased, and that the accuracy of ML learning techniques should thus not be the only criterion of a classifyer efficiency. Although this is worth saying it, you say it repeatedly. E.g. l.360 to 375 is only about this point and is only a repetition of what is already said in the introduction : I don't see what is new in this paragraph. Another unclear point is that you present the Popo2002 catalog as a high-quality catalog, but then suggest that some VTE, LPE and TRE are missclasified (l.396-708). Then, you refer to difference between the catalog and your calssification as "an 'error' that was not really an error" (l.404), but as I understand it is is based only on the judgment of "a geophysical expert" (l. 406). Why is this jugemnt more reliable than the classification obtained thanks to the "quality of the human team" mentionned l.103?

**We appreciate the reviewer's comment, as it provides us with the opportunity to clarify the philosophy of our work and the conclusions we have drawn from it. It is true that throughout the manuscript we emphasize that the classification results, rather than highlighting the system's robustness, emphasize how well the system has learned the information contained in the catalog. This explains why systems trained with the POPO2002 dataset achieve results around and above 85% (Table 2).**

**However, we do not believe this creates a contradiction when we state that the POPO2002 catalog is of high quality, but there are events that were not included in its construction. As mentioned in the introduction, catalog creation is a very time-consuming task, and during periods of high activity, it is humanly impossible to analyze each registered event. In many cases, these events are grouped into more general traces. This does not conflict with the idea of a high-quality catalog, as we have shown that the models can learn the catalog's contents with high performance, demonstrating their self-consistency.**

**However, when we apply our algorithm and detect an event not annotated in the original catalog, from a statistical standpoint, this event is considered an insertion, which counts as an error in the confusion matrix. This is evident in Table 5. Upon expert analysis, though, this inserted event is not considered an error since it is correctly recognized.**

**In conclusion, while the original catalog is of high quality, given the complexity of the problem, it is humanly impossible to meticulously analyze all the information in the seismic trace. As a result, reliable but incomplete catalogs are typically produced.**

7)You state l.355 that you have "verified" that weakly supervised approaches could "significantly enhance the detection and identification capabilities". However it is not clear what the enhancement refers to. In comparison to what? How do you quantify the enhancement? Thus, I don't think the Results section illustrates correctly this sentence. As a metter of fact, Table 4 shows that a weakly supervised approach improves the accuracy in comparison to a direct application of the MASTER-DC classifier to the Popo database, but then you state that the accuracy is not necessairly a good indicator of a classifier efficiency. On the contrary, if you consider the accuracy as a robust indicator, then the classic TL approach yields better results than the weakly supervised approach (compare Tables 2 and 4).

**Regarding the previous point, we would like to emphasize that we were not able to clearly convey our idea in the text. When we conclude that weakly supervised approaches can improve detection and classification capabilities, we base this on the results shown in Table 6. As can be seen, the number of originally annotated events and the number of recognized events differ significantly. We believe that our conclusion is founded on a comparison with the original catalog, and the concept is clear.**

**Table 4 does not show that a weakly supervised approach improves accuracy compared to the direct application of the MASTER-DC classifier to the Popo database. Instead, Table 4 shows that if we compare the performance of a weakly supervised approach with the information annotated in the original catalog, the performance is quite poor, not exceeding 65% (while a TL approach can reach up to 85% or more). However, when we closely analyze the results, we observe that this performance decline is due to the insertion or detection of many events that, for various reasons, were not initially annotated in the catalog.**

Therefore, what our work proposes, and what we want to emphasize, is that while accuracy is a measure of how robust a system is, in our specific and complex problem, accuracy might instead reflect how well the system aligns with the knowledge contained in the catalog. These two concepts are not incompatible, as a model achieving 85% or more average performance could be statistically robust, but a model with 65% accuracy could also be robust if the insertions it makes (if correct) lower the accuracy without compromising robustness. We recognize that explaining this concept is quite challenging, which is likely why our work has not been fully understood by the reviewers.

8) Another argument you put forward is that the weakly supervised approach allows to detect more events than in the catalog. However this is expected, as you apply your classifier to more time windows than the Popo2022 catalog (2139 labelled events in the Popo catalogue, more than 20,000 times windows labelled with your classifier). Besides the number of labelled events is different depending on the classifier (compare sum of columns in Table 6), why is that so? The real question, that I think you don't answer fully in your paper, would be : do weakly supervised classifiers allow to detect more events, and in a more robust way, than classical ML methods and direct TL approaches?

Once again, we would like to thank the reviewer for this comment, which provides us with an opportunity to further elaborate on our results. As the reviewer has pointed out, it is clear that we have not successfully conveyed our idea. Our detection results are not higher because we analyze more windows than events, as the reviewer suggests.

Our approach analyzes the signals by windows. For each window, a label is assigned, but each window is not considered a detected event. Instead, consecutive windows with the same label, meeting the minimum average duration required for each event type in the Deception catalog, are grouped into a single event. In other words, if our approach detects several consecutive windows for the same event, but their combined duration does not meet the average duration imposed by the master catalog, that event is considered unknown and is not even annotated in the table.

Thus, the table reflects recognized events, not windows. This explains why the total number of events in the columns of Table 6 differs for each system. Each system detects events differently. If we were considering windows, the three columns would add up to the same number, as the reviewer suggests. However, that is not the case, nor does such an approach make sense in our context. This methodology is thoroughly described in the references included in the text.

Finally, we would like to emphasize that our work does not argue that weakly supervised classifiers allow the detection of more events, or in a more robust way, than classical ML methods and direct TL approaches. Our argument is that by applying a weakly supervised approach and leveraging TF, we can build less biased catalogs and help develop rapid and robust automatic monitoring systems for seismo-volcanic signals.

For these reasons, I suggest the authors to review thouroughly their work before considering it agin for submission. Their work is of great interest and importance. However, its implications both in terms of pure classification problems, and in terms of volcano monitoring, are not sufficiently investigated.

---

## Author Comment (AC6)

**Reviewer#5:**

**Dear reviewer#5, we deeply appreciate your feedback, which has enriched our work. We now focus on your suggestions.**

The Authors apply different machine learning techniques to create, from a database of daily seismic registrations (if I understood well) of a certain volcanic area (Deception Island Volcano), the seismic catalogue of another volcanic area (Popocatepetl volcano) labelling the type of event following some criteria (not adequately and quantitatively described). To do this, the Authors use a high-quality database of seismic events, already labelled by human supervision, and collected in another volcanic area. The purpose is to reduce or eliminate the use of human work for labelling the seismic events in catalogue. The purpose is very important and interesting, since the increase of seismic networks in the recent years has the undebatable advantage of having increased the seismic monitoring both in volcanic and in tectonic areas but at the same time it has increased the number of data to be analysed by seismologists. So, an automatic system that can be able to detect and label seismic events in volcanic or tectonic context can be very useful if the system is reliable and it will reduce human supervision to a minimum or even it will eliminate it altogether, working as a human would do.

The Authors conclude that the three ML approaches produce different results and all of them are able to detect a number of events much greater than the existing catalogue (except in one case) and I think that this a very important and intriguing result.

I appreciate the work and the idea, but the manuscript has many problems that I try to list.

Reading the description of the work done, it is not clear how a researcher could verify the results and reply the work with its own database. The description of the method is very confused and only who has already used the same techniques could follow and understand the steps. Moreover, the description both of the method and of the data used is only qualitative and discursive, never detailed and quantitative.

**We appreciate the reviewer's constructive comment. In the next version of the manuscript, we will aim to detail the proposed algorithm more thoroughly and in a more didactic manner, including pseudocode for easier understanding. Additionally, we will make the source code available to the scientific community.**

The manuscript is full of non-useful repetitions and the Authors should do an effort to re-read the manuscript and be more concise. As an example: Line 281-294. This is a repetition of something already written in the manuscript. Line 170-184. This paragraph should be moved in the Introduction section and rewritten to avoid repetition.

The same acronyms are referred to with other acronyms. As an example, the three techniques employed to achieve the purposes of the manuscript (specified in Line 184-186) are referred to ANN or to ML in different part of the manuscript, generating a

great confusion among all the acronyms. I suggest simplifying and reducing the use of the acronyms to the strictly needed.

**Once again, we agree with this constructive comment. We will strive to be more concise and avoid repetitions and redundancies in the new version of the manuscript.**

Regarding the used database, the Authors use a qualitative language that does not help to understand. Some examples:

Line 115. Where the original labelled database can be consulted? Is it already released?

Line 134. What do the Authors mean with "subset of data considered the most reliable"? Which criteria did they use for reliability?

Line 139. As above, what do the Authors mean with " the most representative and of the highest quality"? How do they measure the quality?

Line 149-154. Where the Popocatepetl 2002 catalogue can be found? A citation is missing here

Line 152. Which are the classes of event, adding a table can hep.

Line 229. The phrase "The target domain (denoted as Dt) is the Popo2002 dataset (whose available seismic catalogue will not be considered)" what does it mean? At line 149 the Authors state that Popo2002 consists of 4,883 events, what type of event they are? I can understand that the data used for the target domain is a subset of the Popo2002 collected excluding earthquakes. Is it true? I think that the authors should be more concise and clearer in describing both the technique and the data used.

**All of these suggestions will be taken into account and included in the new version of the document to enhance understanding and readability. We are committed to providing a more detailed and straightforward description of these concepts. We will also include references to both catalogs**.

In conclusion, I suggest publication after a deep rewriting of the manuscript that does justice to the work done, makes it understandable also to those who have never used the specified ML techniques before and makes the proposed method replicable for other interested scientists.

---

## Author Comment (AC7)

**Reviewer#6:**

**Dear reviewer#6, we deeply appreciate your feedback, which has enriched our work. We now focus on your suggestions.**

This is a solid paper that reports on systematic evaluation of volcanic earthquakes using several machine learning (ML) techniques. I should state at the outset that I am a volcano seismologist with more than a decade of experience, but I have never directly used any ML or AI techniques. Hence my comments are of a more general nature.

The approach in the paper is thorough. The results are repeatable, which is good. The results also show that it is possible to get more out of the data, which is always welcome. The procedures are efficient, so it is possible, in principle, to obtain similar results in much less time than it takes an experienced geophysicist to manually process the data. But this brings up a philosophical point: what is reality? Manual or ML? I would think that a manual effort by an experienced person would be the benchmark, and ML results would be judged relative to them. The paper mostly does this, with a few exceptions.

**We thank the reviewer for their constructive comments and take this opportunity to clarify an important aspect of our work. The reviewer raises a philosophical question: What is reality, human or ML detection? We believe this reflection is fundamental and requires a detailed response.**

**From our experience, both are real and complementary. On one hand, manual effort by an experienced person serves as the benchmark but has inherent limitations, influenced by emotional and intellectual conditions due to the subjective nature of human supervision. On the other hand, the ML version is based on human benchmarks but has the advantage of conducting a systematic and rapid analysis of all information with a unified criterion, free from factors beyond what has been learned. Therefore, we can conclude that neither is more real than the other; rather, the automated method based on expert knowledge offers a competitive advantage in terms of the universality and speed of analysis, allowing for the construction of more complete catalogs but not necessarily more real ones, since those obtained manually are also real but subject to the constraints of the specific task and eruptive state in which the catalog is created.**

The paper is mostly well written. Here are a few corrections keyed to line numbers:

36 – the V in VLP stand for very, not ultra

44 – is this frequency in Hz or frequency of occurrence?

51-52 – inconsistent use of parentheses ( )

143 – confront? Odd word choice

184-186 – at this point I had a hard time keeping all the acronyms in my head. I suggest adding a table of acronyms.

233 – spacing

Table 2 – are all values percentages? Needs better labeling

Table 3 – add bolder vertical lines between three main sections; spell out the abbreviations in notes at the bottom of the table (Tables should stand alone)

Table 5 – same comments as Table 3

357 – used

381 – "unbiased" but how determined? This is the place that made me rethink the question of what is reality, as described above.

Table 6 – is all the time with no events equal to the background?

**In the new version of the manuscript, all these suggestions will be addressed to improve the readability and understanding of the work.**

Overall, the paper is in good shape and is suitable for publication with minor revisions as indicated above*.*

---

## Author Comment (AC8)

**Reviewer#8:**

**Dear reviewer#8, We are very thankful for your thoughtful suggestions. Below, we present how we have addressed them.**

General Comments:

The manuscript presents a highly relevant approach that combines machine learning with weakly supervised methods for seismic-volcanic event detection. The application of these techniques to geophysical event detection is an exciting and promising field of study, and I commend the authors for their effort in tackling such a complex problem. The subject matter is particularly valuable given the growing interest in leveraging machine learning models for natural hazard monitoring, and the use of weak supervision opens new possibilities for working with limited labeled data, a common challenge in seismology and volcanology.

However, while the approach is interesting, the manuscript, in its current form, requires substantial rewriting to improve clarity, structure, and the strength of its arguments. There are several critical issues that need to be addressed before the manuscript can be considered for publication:

1. Methodology Section Reconstruction: The methodology section lacks sufficient clarity and structure. Key concepts such as UMAP, the Leave-One-Out cross-validation method, and the iterative processes involved in the pseudo-labeling task are either insufficiently explained or poorly integrated into the overall narrative. The methodology needs to be rewritten to clearly define these elements and their role in the overall framework, ensuring that readers can follow the steps taken in the model development and evaluation process.

   **We will aim to improve the wording of this section to streamline the reading and comprehension of the proposed ideas. To achieve this, we will include a schematic representation of the algorithm itself and define some of the mentioned concepts to make the work more self-contained.**

2. Justification for Using a Single Dataset in Transfer Learning: The authors attempt to justify the use of a single dataset in their transfer learning approach, but the arguments presented are not convincing. As the authors themselves note, 'it could change when using a different test dataset,' suggesting that model performance may not generalize well to other geological settings. The authors need to make a stronger case for why the use of a single dataset is valid for this weakly supervised learning approach. Ideally, the manuscript should explore the potential limitations of this approach or, alternatively, incorporate multiple datasets from different volcanic settings to demonstrate broader applicability.

   **We completely agree with the reviewer's suggestion. The only reason our work focuses on a single volcano in TF is that we do not have access to other available seismic catalogs and data from which to build a comparative base. As we have extensively mentioned to the majority of**

**reviewers, we are fully open to testing our system with different databases and volcanoes of varying nature. We invite the reviewers to join this initiative and collaborate on a project that could result in a universally applicable work. Otherwise, we kindly ask for guidance on where we can obtain reliable seismic data and catalogs for further experimentation.**

3. Overall Structure and Writing Quality: The manuscript, though scientifically significant, suffers from poor structure and unclear writing, which detracts from its scientific contributions; this has resulted in several instances where key ideas are poorly expressed or ambiguously presented. A thorough revision of the manuscript is needed to ensure that the concepts and findings are communicated effectively. I suggest the authors consider restructuring the entire manuscript to enhance readability, focusing particularly on tightening the introduction, improving transitions between sections, and making the arguments in the discussion more robust.

   **We will work on improving the structure and writing of the article to meet the expectations of the several reviewers who have suggested this.**

In conclusion, while the study introduces an interesting and timely approach to seismic-volcanic event detection using machine learning, the manuscript requires significant rewriting to better articulate its methodology and address critical gaps in the explanation of its approach. I recommend a major revision to enhance clarity, strengthen the justification for key methodological choices, and improve the overall presentation of the research.

Specific comments & Technical corrections:

**We will address the key issues raised by the reviewer. The remaining suggestions will be implemented without further discussion in order to expedite the review process, as most of them are technical and grammatical corrections.**

- 1. Introduction.

     line 50:"Bayesian" misspelled
     line 99: ¿references for master dataset?
     lines 99-100: "*has already been successfully applied in different DL architectures*"; ¿references?
     line 102: references for the Popo dataset?, and ¿why it is of high quality?
     line 105: It would be very useful to provide more information about the volcanic dynamics observed in the proposed datasets, especially as machine learning developments and methodologies are evolving to incorporate physics-based input.

- 2. Seismic data and catalogues.

     line 125: "..on the applicationof HMM models, etc."; ¿references?
     line 130: "*While it is true that not all types of signals are present in this 'Master database', especially those associated with ongoing eruptive processes.*", so,

perhaps it would be important to have a master dataset that includes this information as well. It is crucial to incorporate datasets representing different stages of volcanic unrest and to clarify which specific stages the machine learning models are most useful for.

line 145: A more detailed description of Popocatépetl's volcanic activity is needed, including its cyclical behavior of effusive activity, dome formation followed by explosive events, tremor signals, and other relevant features.

line 148: Are there any references available for this group of geophysicists or their work?

Table 1: nice.

Data & sensors: It would be ideal to provide a clearer explanation of the types of instruments being used, including whether all components are available, sampling, etc., as well as details on the sensors. For example, are all instruments capable of measuring all types of events in both datasets? Nowadays, seismic networks are densified with a combination of broadband and short-period sensors, which may influence data quality, coverage, and distance to volcanic sources. The proximity of sensors to the volcanic source is critical, as it directly affects the resolution and accuracy of the recorded data.

**In the new version of the manuscript, all these suggestions will be addressed to improve the readability and understanding of the work**

- 3. Methodology.

lines 234 - 246: about marginal and conditional distributions: a need for clarity:

The authors' explanation regarding the assumptions of marginal and conditional distributions in the pseudo-labeling task could benefit from greater clarity. Specifically, they state that the marginal distributions of the source and target domains are assumed to be the same ( $P_s (X_s) = P_t (X_t)$ ), maybe implying that the input features (seismic windows) in both domains are similarly distributed? However, they also assume that the conditional distributions of the source and target domains are the same ( $Q_s (Y_s | X_s) = Q_t (Y_t | X_t)$ ), suggesting that the relationship between input features and event types is identical across both datasets.

Key Challenge and, Potential Problem?:

The text acknowledges that while the marginal distributions of the input features may be the same, the conditional distributions might differ between the source and target domains. This introduces a key challenge: even though seismic signals may "look similar" across different datasets (i.e., the marginal distributions are similar), the relationship between these signals and the seismic events they represent (i.e., the conditional distribution) may vary.

This discrepancy can create a potential problem when using pseudo-labeling and transfer learning techniques. If the model is trained assuming that the conditional distributions are the same, it may misclassify events in the target domain, especially if the seismic signatures there correspond to different types of events than in the source

domain. This issue could result in reduced accuracy and reliability of event detection in the target domain, undermining the effectiveness of the model's generalization.

Conclusion: The Need for Diverse Datasets

This challenge is crucial because it highlights a potential flaw in the transfer learning approach: the assumption that conditional distributions are the same across different volcanic settings may not always hold. To address this, it may be necessary to collect and incorporate datasets from a wider range of volcanic regions, where the relationships between seismic features and event types can vary. Doing so would enable the development of more robust models that can better generalize across domains, improving the accuracy and reliability of event detection in different geological contexts. This would strengthen the use of transfer learning techniques and ensure that models are more adaptable to varying volcanic behaviors.

**We fully agree with the reviewer's comment, which is why we once again encourage the reviewers to join this initiative so that we can conduct a study that includes a wide range of volcanoes and catalogs. This would help reduce the gap between domain distributions and bring us closer to a more universal model.**

Figure 1: bad quality figure in the PDF file. Do steps A, B, etc., correspond to the actual process in your proposed methodology? line 276: reference missing.

**In the new version of the manuscript, this suggestion will be addressed to improve the quality of the image.**

- 4. Results.

> line 291: review grammar ("..using as training..")
> line 302: The text on self-consistency should be explained and included in the methodology section ('*We apply the Leave-One-Out cross-validation method*').
> line 329: Should Section 4.3 be renumbered as Section 4.2?
> line 343: These iterations need to be clearly specified in the methodology section, as you mention the goal of "*until a reliable catalog is achieved*".
> Line 344: The authors mention that "*however, it could change when using a different test dataset*" which highlights an important point regarding model generalization. While their approach is based on a single dataset, this raises questions about its robustness across varying geological settings. To truly validate the effectiveness of the model, it would be crucial to demonstrate its performance using multiple datasets from different volcanic environments. By doing so, they could provide stronger evidence that the model can generalize across diverse conditions, rather than being tailored to a specific dataset. The authors need to convincingly argue why relying on a single dataset is sufficient, or alternatively, why incorporating multiple datasets might be necessary for ensuring broader applicability.

**In the new version of the manuscript, all these suggestions will be addressed to improve the readability and understanding of the work**

- 5. Discussion.

   line 357: It would be helpful to clarify the phrase "when effectively use" throughout the text to strengthen the main arguments. Perhaps the grammar could be reviewed in that sentence.
   line 365: Should Fig. 1 be renumbered as Fig. 2?
   2: UMAP should be introduced in the methodology section and connected to the general objectives.

**In the new version of the manuscript, all these suggestions will be addressed to improve the readability and understanding of the work.**

---

## Author Response (AR1)

**December 12, 2024**

**Editor-in-Chief**

Natural Hazards and Earth System Sciences

Dear Editor,

I am pleased to submit this cover letter regarding the original research article (nhess-2024-102) entitled "How can seismo-volcanic catalogues be improved or created using robust neural networks through weakly supervised approaches?" by Titos M., et al., for consideration in NHESS. We have carefully reviewed the feedback from **all eight reviewers** and greatly appreciate the time and effort they have invested in evaluating our work.

Creating a detailed response letter for this revision process has proven highly challenging due to the number of reviewers and the varied nature of their suggestion, which often overlap or conflict with one another. Furthermore, in our previous response within the discussion forum, we addressed each reviewer's feedback comprehensively.

In this letter, our goal is to highlight that we have made every effort to implement as many of the suggested changes as possible. Rather than responding separately to each reviewer again, we are submitting a revised manuscript alongside a detailed version highlighting the changes compared to the previous version. This approach enables each reviewer to directly assess the scope and impact of the modifications made.

Many of the reviewers' comments focused on improving the clarity of the methodology, enhancing the quality of the figures, and incorporating additional datasets for analysis. In response:

1. **Writing and Structure**: The manuscript has undergone significant revisions, including restructured sections, to improve clarity and readability.
2. **Figures**: All images have been updated to vector formats to preserve resolution and ensure high-quality visuals.
3. **Additional Data**: Despite our efforts to contact multiple observatories and colleagues, only the Canary Islands Volcanological Observatory provided a response. Consequently, we have included data from the 2021 La Palma eruption period.

We hope that these revisions, alongside the provided documentation of changes, meet the reviewers' expectations and adequately address their feedback.

Thank you for the opportunity to improve our work. We look forward to your advice.

Yours sincerely

Manuel Marcelino Titos Luzón
Postdoctoral Researcher, University of Granada, Spain
mmtitos@ugr.es.

---

## Referee Report (RR1)

In this revised version of the manuscript, the authors have clarified their work and the purpose of their research. In my opinion their work is worth publishing however I do think some points still need to be clarified / improved.

- Overall, the manuscript is well written but is sometimes very verbose (e.g. "to dive into these results" l.498, "Considering this information, we now proceed to discuss the results", ...). It sometimes makes the reading difficult, I would simplify the text to highlight the conclusions and observations.
- You still do not describe the features used to classify the signals. You do not have to describe them in details, especially if this is done in another study (otherwise put it in Supplementary). But you still need to describe them broadly in the manuscript.
- The methodology is now more clearly explained, but it still needs to be improved :
  - I understand the aim of the authors when they first present the overall method in Section 3.1 before explaining the application to the different cases, but it is hard to follow as we need some elements of section 3.2 to clearly understand section 3.1. Thus I suggest the following structure for the methodology section, that follows the overall structure of the article : 1) Description of the pre-trained systems (including all the technical details given at the beginning of section 4 and some insights on the accuracy/scores of the classifier on the MASTER Dataset), 2) Application of the pre-trained systems to event detection and classification, 3) Direct transfer learning, 4)Weakly supervised approach, 5)Outline of experiments on the POPO2002 and LAPALMA2021 datasets
  - It is still not clear to me what are the implications of the assumptions made l.210 and following. You assume that conditional distribution are the same l.213, but then acknowledge that they could be different (l.216). As said in my first review, I don't understand the logical link "Therefore" l.220. Are you suggesting that using weakly-supervised approaches allows to overcome the problem that conditional distributions are not the same? If so, why?
  - You must provide more details on how you carry out the direct transfer learning approach. I'm not an expert but I understand there are different approaches.
  - You must explain more clearly how classified events are compared the database events. From my understanding, the scores are computed on the labels associated to consecutive time windows of fixed length. If this is the case you must state it explicitly in the Methodology. You must also explain how you transform the datasets into labels associated to time windows.
- Although integrating the LAPALMA2021 dataset is interesting, I do not really see a clear link with the main subject of this paper, that is transfer learning. Indeed, you apply directly the Master dataset classifier to the dataset and explore how it allows you to detect events. So there is no added value on the "transfer learning" subject. In my view, to remain in the scope of the paper, you would need for example to compare the results of the Master dataset classifier, to results of the classifier re-trained on thePOPO2002 dataset (by direct transfer learning and/or weakly supervised transfer learning). That would show how data from different volcanoes can be combined to classify events on a new volcano.
- In my view the Results section must be expanded a little bit to highlight the main results. Instead of just stating that results are given in Table XX and Figure XX, comment them objectively (e.g. the best accuracy score are obtained with XX, the event with the highest

confusion rate is XX, ...). Then you can discuss and interpret these Results in the Discussion section.

- It is interesting to see the influence of the probability detection threshold on the Results, why not do it for the POPO2002 experiment as well? How would the confusion matrices of Tables 5 and 7 change with a different probability threshold? Besides, I don't think you mention the probability threshold you use to derive the Results presented in Section 4.

- You do not clearly explain why you test three different classifiers (RNN-LSTM, Dilated-LSTM and TCN). Is it to determine the best method? To study the variability of results depending on the classification methodology? Although interesting, this is beyond the main scope of this paper which deals with the pros and cons of direct / weakly supervised machine learning techniques. So you should investigate this point in a dedicated Discussion paragraph, rather than throughout the Results section. You can say in the methodology that you tested different methods and retain only one for the main results presentation, but investigate the influence of the classifier in the Disucssion. The same remarks stands for the size of the training dataset : In Section 4.1 you test 20% and 40%, but you do not carry out the same sensitivity analysis for the other applications. I would use the same percentage for all tests (e.g. 40%), and if you deem it important discuss the influence of the training test size in the discussion.

- Although he objective of the paper is not to point out that weakly supervised TL approaches can detect more events that direct TL approach or direct application of pre-trained classifiers, I would still expect a quantified comparison on this point. In this respect, I would include the results of the pre-trained classifier, and of the direct transfer learning approach. Besides, you do not clearly show that weakly supervised approaches allow to build less biased catalogues in comparison to other approaches. You do show that events that are not detected in the manually constructed catalogues are identified by weakly supervised classifiers, but you do not show clearly that direct transfer learning are less efficient in building less biased catalogues. In this perspective, the advantage of using weakly supervised approaches in comparison to direct transfer learning approaches is not clearly shown in your manuscript. For instance, how would direct transfer learning approaches for the seismic signal presented in Figure 4?

- You must improve the legends of all Figures. The reader must be able to understand their content without referring to the manuscript.

Specific remarks:

- To avoid misunderstandings, I would use "classifier" throughout the manuscript instead of "systems"

- I would mention the data used in the manuscript in the abstract, in its present form it is rather general and the reader does not know how the authors reached, in practice, their conclusions.

- Table 1 : Add the acronyms used for the events in the first column. Make it clear in the Table / the legend what classification/names you use in your work.

- L.34 : "such signal processing", what are you referring to?

- L.94 : You must expand and explain chat Transfer Learning consists in, with references and examples in the literature. Otherwise, a reader that is not familiar with this concepts will not understand what you mean by "re-train".

- L.96-99 ("The outcomes ... volcanic dynamics") and l.102 – 104 ("The outcomes ... dynamics") : This is a conclusion of your work, it should not be in the introduction.

- L.130 "over various time periods or at different volcanoes): I agree that you processing can minimize the difference in signals due to the sensor type, but you do not eliminate the variations associated to temporal evolution of the volcanic system, nor the variations associated to differences in volcanic processes or associated to different paths properties between the source and the sensor.
- L.139 : Define "pre-eruptive processes ». Do you mean everything that happens in between eruptions, or events that can be interpreted as eruption precursors?
- L.156 : Although I understand you may not have all the information on the sensors (but do check it, if you use mseed files you should have access to metadata), you must at least say how you got the data. Is it on a public repository? Is there a paper describing the acquisition and data? Where were the stations positioned on the volcano slopes? Same questions for the LAPALMA2021 database.
- L.247 : You do not explain how you choose the probability threshold.
- L.252 : You do not explain what the "desired result" is.
- L.257 : "some of these methods may not be as effective …" : Be more specific, give examples. Besides, this part should be in the introduction when you explain why (weakly supervised) transfer learning approaches are needed.
- L.231 : What difference do you make between "continuous" or "streaming"? besides you should make it clear at some point that all transfer learning approaches can't be used in real time.
- Figure 3 : Shouldn't the lines in C) sum to 1? I.e. a frame is necessarily classified as one of the 5 categories? If there's no detected event, then the window should be classified as noise.
- L.295 : "a subset", how do you construct it? What portion of the dataset does it represent?
- L.326 – 341: "All results … during training" : this should be in the Methodology section.
- L.339 - : "the model", which one?
- L.340 : What is "early stopping" ?
- Table 4 and 6 : As mentioned in my main remarks, it is not clear why you test 20% and 40% for the training dataset. Besides as you focus in the following on 5 categories only, I would keep the results of the 7 categories for the discussion (if relevant). Thus, I would only keep Table 5 and add a column for the accuracy of the direct transfer learning approach.
- Table 5 and 7 : Why did you not include the 7 categories of the POPO2002 dataset? Even if you can't predict all categories, it's interesting to see how they are classified. Otherwise, explain in the text why you do not display the 5 categories in the tables.
- Table 8 : As stated in the main comments, I would add the number of detected events for the original classifier and for the classifier obtained with the direct transfer learning approach. You should also add a line for the HYB events.
- L.419 "The vast majority", l.422 "many times" : You must quantify these statements. Besides, your remarks questions indeed the validity of the accuracy score computation. Couldn't you compute differently using events rather than time windows? E.g. for an event with start time t1 and end time t2, you label it with the label most represented in the successive tie windows. It would be a more robust accuracy estimation, eliminating "artifacts" associated to SNR or nested subevents, and prove that (i) you do detect rather correctly the events of the catalogue, and (ii) are able to refine the events duration and detect sub-events.
- L.475-476 ; "previously hidden information (…) can be obtained"

- L.537 : It is strange to have a paragraph "Summary of findings", and a paragraph "Conclusion". The conclusion is precisely about summarizing the findings.
- L.552 : If you mention the issue of membership threshold in the conclusion, I would expect you to investigate this issue not only for the construction of a new catalogue from scratch, but also for the weakly supervised methodology to train the classifier.
- L.572 : You do not investigate unsupervised learning techniques in your work, so your work does not "demonstrate" that these approaches are more successful.
- L.575 : I agree that using data from several catalogues could help develop "universal" monitoring tools, and you have the opportunity to investigate this in your work: use classifier trained on the master dataset, transferred with weakly supervised approaches to the Popo2 catalogue, and tested on the LAPALMA dataset. You could then compare the result with the ones obtained with the original classifier from the Master dataset, transferred directly to the POPO2 catalogue, and tested on the LAPALMA dataset. This would be very interesting.

Minor remark :

- Title : "catalogus" -> catalogues
- L.34 "frequency", it is not clear whether you speak of the signal frequency content or of the occurrence frequency.
- L.49-52 : there are too many references. you should develop on a few of them to explain their main results / methods.
- L.54 and following : why is this part in italic?
- L.83 : Deceptio -> Deception
- L.118 : "our hypothesis", what are your referring to?
- L.124 : How can you have 8 channels on a three-components seismic sensor? Chat are these channels?
- L.144 : "UMAP", give a reference, how did you compute it?
- L.212 : "domain information", what do you mean?
- L.213 : You do not explain what Ys and Yt are.
- L.230 : You have not yet explained what are RNN-LSTM, Dilated-RNN and TCN, you do it only l.264. As stated in my main comments, the Methodology section can be re-organize to avoid this kind of problem.
- L.279 : "three systems", I understand that you refer to RNN-LSTM, Dilated-RNN and TCN trained on the Master datasets, but when first reading the sentence it is not obvious.
- L.291 : "our initial hypothesis", at this point, the reader may not remember what your initial hypothesis is.
- L.295 : "Each" -> each
- L.496 : "because the" -> because of the
- L.512 – 516 "The first row … respectively". This should be in the legend, not in the main text.
- L.532 : There a missing number after "Figure"

---

## Referee Report (RR2)

**General Comments**

In this second iteration of the manuscript, the authors have clearly devoted significant effort to refining and restructuring their work. The result is a substantially improved document that showcases clearer objectives, methods, and outcomes. Across all sections, the organization and writing style have been noticeably enhanced, making the overall manuscript much more coherent and accessible.

**The Introduction** is particularly strong, providing both a concise background and a clear statement of the research motivation. In addition, **Section 2**, which focuses on seismic signals and data catalogs, has been reconstructed in a way that captures the essential details of catalog construction and usage. This section now offers a thorough explanation of how seismological data is collected, cataloged, and analyzed, setting a solid foundation for the subsequent methodological discussion.

The **Methodology** section has also undergone a marked improvement compared to the previous version. The authors' decision to outline each step more systematically—especially how the three experiments are structured—makes it much easier for readers to follow the logic and replicate the work. Notably, the emphasis on **pseudo-labeling** as part of their weakly supervised learning strategy deserves commendation. By using a pre-trained model as a pseudo-labeler and then re-training with the newly labeled data, they demonstrate an innovative approach to semi-supervised or weakly supervised classification in seismo-volcanic signals.

**Regarding the Discussion**, one of the central points the authors address, which is particularly interesting for the field, is the relatively low recognition rate compared to existing reference catalogs. They offer a plausible explanation that these catalogs, while established, may be incomplete or biased toward particular classes of events. Consequently, a strict comparison against them can underestimate the efficacy of the new system.

Along the same line, the authors highlight the **quality vs. quantity** dilemma. While the weakly supervised methodology might introduce some degree of noise or misclassification, it also increases the overall number of detected events, thus expanding the catalog. According to their description, it would be ideal for future users of this methodology to strike a balance by conducting manual checks on a fair portion of newly labeled events to verify their authenticity. These checks not only help mitigate the risk of accumulating errors from pseudo-labels but also lend credence to the claim that genuinely overlooked events are being discovered. Nevertheless, **we recommend** that the authors (and future users) **explore additional statistical consistency checks and cross-comparison with alternative detection methods** to further

strengthen the reliability of these expanded catalogs in subsequent research projects. By systematically verifying or filtering pseudo-labeled events—through model agreement, confidence thresholds, statistical checks, and domain-expert reviews—one can reduce the risk of error accumulation and improve the quality of the final training data.

From a **contextual usefulness** standpoint, the authors argue that any additional events—correctly identified or carefully verified—enrich our understanding of volcanic processes, potentially offering earlier or more nuanced insights into volcanic unrest. They stress that while it is important to measure success against established reference catalogs, it is equally crucial to recognize the value in uncovering smaller or subtler events that might have gone undetected. As a result, even if the system does not perfectly align with existing catalogs, it may enhance real-time monitoring, inform hazard assessments, and ultimately lead to more comprehensive research in volcano seismology.

Nonetheless, **further elaboration** on the potential pitfalls of pseudo-labeling, along with **additional quantitative** or **expert-driven validations**, would strengthen the overall argument.

Despite these minor weaknesses, this manuscript now provides a valuable contribution to the application of machine learning within volcano seismology. The authors' demonstration of how to construct and refine catalogs, leverage pre-trained models, and evaluate performance across multiple experiments will be extremely useful in guiding future research. Overall, the revision is a notable success, and the text should serve as a new reference for continued advances in the automated recognition and analysis of seismic-volcanic signals.

About very Minor writing issues:

- Introduction.

- line 52: A period "." is missing after "etc".
- line 54: Maybe lose instead of *loss*?
- line 54: an interesting topic: "*monitoring systems loss effectiveness when recognizing events over time..*" it would be ideal to include some references to support this point.
- line 83: *Deception* misspelled.
- line 108: there is an extra period ".".
- line 110: "volcano" misspelled.

- Seismic data and catalogs.

- line 123: as stated in fig.1., the data was also collected in 1996 and 2001-2002?
- lines 150 and 151: are we using "Popocatépetl" with or without an accent?

---

## Referee Report (RR3)

Detailed Review Report

**General Impressions**

The authors clearly propose that traditional machine learning models for seismic event detection often carry biases due to training on specific, limited catalogs. Their approach, utilizing pseudo-labeling based on pre-trained systems to enhance model generalization, is compelling and well-motivated. The manuscript presents a highly relevant and interesting investigation into improving seismic-volcanic catalogs through weakly supervised machine learning techniques. After multiple rounds of review, significant improvements have been made. However, readability and methodological clarity remain core concerns. Below, I outline detailed recommendations to address these issues constructively.

- **About Methodology Section.**

- **Section 3.1 (Methodology Clarity):**
  The manuscript currently states that the proposed method aligns with the open-set domain adaptation paradigm, explicitly designed to handle novel event categories. However, the authors subsequently note a significant limitation: the method only labels events within categories already present in the master database. This limitation appears to directly contradict the previously stated open-set capability. I recommend clarifying this contradiction explicitly. The authors should specify whether the approach is truly open-set (capable of detecting and handling unseen seismic categories) or acknowledge clearly that it is currently limited to closed-set scenarios.
  Although the assumptions clearly indicate that label spaces may only partially overlap, and thus novel categories could be present, the authors later explicitly state their methodology can only label categories that exist in the master database. Therefore, the authors should clarify how their approach practically handles (or does not handle) the novel categories mentioned in their assumptions. If the method currently doesn't handle these novel categories, explicitly stating this limitation and distinguishing clearly between the theoretical scenario and the actual method implementation would strengthen the manuscript.

- **Main issues in the Methodology (Experiment 3.2.1):**
1) Insufficient methodological details to reproduce the experiment
   o Problem: Currently, the authors only briefly mention:
     ▪ Three model architectures (**RNN-LSTM, Dilated-LSTM, TCN**),

- Pre-training on **MASTER-DEC**, then re-training with **POPO2002**

However, readers might ask:

- How were the models pre-trained initially (hyperparameters, training set sizes, epochs, loss functions, etc.)?
- What specific transfer learning strategies were applied (e.g., layer freezing, fine-tuning, learning rate adjustments)?
- How exactly were data split (train-validation-test)?
- What were the evaluation metrics or validation methods?
- Did authors address class imbalance or category distribution?

Without these details, readers cannot reproduce the experiments. It seems that some important information about this is in section 4 (results). We suggest to the authors that change the text to the methodology section.

2) Confusion caused by two alternatives, stating:
   - Problem: The authors propose two alternatives, stating:
     - **Option A**: Only use the **5 categories** in common with the MASTER-DEC catalog.
     - **Option B**: Adapt the model output to accommodate all **7 categories** present in POPO2002 by updating the output layer only.

But, critically:

- Authors state vaguely: "these two approaches have no major implications from a ML perspective".
- This statement is confusing because, practically, these two options have **very different implications**:

**Option A** completely excludes new categories, simplifying the task significantly. **Option B** involves at least minor model changes (output layer modification), and crucially, implies retraining with novel data categories (a clearly significant ML implication). This confusion significantly weakens methodological clarity.

- Second Experiment (3.2.2):

Clearly distinguish the novelty of Experiment 2 by explicitly stating upfront that the primary difference from Experiment 1 is the source of labels (pseudo-labels rather than true annotations). Emphasizing this difference early in the description would enhance readability.

- **About Results Section.**
  - **Lines 334-357:** This methodological detail is beneficial and should be moved explicitly into the methodology section for clarity and improved reproducibility.

- **Line 362:** The statement regarding "two experiments" conducted at this stage is confusing and should be simplified in the methodology section.

- Should the reader be benefced with more transfer learning details? (e.g., fine-tuning strategies, freezing layers explicitly, loss functions and training epochs?).

- **First Experiment Results:**

  While the overall self-consistency result (e.g., 77.38% accuracy for the RNN-LSTM model) provides a general sense of model performance, the confusion matrix reveals important class-specific differences—most notably, the relatively low recall for VT events (0.51) compared to much higher values for noise (0.97) and other event types. This suggests that the model may be biased toward the dominant class (likely noise), potentially inflating the global performance metric. I recommend that the authors include additional evaluation metrics, such as precision, recall, and F1-score for each class, as well as macro-averaged or balanced accuracy scores. These would provide a more nuanced understanding of how well the model generalizes across all event types, especially the underrepresented or more challenging classes like VT. Including this information would strengthen the assessment of the model's real-world applicability in diverse seismic scenarios.

- **Second Experiment Results:**

  While the weakly supervised fine-tuning improved global accuracy, the model's ability to detect meaningful seismic events—especially VT and LP types—remains limited, with VT nearly absent in the confusion matrix. The dominance of the noise class likely inflates the global metric. Additionally, the model's detection rate far exceeds the label count, which may reflect over-sensitivity rather than true discovery. More rigorous evaluation, including precision-recall analysis, event-level validation, or expert review of excess detections, would strengthen confidence in the weak supervision pipeline.

- **Third Experiment Results:**

  The use case of applying weakly supervised models during a pre-eruptive crisis is compelling and highlights the practical value of such approaches. However, the presentation of results—particularly the so-called "recognition results" table—is unclear. It is not evident whether the numbers reflect validated detections, raw counts, or comparisons to any ground truth. The sudden introduction of PhaseNet, while relevant, is also only partially integrated, with no evaluation metrics provided to contextualize its outputs or compare them to the proposed models. A more transparent and consistent presentation of results, including quantitative

comparisons, ground truth validation, and clearer labeling of what each table or number represents, would greatly improve the interpretability and impact of this section.

While the discussion highlights VT confusion rates exceeding 60% in some cases, this appears to reference only the worst-performing model (Dilated-LSTM). The other models achieve higher recall (e.g., 59% for TCN), and the average across all three models is closer to 47%, not 40%. A more balanced summary would acknowledge this range to accurately reflect performance variability across architectures.

**Summary about results:**

Throughout the results and discussion sections, the manuscript refers to "confusion matrices" and reports numerical values (e.g., 0.51, 0.31, 0.59 for VT events across models) without clearly stating whether these represent recall or confusion rates. However, the structure of the matrices—particularly the fact that each row sums to one—strongly suggests that the values correspond to **per-class recall**, i.e., the proportion of correctly classified instances for each true class. This is the standard interpretation for row-normalized confusion matrices in the machine learning literature. The ambiguity around this point makes the discussion difficult to follow and may contribute to the impression of poor presentation. For instance, the statement that "confusion rates exceed 60%" appears to refer to only the worst-performing model and does not align with the higher recall values seen in other models unless the reader assumes a confusion rate = 1 - recall. For the sake of clarity and consistency, it is essential that the manuscript explicitly define how these matrices are computed and what the reported values represent. This will not only improve readability but also help readers interpret the results accurately.

While the qualitative example shown in Figure 4 is compelling and suggests the model is capable of discovering events missed during the initial labeling, these anecdotal demonstrations are not sufficient to validate the effectiveness of the weakly supervised system. To move beyond suggestive visuals and convincingly argue for the scientific value of these new detections, the study would benefit from a more rigorous validation approach—such as expert review, waveform similarity analysis, or cross-comparison with independent models like PhaseNet. Without such steps, the claim that these new detections are not false positives remains speculative and limits the broader impact of the proposed method.

- **About the Discussion Section:**
  - **Line 416:** Verify if percentages presented in the discussion exactly match the results section; discrepancies would confuse readers.

While the qualitative example shown in Figure 4 is compelling and suggests the model is capable of discovering events missed during the initial labeling, these anecdotal demonstrations are not sufficient to validate the effectiveness of the weakly supervised system. To move beyond suggestive visuals and convincingly argue for the scientific value of these new detections, the

study would benefit from a more rigorous validation approach—such as: expert review, waveform similarity analysis, or cross-comparison with independent models like PhaseNet (we'll talk about this later). Without such steps, the claim that these new detections are not false positives remains speculative and limits the broader impact of the proposed method.

The discussion attributes the weak performance of the model on volcano-tectonic events (VTEs) to discrepancies in labeling criteria, subjective annotation boundaries, and prototype mismatches. While labeling inconsistency is a known challenge in volcano seismology, VTEs are typically among the most well-defined and reliably detectable seismic signals due to their impulsive, high-frequency nature. Numerous existing models (e.g., PhaseNet) have shown robust detection of such events across different volcanoes. The fact that the system recovers only 5% of annotated VTEs suggests that the problem may lie more in the modeling strategy or prototype selection than in catalog inconsistency alone. A more balanced discussion should consider whether the weak supervision framework fails to generalize to realistic variability within VTEs and whether model or prototype refinement could improve performance.

The comparison with PhaseNet in the third experiment raises concerns regarding methodology. The authors assess PhaseNet's performance by comparing the number of detected phases across different score thresholds, arguing that only detections above 0.8 correspond well with the labeled dataset. However, this approach overlooks the fact that many valid seismic picks—especially low-amplitude or emergent phases—often have lower phase scores (e.g., 0.3–0.6), yet still align with cataloged arrivals. Furthermore, raw pick counts do not constitute a meaningful evaluation metric unless aligned with ground truth picks using a timing tolerance. To make a valid comparison, the authors should report precision, recall, and pick timing accuracy against the labeled dataset across multiple thresholds. Without this, the argument that PhaseNet underperforms is not well supported and may misrepresent the model's actual capabilities.

- **About Summary of Findings Section:**

  - Figure 10: Clearly label differences between rows 3 and 4.
  - Ensure consistent PSD plotting style across Figures 10, 11, and 12 for clarity.

**Other (Very) Minor Remarks:**

- **Abstract:**
  - **Lines 2 & 4:** avoid unnecessary repetition of word "however" within the same paragraph; consider synonyms or rephrasing to improve readability.

- Introduction:
  - **Line 37:** a space is missing, "..crises.However"
  - **Line 51:** there is an extra space; "Canario et al., 2020 ;"
  - **Lines 57-63:** suggestion: another challenge is that upgrades and updates to seismic instrumentation over decades complicate the review of historical seismicity, as the digital signals may not share a consistent framework.
  - **Lines 56, 66, 71, 95, etc.:** There are inconsistencies in citation formatting throughout the manuscript. Please ensure that references within parentheses follow the standard format, e.g., "(Weiss et al., 2016)2, rather than "(Weiss et al. (2016))".
  - **Lines 168:** space missing at "MASTER-DEC(1-50HZ)"

- Methodology and experimental framework:
  - **Line 247:** repetitive vocabulary again (stream).

- Discussion.
  - **Lines 445-446:** review grammar of "According to such table, on average, only 5% of the analysis windows labeled as VTE in the original catalog were recognized by the retrained systems."
  - **Line 473:** please check the text "(Fig. 7)a."

---

## Referee Report (RR4)

NHESS reviewer report 4, 2025-07.

The authors have addressed the major concerns raised in previous review rounds, providing clarifications, methodological adjustments, and additional analysis that substantially improve the clarity and scientific value of the manuscript. The experiments are now better contextualized, the performance metrics have been calculated as requested, and the discussion reflects a more balanced interpretation of the model's strengths and limitations. With these revisions, I believe the manuscript is suitable for publication and will be a valuable contribution to the field of machine learning in seismology.

As a final recommendation, I suggest that the authors consider including the precision, recall, and F1-score metrics—either in the appendix or as supplementary material—to enhance transparency and allow for easier comparison with related studies.

---

## Author Response (AR2)

**March 12, 2025**

**Editor-in-Chief**

Natural Hazards and Earth System Sciences

Dear Editor,

I am pleased to submit this cover letter regarding the original research article (nhess-2024-102) entitled "Could seismo-volcanic catalogues be improved or created using weakly supervised approaches with pre-trained systems?" by Titos M., et al., for consideration in NHESS. We have carefully reviewed the feedback from **all four reviewers** and greatly appreciate the time and effort they have invested in evaluating our work.

We hope that these revisions, alongside the provided documentation of changes, meet the reviewers' expectations and adequately address their feedback.

Thank you for the opportunity to improve our work. We look forward to your advice.

Yours sincerely

Manuel Marcelino Titos Luzón
Postdoctoral Researcher, University of Granada, Spain
mmtitos@ugr.es.

**ANSWER TO THE REVIEWER'S COMMENTS**

In the following, we have provided detailed answers to the comments of the reviewers. The original texts from the reviewers are in normal font. Our answers are in bold font. We would like to take this opportunity to thank the reviewers for their valuable comments and for their time and resources.

**Answer to comments of Reviewer#1**

**We would like to thank Dr. Gordon Woo for the careful reading of this manuscript and the thoughtful comments that have improved the quality of this manuscript. Furthermore, below are those comments that need more clarification.**

The limitations of the paper should be more clearly presented.

**Answer to comments of Reviewer#2**

**Dear reviewer#2, We are very thankful for your thoughtful suggestions. Below, we present how we have addressed them.**

In this revised version of the manuscript, the authors have clarified their work and the purpose of their research. In my opinion their work is worth publishing however I do think some points still need to be clarified / improved.

- Overall, the manuscript is well written but is sometimes very verbose (e.g. "to dive into these results" l.498, "Considering this information, we now proceed to discuss the results", …). It sometimes makes the reading difficult, I would simplify the text to highlight the conclusions and observations.

**In the new version of the manuscript, We have made some sentences simpler to enhance readability.**

- You still do not describe the features used to classify the signals. You do not have to describe them in details, especially if this is done in another study (otherwise put it in Supplementary). But you still need to describe them broadly in the manuscript.

**The objective of this work is the application of the weakly supervised approach to create reliable seismic catalogs with less human effort. This approach can be used both with parameterized signals and with the raw waveform itself (if a sufficiently large dataset is available). Therefore, we understand that the description of the parameterization paradigm is not a goal of this work. This is why, in the experiments section, we briefly include a small description of this parameterization and reference. Our prior work provides a detailed description of the applied raw signal parameterization procedure.**

**In the actual version of the manuscript, we had introduced this paragraph: 'The data stream illustrates continuous or streaming analysis (allowing near real-time processing). To carry out the recognition step using the network seed (trained with the MASTER-DEC dataset), streaming or continuous signals are filtered between 1 and 20 Hz and split into frames or windows; the same feature extraction algorithm used in MASTER-DEC is applied. For each window, a feature engineering pipeline based on a logarithmic frequency scale filter bank is applied. This pipeline reduces the dimensionality of the input vector associated with each analysis window (compared to raw signals), which facilitates the training and convergence of the systems, as it increases the separability of the data based on well-studied features in the literature (see Titos et al., 2024 for a detailed understanding of the parameterization pipeline).' '**

- The methodology is now more clearly explained, but it still needs to be improved :

      o I understand the aim of the authors when they first present the overall method in Section 3.1 before explaining the application to the different cases, but it is hard to follow as we need some elements of section 3.2 to clearly understand section 3.1. Thus I suggest the following structure for the methodology section, that follows the overall structure of the article : 1) Description of the pre-trained systems (including all the technical details given at the beginning of section 4 and some insights on the accuracy/scores of the classifier on the MASTER Dataset), 2) Application of the pre-trained systems to event detection and classification, 3) Direct transfer learning, 4)Weakly supervised approach, 5)Outline of experiments on the POPO2002 and LAPALMA2021 datasets.

**Regarding the structure of the article, we have followed the suggestions of all the reviewers from the previous version. After reviewing your recommendations and those of the others, we found that the simplest and clearest structure is the one we have outlined in the manuscript.**

**First, we introduce the proposed methodology. Second we describe the experiments: 1. a classical knowledge transfer experiment; 2. an experiment with a weakly supervised approach where we use a seismic catalog to compare results with our approach; 3. and finally, we conducted an experiment with a set of seismic signals for which no catalog is available. Through these experiments, we demonstrate the capability of the proposed model to automatically improve existing catalogs or build them from scratch.**

**We believe this structure most effectively aligns with the previous suggestions and enhances the clarity of the methods and results presented.**

o It is still not clear to me what are the implications of the assumptions made l.210 and following. You assume that conditional distribution are the same l.213, but then acknowledge that they could be different (l.216). As said in my first review, I don't understand the logical link "Therefore" l.220. Are you suggesting that using weakly-supervised approaches allows to overcome the problem that conditional distributions are not the same? If so, why?

**The wording of our hypothesis in the manuscript has been rewritten to improve its clarity.**

**The conditional distributions may differ between the source and target domains, which is a common challenge in domain adaptation tasks. Weakly-supervised approaches, such as pseudo-labelling, do not completely overcome this problem, but they provide a practical way to mitigate its effects under certain assumptions:**

1. **Leveraging High-Confidence Predictions:**
   **Weakly-supervised methods rely on the model trained on the source domain ($D_S$) to generate probabilistic predictions for the target domain ($D_T$). By selecting only those instances in $D_T$ with high per-class probability (i.e., high confidence), we assume that these predictions are more likely to be correct. This approach implicitly assumes that, for high-confidence predictions, the conditional distributions $P(Y|X_s)$ and $P(Y|X_t)$ are approximately similar, at least for the shared classes between domains.**

2. **Reducing the Impact of Distribution Mismatch:**
   **While the conditional distributions may differ globally, weakly-supervised methods focus on the subset of target data where the model's predictions are most reliable. This subset is likely to have a smaller discrepancy between $P(Y|X_s)$ and $P(Y|X_t)$, as the model's confidence reflects a degree of similarity in the feature-label relationships. By iteratively refining the pseudo-labels and retraining the model, we can gradually adapt the model to the target domain's conditional distribution.**

3. **Handling Shared and Novel Classes:**
   **In the context of open set domain adaptation, where the target domain may contain classes not present in the source domain, weakly-supervised methods help identify and separate shared classes from novel ones. High-confidence pseudo-labels are typically assigned to shared classes, while low-confidence predictions may indicate novel classes or domain-specific variations. This selective approach reduces the risk of negative transfer caused by mismatched conditional distributions.**

4. **Justification:**
   **While weakly-supervised methods do not guarantee that $P(Y|X_s)=$**
   **$P(Y|X_t)$ , they provide a computationally efficient and scalable way to**
   **adapt models to new domains when labelled target data is scarce. The**
   **key assumption is that the model's high-confidence predictions in the**
   **target domain are sufficiently accurate to bootstrap the adaptation**
   **process, even if the conditional distributions are not identical.**

   **In summary, weakly-supervised approaches do not entirely overcome**
   **the problem of differing conditional distributions, but they offer a practical**
   **framework to mitigate its effects by focusing on high-confidence predictions**
   **and iteratively refining the model's understanding of the target domain. This**
   **makes them a valuable tool in scenarios where obtaining labelled target data is**
   **expensive or impractical.**

   o You must provide more details on how you carry out the direct transfer
   learning approach. I'm not an expert but I understand there are different approaches.

   **Section 3.1 Methodology has been rewritten with the intention of including**
   **those aspects that help clarify the transfer learning and domain adaptation**
   **approach followed in this work. Above in this letter, we have also described in**
   **detail how we carried out the knowledge transfer.**

   o You must explain more clearly how classified events are compared the
   database events. From my understanding, the scores are computed on the labels
   associated to consecutive time windows of fixed length. If this is the case you must
   state it explicitly in the Methodology. You must also explain how you transform the
   datasets into labels associated to time windows.

   **In Section 4, where we describe the results of the experiments, we have added**
   **this paragraph indicating how we map the information from the seismic**
   **catalog to labels: To perform a robust analysis of system performance based**
   **on the accuracy metric (%) and build confusion matrices, it is necessary to**
   **transform the information contained in the catalog into labels from which the**
   **study can be conducted. Since in experiments 1 and 2 we start with a seismic**
   **catalog that contains annotations for the start and end of each event present**
   **in each seismic signal, once the signals are preprocessed and windowed, we**
   **can associate a label with each window. In this way, each window can be**
   **analyzed based on its classification according to its label.**

   - Although integrating the LAPALMA2021 dataset is interesting, I do not really see a
   clear link with the main subject of this paper, that is transfer learning. Indeed, you
   apply directly the Master dataset classifier to the dataset and explore how it allows
   you to detect events. So there is no added value on the "transfer learning" subject. In
   my view, to remain in the scope of the paper, you would need for example to

compare the results of the Master dataset classifier, to results of the classifier re-trained on thePOPO2002 dataset (by direct transfer learning and/or weakly supervised transfer learning). That would show how data from different volcanoes can be combined to classify events on a new volcano.

**As we have argued previously, the experiments and results presented in this article address the suggestions of the different reviewers. To this end, three distinct experiments have been conducted. In the first experiment, classical transfer learning is carried out, where a model trained with Deception Island data is retrained with data from Popocatépetl. The goal of this experiment is to assess how well the system can adapt to the labeled data in the catalog and achieve highly effective results, with a performance of around 90%.**

**In the second experiment, the goal is to introduce our weakly supervised learning methodology and demonstrate how the catalog obtained using this methodology greatly differs from the preliminary catalog available for Popocatépetl. To do this, similarity results are shown, where it is observed that only 50% of the events initially annotated in the catalog are recognized. Meanwhile, many other events that are now recognized were never considered previously.**

**Finally, in the third experiment, included in the first round of revisions, the aim is to demonstrate how effective our methodology is for building catalogs from scratch, where no prior information exists. For this, we use data from the 2021 La Palma eruption and compare our approach with a widely used tool like Phasenet, which is also based on AI.**

**We believe that these three experiments cover the full spectrum of the use of the proposal introduced here, with different use cases. And we do this in response to the demands of previous reviewers.**

- In my view the Results section must be expanded a little bit to highlight the main results. Instead of just stating that results are given in Table XX and Figure XX, comment the objectively (e.g. the best accuracy score are obtained with XX, the event with the highest confusion rate is XX, …). Then you can discuss and interpret these Results in the Discussion section.

**Given the difficulty raised by several reviewers in following the workflow and following the template of some articles published in this journal, we believe that to facilitate this, it is necessary to include a separate section for describing the results and another for discussing them. Therefore, in Section 4 (Results), we simply describe the obtained results and their meaning. In Section 5 (Discussion), we analyze these results in detail for each experiment.**

- It is interesting to see the influence of the probability detection threshold on the Results, why not do it for the POPO2002 experiment as well? How would the confusion matrices of Tables 5 and 7 change with a different probability threshold? Besides, I don't think you mention the probability threshold you use to derive the Results presented in Section 4.

**As the reviewer points out, the detection probability threshold is a crucial parameter in the weakly supervised algorithm proposed here, as it controls the system's sensitivity. A very high threshold will only allow the inclusion of events highly similar to those learned in the source domain. A very low threshold will include more diverse events, ultimately enabling domain adaptation.**

**However, in the context of the classical transfer learning experiment, specifically regarding Table 5, the results remain unchanged because the probability threshold does not exist. In this case, events are classified by assigning the seismic category with the highest probability in the output layer, meaning they are always classified into the most probable category.**

**Finally, we have added to the experiment description using the weakly supervised approach that the selected detection probability threshold was 50%, aiming to include as many events as possible, even if they were less rigorous.**

- You do not clearly explain why you test three different classifiers (RNN-LSTM, Dilated- LSTM and TCN). Is it to determine the best method? To study the variability of results depending on the classification methodology? Although interesting, this is beyond the main scope of this paper which deals with the pros and cons of direct / weakly supervised machine learning techniques. So you should investigate this point in a  dedicated Discussion paragraph, rather than throughout the Results section. You can say in the methodology that you tested different methods and retain only one for the main results presentation, but investigate the influence of the classifier in the Disucssion. The same remarks stands for the size of the training dataset : In Section 4.1 you test 20% and 40%, but you do not carry out the same sensitivity analysis for the other applications. I would use the same percentage for all tests (e.g. 40%), and if you deem it important discuss the influence of the training test size in the discussion.

**The reason for including these three methodologies and not others in the paper was primarily to test the robustness of the method. We agree with the reviewer that any other methodology capable of analyzing temporal signals could have been used, from Hidden Markov Models to Transformers. However, since this study builds upon previous work using pre-trained and already published systems, we chose these three so that readers can easily find extensive information about these systems and their characteristics, facilitating and streamlining the reading of this paper. Otherwise, we would**

**have had to describe both the proposed models and their training before addressing classical transfer learning and the weakly supervised approach.**

**Regarding the percentage of the dataset used for training, we would like to clarify that we included it as an illustrative example to show that when performing classical transfer learning between related domains, it is not necessary to use a very large training dataset to achieve good results. This allows most of the data to be used for testing while still obtaining a high performance, close to 89%.**

**In the case of the weakly supervised approach, the size of the training dataset depends on the complexity of the signals, and it is up to the user to determine the appropriate size. In this study, we decided to set it at 40% to better analyze the number of detected events, even in scenarios where the training set is relatively small.**

- Although the objective of the paper is not to point out that weakly supervised TL approaches can detect more events that direct TL approach or direct application of pre- trained classifiers, I would still expect a quantified comparison on this point. In this respect, I would include the results of the pre-trained classifier, and of the direct transfer learning approach. Besides, you do not clearly show that weakly supervised approaches allow to build less biased catalogues in comparison to other approaches. You do show that events that are not detected in the manually constructed catalogues are identified by weakly supervised classifiers, but you do not show clearly that direct transfer learning are less efficient in building less biased catalogues. In this perspective, the advantage of using weakly supervised approaches in comparison to direct transfer learning approaches is not clearly shown in your manuscript. For instance, how would direct transfer learning approaches for the seismic signal presented in Figure 4?

**The results of the pre-trained classifier and the direct transfer learning approach are included in the manuscript. Once again, we would like to clarify that classical transfer learning uses a pre-trained model as a starting point to train a new system with data from a new seismic catalog, in our case, using labels under a supervised learning paradigm. The results are presented in Tables 4 and 5 in Section 4 and discussed in Section 5.**

**Regarding the results of the pre-trained classifier, these can be found in Table 6. This table consists of two result columns: blind test and weakly supervised. The blind test column corresponds to the results obtained by the pre-trained system when compared with the POPO2002 catalog without re-training. The weakly supervised column presents the results obtained after re-training with the data included in the new dataset, compared to the annotations in the same POPO2002 catalog.**

**As seen from the results in both tables, along with Table 8, classical transfer learning techniques before re-training are responsible for creating the training dataset for domain adaptation and, as such, contribute to the creation of less biased catalogs. Therefore, the weakly supervised algorithm simply uses the events recognized and labeled by the pre-trained system as labels and training events, adjusting the system to the characteristics of the new events. Applying this use case to the example in Figure 4, the pre-trained system will recognize the inserted LP events and include them in the new database if they meet the detection probability threshold criterion. In this way, once retrained, the model will be able to detect these types of events if they are present in the traces.**

**As previously mentioned, the results in Table 8 show that pre-trained models detect many events that are not annotated in the catalog, since the weakly supervised approach originates from these systems.**

- You must improve the legends of all Figures. The reader must be able to understand their content without referring to the manuscript.

**All the legends have been improved for the sake of clarity**.

Specific remarks:

- To avoid misunderstandings, I would use "classifier" throughout the manuscript instead of "systems"

**The proposed system is not simply an implementation of a machine learning-based classification algorithm. These systems are built around events that are precisely delineated in time, commonly known in the literature as isolated event classification systems. The final output of our system is one that, given the signal's waveform, detects an event and assigns the appropriate label to a specific class. In other words, the system performs both event detection and classification tasks.**

- I would mention the data used in the manuscript in the abstract, in its present form it is rather general and the reader does not know how the authors reached, in practice, their conclusions.

**The abstract has been modified including the general idea behind the work and the dataset used: 'When a system trained on a master dataset and catalog from Deception Island Volcano (Antarctica) is used as a pseudo-labeller in other volcanic contexts, such as Popocatépetl (Mexico) and Tajogaite (Canary Islands) volcanoes, within the framework of weakly supervised learning, it can uncover and update valuable information related to volcanic dynamics'**

- Table 1 : Add the acronyms used for the events in the first column. Make it clear in the Table / the legend what classification/names you use in your work.

**The acronyms have been added to the table.**

- L.34 : "such signal processing", what are you referring to?

**Signal processing in this context refers to the spatio-temporal analysis of the seismic signals for comprehending the underlying physics behind the eruptions, and thus understanding why they occur. In the previous sentence, we discussed signal processing to analyze volcanic dynamics, which is why we refer to this signal processing.**

- L.94 : You must expand and explain chat Transfer Learning consists in, with references and examples in the literature. Otherwise, a reader that is not familiar with this concepts will not understand what you mean by "re-train".

**In the introduction, we have added a reference to one of the most widely cited works on the transfer learning paradigm. In the methodology section, the previous version already included a brief description of this concept. We have cited the same work again to provide clearer guidance for the reader.**

- L.96-99 ("The outcomes … volcanic dynamics") and I.102 – 104 ("The outcomes … dynamics") : This is a conclusion of your work, it should not be in the introduction.

**These sentences appear in the introduction as they provide a general overview of our methods and findings from the experiments. If the reviewer thinks they should be removed, we are open to doing so. However, we feel these sentences help the reader better understand the context of the paper.**

- L.130 "over various time periods or at different volcanoes): I agree that you processing can minimize the difference in signals due to the sensor type, but you do not eliminate the variations associated to temporal evolution of the volcanic system, nor the variations associated to differences in volcanic processes or associated to different paths properties between the source and the sensor.

**The sentence has been revised to incorporate the reviewer's suggestion.: 'This filtering minimizes the influence of the sensorization used for signal recording and ensuring the comparability of the data recorded by different sensors over various time periods or at different volcanoes (it does not fully eliminate variations related to the temporal evolution of the volcanic system, nor those stemming from differences in volcanic processes or path properties between the source and the sensor)'**

- L.139 : Define "pre-eruptive processes ». Do you mean everything that happens in between eruptions, or events that can be interpreted as eruption precursors?

**With pre-eruptive processes we refer to a set of phenomena occurring within a volcanic system before an eruption. We added this explicative sentence in the**

**manuscript: 'set of geological, geophysical, and geochemical phenomena occurring within a volcanic system before an eruption.'**

- L.156 : Although I understand you may not have all the information on the sensors (but do check it, if you use mseed files you should have access to metadata), you must at least say how you got the data. Is it on a public repository? Is there a paper describing the acquisition and data? Where were the stations positioned on the volcano slopes?

**In Section 2: Seismic Data and Catalogs, we provide a detailed explanation of all information related to the sensors and databases available. Most data files are in binary format, containing only waveform information. The data were provided by three different observatories, as noted in the acknowledgments. Therefore, in the Data Availability section, we recommend contacting the corresponding author.**

Same questions for the LAPALMA2021 database.

- L.247 : You do not explain how you choose the probability threshold.

**The following sentence has been included for the sake of the clarity: 'The system's sensitivity is directly influenced by the chosen threshold: a lower value increases sensitivity, allowing more events to be included but potentially reducing specificity. Conversely, a higher threshold enhances specificity by selecting only the most confident detections, though at the risk of lowering sensitivity The threshold value will be determined by the user based on their needs when addressing the problem. In our case, we have set it at 60%, allowing the inclusion of a greater number of events and better adaptation to the new domain.'**

- L.252 : You do not explain what the "desired result" is.

**We thank the reviewer for this observation. This was a drafting error. The sentence has been corrected to: "Repeat steps 2 to 4 iteratively until the results converge and no further improvements are observed in the catalog creation, or until the user deems it appropriate."**

- L.257 : "some of these methods may not be as effective …" : Be more specific, give examples. Besides, this part should be in the introduction when you explain why (weakly supervised) transfer learning approaches are needed.

**This sentence aims to highlight that some of the most widely used techniques in the recognition of continuous seismo-volcanic signals, both offline and in real-time, where a signal may contain multiple seismic signals and the goal is to detect and classify all of them, are not as effective as they should be. Since this is the experimental framework section, this paragraph aims to explain to the reader that, given the nature of the signals and the goal of the problem, many of the classification systems used are not suitable. It also introduces or**

**justifies the use of the systems proposed in this study (LSTM, Dilated-LSTM, and TCN). We believe that including this in the introduction could be confusing for the reader, as the introduction only addresses the problem of catalog construction, and the type of architecture used to carry out this task is secondary. As we have already mentioned in this response letter, we used these three architectures as a baseline because we started with systems trained on MASTERDEC, and there are publications that support their results. We believe this paragraph is well-placed in the experimental framework section because it motivates the reader to understand the choice and use of these architectures.**

- L.231 : What difference do you make between "continuous" or "streaming"? besides you should make it clear at some point that all transfer learning approaches can't be used in real time.

**The main difference between continuous and streaming lies in the type of signal analysis. Continuous analysis involves the examination of signals with the goal of detecting and classifying different types of events. Therefore, streaming refers to the real-time or near real-time analysis of continuous signals, where data is processed as it is received, meanwhile continuous (offline) analysis involves the examination of pre-recorded signals, where the data is analyzed retrospectively.**

**The sentence has been modified: 'Some of these methods may not be as effective for the specific challenges posed by continuous or near real-time data processing'**

- Figure 3 : Shouldn't the lines in C) sum to 1? I.e. a frame is necessarily classified as one of the 5 categories? If there's no detected event, then the window should be classified as noise.

**Since these are classifiers with a softmax layer in the output layer, the sum of each output unit corresponds to the probability of the input belonging to each seismic category. Therefore, each input will always be classified into one seismic category, and the number of seismic categories will depend on the catalog from which the classifier was built. In our case, the seismic categories are 5 (Noise, TRE, HYB, LPE, and VTE). In this regard, any input window will always be classified with one of the 5 labels. Once the signal has been analyzed and all the obtained labels are generated, a post-processing step is applied, and consecutive windows with the same label are grouped together, which we interpret as part of the same event (see Titos et al. 2018). If labels of very short duration (e.g., a single frame) of different seismic categories appear consecutively, that part of the signal is detected and classified as an 'unknown event', as there is no pattern indicating its association with a specific event. On the other hand, if multiple consecutive labels from the same event are obtained, their average probability of belonging is analyzed, and if it exceeds**

the established probability threshold, they are added to the new training dataset.

- L.295 : "a subset", how do you construct it? What portion of the dataset does it represent?

**As described in the manuscript, the aim of this work is to construct a robust seismic catalog with minimal human effort. To achieve this, we start with a system that we consider a "master" system, as it was built from a database that we also consider "master." As outlined in the methodology and Figure 3, the idea is to analyze a subset of data from a new volcano to include some of the detected events from that new subset into a new database, and train the new system with this new data to build a classifier adapted to the new volcanic environment. When we refer to the subset, we are specifically talking about that subset of data from the new volcanic environment that will be used to create the database that drives the domain adaptation of the classifier between volcanoes. Therefore, if this subset is too large, the remaining data to test the robustness of the new catalog will be limited. This is why, as shown in Section 4.2, which describes the obtained results, we used 40% of the POPO2002 data, reserving 60% to test the robustness of the method.**

- L.326 – 341: "All results … during training" : this should be in the Methodology section.

**Since we are discussing a characteristic specific to the training and setting up of the systems, we believe this feature should be placed in the experimentation section.**

- L.339 - : "the model", which one?

**Corrected. It was a drafting error.**

- L.340 : What is "early stopping" ?

**As the text indicates, early stopping is a widely used regularization technique in the deep learning field to prevent overfitting of the systems.**

- Table 4 and 6 : As mentioned in my main remarks, it is not clear why you test 20% and 40% for the training dataset. Besides as you focus in the following on 5 categories only, I would keep the results of the 7 categories for the discussion (if relevant). Thus, I would only keep Table 5 and add a column for the accuracy of the direct transfer learning approach.

**Table 4 refers to the results obtained from the POPO2002 dataset when applying a classical transfer learning approach. The inclusion of 20% and 40% of the database in the training set is simply to show the reader that this approach is capable of learning the information contained in the catalog and achieving highly effective results, close to 90%. Including more training data could slightly improve recognition results, but our goal is not this. Instead, we**

aim to demonstrate that the systems are learning the information contained in the catalog without disregarding valuable information, meaning they are learning in a biased way.

The inclusion of 5 and 7 categories has the same objective: to inform the reader that when training a system with a predefined catalog, the results are very good regardless of the number of categories included in the training. However, since the objective of the work is to apply a weakly supervised approach, using a pre-trained system on a master database with 5 seismic categories, we are limited to working with and extracting results from only 5 categories. It is impossible to extract results with 7 categories as our system only recognizes 5.

- Table 5 and 7 : Why did you not include the 7 categories of the POPO2002 dataset? Even if you can't predict all categories, it's interesting to see how they are classified. Otherwise, explain in the text why you do not display the 5 categories in the tables.

The objective of the work is to apply a weakly supervised approach, using a pre-trained system on a master database with 5 seismic categories, we are limited to working with and extracting results from only 5 categories. It is impossible to extract results with 7 categories as our system only recognizes 5.

We have tried to make this clear in the text, line 387: 'As previously stated, since MASTER-DEC consists of five seismic categories and the weakly supervised approach builds on pre-trained models, the results presented here include only these 5 seismic categories.'

- Table 8 : As stated in the main comments, I would add the number of detected events for the original classifier and for the classifier obtained with the direct transfer learning approach. You should also add a line for the HYB events.

The results in Table 8 are the same as those in Table 6; however, one shows the number of original events and the number of recognized events, while the other shows the percentage of matching events between the approaches and the original catalog. Regarding the hybrid events, they are not included in Table 8 because none are detected, and in the original catalog, as shown in Table 3, there is only one.

- L.419 "The vast majority", l.422 "many times" : You must quantify these statements.

Besides, your remarks questions indeed the validity of the accuracy score computation. Couldn't you compute differently using events rather than time windows? E.g. for an event with start time t1 and end time t2, you label it with the label most represented in the successive tie windows. It would be a more robust accuracy estimation, eliminating "artifacts" associated to SNR or nested subevents, and prove that (i) you do detect rather correctly the events of the catalogue, and (ii) are able to refine the events duration and detect sub-events.

**When we refer to the "vast majority," it is difficult to quantify the exact number, as what we are describing is that, in the original catalog, what was initially known as "garbage" or "tremor" in our system is associated with valid event labels. Analyzing the information in Table 7, the confusion ratio of each seismic category can be observed. Regarding the validation of the system performance, applying the approach suggested by the reviewer is complicated, as the start and end labels of each event depend on the subjectivity of the human operator who created the catalog. Therefore, to compare at the event level, we would need to define when a detected event is considered correct, even if the start or end does not exactly match the annotations in the catalog. This is why we opted for window-based recognition, which ultimately indicates what percentage of the events are being recognized. Additionally, once the recognition at the window level is obtained, a grammar is applied, from which the number of recognized events is derived, as noted in Table 8.**

- L.475-476 ; "previously hidden information (…) can be obtained"

**Hidden information has been changed to unannotated information for the sake of clarity.**

- L.537 : It is strange to have a paragraph "Summary of findings", and a paragraph

"Conclusion". The conclusion is precisely about summarizing the findings.

**The summary of findings and conclusions, although similar, address different aspects of the previous review processes. In the summary of findings, the general results and conclusions of the experiments are described. This section was suggested by a reviewer to clarify the experimental framework of the work, which seems to be quite confusing for readers not familiar with these techniques. On the other hand, the conclusions address the overall and final points of the work.**

- L.552 : If you mention the issue of membership threshold in the conclusion, I would expect you to investigate this issue not only for the construction of a new catalogue from scratch, but also for the weakly supervised methodology to train the classifier.

**This is an interesting suggestion. The choice of threshold is a very important parameter, as it determines which future events will be included or excluded**

from the new training database, from which the adaptation to the new domain will be carried out. In this study, we have decided to set a very low threshold, around 60%, to include as many events as possible in the new database. Although studying the effect of the threshold would be interesting, we believe it is highly dependent on the specific objectives of the observatory or the problem being addressed. That is why we have only considered this analysis with LAPALMA2021, because Including this analysis for POPO2002 would significantly extend the work.

- L.572 : You do not investigate unsupervised learning techniques in your work, so your work does not "demonstrate" that these approaches are more successful.

We agree with this comment. In this study, we did not evaluate unsupervised learning techniques. The reason for not including them is that our focus was on studying semi-supervised learning techniques. However, we are currently working on unsupervised approaches based on constructive learning to analyze their capabilities. It is also important to highlight that using unsupervised learning techniques inherently requires a posteriori analysis by experts to interpret the clusters identified by the system. Since our goal is to minimize human review efforts, we believe that utilizing a master database could assist in constructing less biased catalogs.

- L.575 : I agree that using data from several catalogues could help develop "universal" monitoring tools, and you have the opportunity to investigate this in your work: use classifier trained on the master dataset, transferred with weakly supervised approaches to the Popo2 catalogue, and tested on the LAPALMA dataset. You could then compare the result with the ones obtained with the original classifier from the Master dataset, transferred directly to the POPO2 catalogue, and tested on the LAPALMA dataset. This would be very interesting.

We agree with the reviewer that this experiment is very interesting. In fact, the authors of this study previously evaluated it. However, including this experiment in the manuscript presents a challenge. As we have argued in the text, a complete catalog of the seismic-volcanic data from La Palma is not available; there is only a catalog that includes some of the earthquakes detected by human operators during the seismic crisis. Therefore, it would be impossible to conduct a comparative performance analysis of the systems without a reliable reference catalog.

In this regard, we are collaborating with INVOLCAN technicians to create a more comprehensive seismic catalog that will allow us to carry out this experiment. We take this opportunity to invite the reviewer to collaborate with us on this new study. We encourage him/her to contact us, and if she/he has any other databases with reliable annotations, we would be happy to include

**tit in future work. This study would analyze both the LAPALMA dataset and his/her database, using MASTER-DEC and POPO2002 as master datasets within a weakly supervised learning framework.**

Minor remark :

- Title : "catalogus" -> catalogues

**Corrected**

- L.34 "frequency", it is not clear whether you speak of the signal frequency content or of the occurrence frequency.

**Corrected. We were talking about occurrence frequency.**

- L.49-52 : there are too many references. you should develop on a few of them to explain their main results / methods.

**All the references were included to highlight the variety of models developed for recognizing volcano-seismic signals using machine learning approaches. Additionally, we felt it was important not to omit any, as excluding some could create gaps in the discussion. To keep the text concise while maintaining a rigorous state of the art, we decided to include all of them**

- L.54 and following : why is this part in italic?

**This is the most significant challenge when constructing such systems, given the unique nature of the volcanoes and the data. As a result, we aimed to emphasize this challenge.**

- L.83 : Deceptio -> Deception

**Corrected**

- L.118 : "our hypothesis", what are your referring to?

**We introduced our hypothesis earlier, around line 86: 'We hypothesize that, often, automatic recognition systems are not capable of modeling the spatial-temporal evolution of seismic events. Instead, they learn to recognize the probabilistic pattern-matching observed in their training data. In other words, rather than simply learning to characterize volcanic dynamics by describing the latent physical model, catalog-induced learning biases the system's performance as it learns the description of the data annotated in the catalog, potentially discarding useful data that describes volcanic dynamics. Therefore, we conclude that using systems trained with a master database (complete and large) as pseudo-labeler, could help create less biased catalogs from which the systems can be retrained and adapted to different volcanic environments.' This sentence refers to our hypothesis.**

- L.124 : How can you have 8 channels on a three-components seismic sensor? Chat are these channels?

**We believe that the reviewer has not properly understood the sentence describing the sensorization used in the data collection for Deception Island. The text literally states: The Deception Island dataset (hereafter referred to as MASTER-DEC) was created using seismic data collected during the 1994-1995 campaign organized by the Andalusian Institute of Geophysics (IAG) with a short-period array of 8 channels. The array consisted of a three-component Mark L4C seismometer with a lower frequency band of 1 Hz and five Mark L25 sensors with a vertical component frequency of 4.5 Hz, electronically extended to 1 Hz. As can be seen, the array consists of one three-component seismometer and five single-component vertical sensors, adding up to a total of 8 channels.**

- L.144 : "UMAP", give a reference, how did you compute it?

**Included. The application of UMAP approaches are explained in Supplementary material.**

- L.212 : "domain information", what do you mean?

**This section has been rewritten almost in its entirety to enhance understanding and incorporate the mathematical foundations suggested by the reviewer.**

- L.213 : You do not explain what Ys and Yt are.

**This section has been rewritten almost in its entirety to enhance understanding and incorporate the mathematical foundations suggested by the reviewer.**

- L.230 : You have not yet explained what are RNN-LSTM, Dilated-RNN and TCN, you do it only l.264. As stated in my main comments, the Methodology section can be re-organize to avoid this kind of problem.

**Modifying the entire methodology section would conflict with the comments made by the previous reviewers. This sentence has been included in the paragraph to address the lack of information: 'For our experimental framework, we will base our approach on the pre-trained systems previously published in Titos et al. (2018, 2022, 2024). These systems include Recurrent Neural Networks (RNN), Dilated Recurrent Neural Networks (Dilated-RNN), both utilizing LSTM cells, and Temporal Convolutional Networks (TCN). These models, referred to as RNN-LSTM, Dilated-LSTM, and TCN, generate a probabilistic event detection matrix with per-class membership outputs'.**

- L.279 : "three systems", I understand that you refer to RNN-LSTM, Dilated-RNN and TCN trained on the Master datasets, but when first reading the sentence it is not obvious.

**Corrected**

- L.291 : "our initial hypothesis", at this point, the reader may not remember what your initial hypothesis is.

**We have completed the sentence by including a brief reference to our hypothesis: 'To test our initial hypothesis—that automatic recognition systems often fail to model the spatial-temporal evolution of seismic events, relying on probabilistic pattern-matching from training data, which can introduce biases and overlook valuable information about volcanic dynamics—and following…'**

- L.295 : "Each" -> each

**Corrected**

- L.496 : "because the" -> because of the

**Corrected**

- L.512 – 516 "The first row … respectively". This should be in the legend, not in the main text.

**Corrected**

- L.532 : There a missing number after "Figure"

**Corrected**

**Answer to comments of Reviewer#3**

**Dear reviewer#3, We are very thankful for your thoughtful suggestions. Below, we present how we have addressed them.**

The Authors describe the automatic labelling of seismic activity in volcano environment. The manuscript is interesting because it deals with a very current topic, that is, the automatic management of large amounts of data that would require manual work that is very expensive in terms of time and human resources. A solution that has been widely used in recent years uses machine learning techniques, that is, training an algorithm to make decisions by replacing us. The issue at this point becomes that of minimizing the errors made by the algorithm so that the results are reliable. To this end, using a database as a reference point to train a machine learning algorithm that can then be applied to other databases is essential. The Authors declare great knowledge of Deception Island volcano and use the seismic database of this volcano as benchmark for other 2 volcanoes database to build an

automatic machine learning based procedure aimed to recognizing the seismic event type among 5 possible event sources in order to detect pre-eruttive signals. The topic is of great general interest, because the labelling of seismic events concerns modern seismology in general because of the proliferation of increasingly dense seismic networks that collect an ever-increasing amount of data but the paper needs a major revision before publication because Author should do an effort to explain their work in a more concise way.

All sections are too long and repetitive and fail to stress the most important parts of the method and data processing. The manuscript fails to indicate in clearly and concise way the necessary and important parameters used by the (proposed/used it is not clear) methodology and fails to describe the dataset used.

**We agree that some parts of the text are repetitive and lengthy. This is a result of the revision process that was carried out. In the first round of revisions, we had to adapt the manuscript to the suggestions of eight reviewers. Each reviewer proposed improvements in different sections, which extended the text and sometimes made it repetitive. We have attempted to address this by conducting a detailed review and removing sections that do not contribute to the manuscript. However, it is difficult to carry out such an extensive revision with so many reviewers without the text being affected. In the methodology section, we have attempted to clarify which parameters are necessary to implement the algorithm. In this case, there is only one key parameter: the probabilistic membership threshold. Everything else corresponds to training parameters, which are extensively described in the referenced articles.**

Since the Authors state that the code and data are available upon request, it is necessary to show an example of data, to explain how the data is acquired, treated, processed and used, using explanatory figures. The manuscript then fails to describe the software and the dataset to reproduce their results.

**This article incorporates a weakly supervised methodology to the results obtained from systems that have been previously published. The recognition systems used in this work, namely LSTM, Dilated-LSTM, and TCN, are not only referenced in the paper (having been published and analyzed with the same MASTER-DEC dataset), but are also widely known within the scientific community (Hochreiter, S., Schmidhuber, J., 1997; Schmidhuber, J., 2015; Chang et al. 2017; Lea et al., 2017).Therefore, we believe that they do not require a detailed description in the text, but rather just a citation. A similar approach applies to the seismic signal preprocessing pipeline (Titos et al. 2024). We make use of a widely used and well-known pipeline in the scientific community, based on a log frequency scale filter bank.**

The citations are not in the correct position in the text and do not help the reader understand what they refer to. So, it is not clear if authors used a particular software for their automatic labelling or they propose a new software that they wrote

themselves. The data used are not well described together with the construction of the sub-dataset for training. Maybe a schematic sketch of the seismograms' processing can help. The use of acronymous should help the reading but are explained after their use. The language is too qualitative (what do Authors mean for reliability, for acceptance? Did the Author set any thresholds? Which are the parameters used to build the initial sub-datasets? How are results affected by these decisions?) and the reading is in some parts frustrating since it is hard to get to the heart of the problem that is: can we leave these algorithms work alone? If so, which is the uncertainty of the results?

**Citations: we sometimes place them at the end of the text when referring to a general idea, or within a sentence when we are referring to a specific concept within that sentence, as is the case described by the reviewer. We have tried to italicize the cases where the concept within the sentence itself is important, in order to avoid misinterpretations.**

**Automatic labelling: This work proposes a methodology to create robust seismic catalogs while reducing the cost or human effort required for their development. To achieve this, systems trained on a master database are used as labelers to generate a new database, which is then used to retrain these systems and adapt them to a new volcanic environment. In this study, this software (developed by us and previously published) is utilized, and only the code necessary for selecting events—using a weakly supervised approach—is developed in this work to determine which events will be used for system retraining.**

**Data sets: We have used three different databases, all of which have been thoroughly documented. We have provided details on both the databases themselves and the sensorization used to record these data (when such information is available). Regarding preprocessing, we have outlined the general pipeline and referenced articles that describe this process in depth, as it is a widely used and well-known concept in the field. We have attempted to adjust the language to improve readability and comprehension, as well as to organize acronyms and references more systematically. Finally, by "reliability," we refer to the system's robustness in detecting and classifying seismo-volcanic events. Specifically, we use this concept to highlight that the reliability of an automatic recognition system is traditionally assessed by its accuracy across all events in a catalog. A system with an average performance below 75% is generally considered unreliable. However, this often happens not from the system's ability to differentiate between events but from how the catalog is constructed. If seismic categories are not homogeneous and events of different natures are grouped under the same type, system performance declines. Inconsistent categorization undermines learning, leading to recognition accuracy below 70%.**

**Threshold and hyperparameter:** The choice of the threshold is a very important and interesting issue. We are considering writing a new paper analyzing this parameter, as including it in this work would make it excessively long. In this study, we opted for a low threshold to allow the inclusion of a larger number of events in the retraining database in order to demonstrate the usefulness of the method. Choosing a higher threshold would bias the dataset toward events that are almost identical to those in the master database, thereby reducing the system's ability to adapt to the new domain. Therefore, when creating a retraining database, only the threshold needs to be determined. However, the user can also adjust other typical training parameters of neural network-based models, though this is not strictly necessary—rather, it is a user decision to improve convergence or adaptation to the new domain. In this work, none of these parameters are modified. We simply choose the threshold and keep the hyperparameters inherited from the previously trained models fixed.

**Uncertainty:** The uncertainty of the results can be analyzed from two perspectives.

The first is through the detection matrices. If the detection results in probabilistic terms are very high, we can assume that the detected events are reliable. As shown in the attached image, which describes the cumulative distribution function for each event in the master database, in the case of LPEs, for example, 90% of the events are detected with probabilities higher than 85%. A similar pattern is observed for Noise and TRE events. A slightly less robust recognition is only noticeable in the case of earthquakes; however, given their characteristics, the systems still detect them quite effectively.

[Figure]

**The second approach would involve a case-by-case analysis by an expert on the volcano, where each detection and classification would be validated or discarded. However, this is a very costly task, so we rely on the class membership probabilities obtained by the systems, based on the previous argument.**

I think that the manuscript should be rewritten because some parts must be moved in other sections (some examples are reported in the comments' list below). The sections are introduced with phrases that can be eliminated without changing the meaning of the speech. The manuscript should assume a consequential structure aimed at making it clear exactly what the data are, how data are treated and processed, how data are organised, how data are used in what sequence, how the software works and how each choice made previously can influence the subsequent choices and the final results. When a new approach is proposed, it is essential that the comparison between the decision of the algorithm and that of the human seismologist be clear in order to assign a reliable uncertainty to the results.

**As we mentioned earlier, this article was reviewed in its first iteration by 8 reviewers, each with a different background and expertise. Each of these reviewers conducted an exhaustive review, and the authors addressed almost all the changes, which is why the article has changed significantly since its first version. Some reviewers suggested including experiments and results comparing different artificial intelligence approaches or architectures for creating catalogs with different databases, while others suggested expanding the introduction and structuring the discussion and results section. Therefore, we structured the article into 6 sections. The introduction, where we motivate the problem of catalog construction and why monitoring systems might be biased. The second section describes the data and the volcanoes under study. In this section, we describe both the database and the available catalogs. In section 3, we describe the proposed methodology and the experimental framework of the work, where the three experiments conducted to test the robustness of the methodology are outlined. In section 4, we present the results for each experiment. In section 5, we describe and discuss the results of each experiment, and finally, in section 6, we conclude the work. We believe this structure is appropriate for describing a work that is complex to understand for those who are not familiar with these types of techniques. Although the structure suggested by the reviewer could also be suitable, we must maintain a balance between the suggestions from different reviewers, which is why we cannot address such a profound structural change, especially when, in the second review, 2 out of the 4 reviewers suggested only minor changes. What we have done is a thorough reading of the article, removing redundant and repetitive phrases, lightening the text, and making it easier to understand.**

In the following, some non-exhaustive comments:

Line 83 Deceptio instead of Deception.

**Corrected**

Line 140. "For the current study, we extracted a subset of reliable data, consisting of 2,193 seismic events". Can the Authors quantify the meaning of reliable? Do they refer to any quality indicator of location as residuals, hypocentral errors,…

**When we refer to reliable data, we mean that we have selected seismic events that meet prototype standards and that the geophysical experts who have been monitoring the volcano since 1986 agree that these events correspond to the type they have been categorized as.**

Line 144. What are the event parameters that make up the UMAP projection and that are represented graphically? It is not explained so the Figure 2 does not make any sense.

**The text states:** *Figure 2 illustrates the UMAP (Uniform Manifold Approximation and Projection) projection (McInnes et al. 20218), showing the distribution of the five MASTER-DEC event types within the feature representation space.* **The feature representation space refers to how each event analysis window is represented with UMAP, after performing parameterization using the log frequency  scale filter bank, which is the chosen preprocessing pipeline. We have added a sentence in the text that describes how the events have been parameterized using a logarithmic scale filter bank, which represents how the energy of the different events is distributed across the various frequency bands, in order to facilitate understanding.**

**Sentence included: 'The representation space aligns with a log frequency scale filter bank, which captures the energy distribution of each event across various frequency bands. For a more detailed explanation of how the workflow constructs the feature vectors, please review Titos et al.,2024.'**

Table 2. Did the Author classify the seismic events for this paper, or the classification refers to another paper? If they did the classification in this paper, they should explain how they did it. If not, they should refer to the exact citation in the table caption.

**None of the databases used in this work were created here. All the databases are inherited and constructed by experts from the different volcanoes. In the case of Table 2, which presents the events in the MASTER-DEC database, we simply describe some of the event characteristics. The construction of this database was carried out by a team of expert geophysicists with extensive experience in monitoring the volcano. As referenced in the text, they have published a wide range of works providing a comprehensive understanding of the structure and dynamics of Deception Island volcano through numerous campaigns conducted since 1986. The database corresponds to the data collected during the 1994-1995 seismic campaign. This information is already described in the text. In the case of Table 3, which describes the POPO2002 database, the data was analyzed and labeled by experts from that volcano and provided by Dr. Raúl Aránbula for conducting the study and experiments in this work. Finally, the LAPALMA2021 database was provided by Dr. Luca D'Auria, along with the seismic catalog describing the recorded earthquakes obtained by INVOLCAN staff.**

Line 163. "the signals were first filtered to match" do they authors mean 1-20 Hz filter? Please specify in the text.

**This sentence explains that during the data preprocessing, before applying the filter bank, a filter is applied to adjust the sampling frequency of both databases to 50 Hz, ensuring that the obtained feature vectors are comparable. This step is necessary because the Deception Island data corresponds to the**

**1994-1995 campaigns, while the data from POPO2002 and La Palma come from more recent campaigns, where the seismic sensors recorded signals at higher frequencies, such as 100 Hz.**

Line 177. "the inclusion of this use case could be of interest" rephrase.

**Corrected**

Table 3. See the comment to Table 2.

**The same response applies to Table 2. The POPO2002 database was constructed by experts from that volcano during the 2002 seismic campaign. All available information regarding the construction of the database is included in the text.**

Line 179. Same as line 163.

**Corrected**

Line 183-189. The introduction to Section 3 is quite confused. The first sentence is too long and the meaning is lost. Please rephrase. Regarding "once its functioning is understood" do you mean understood by you or by the reader? It is a rude language that insinuates that the reader might not understand. The manuscript needs a thorough rereading and rewriting to be better understood.

**To streamline the reading of the article and avoid redundancy, we have removed this sentence. In the previous version, it was simply a reminder of the initial hypothesis we aimed to test. Regarding "once its functioning is understood," we just wanted to explain that, once the methodology is described, we would proceed with the experiments. This part of the paragraph has also been revised to prevent any misinterpretations.**

**New paragraph: 'This section outlines the methodology and experiments conducted in this work. The proposed algorithm will be described, followed by a detailed explanation of the three experiments conducted. The results of these experiments will be presented in the results section'**

Line 202-206. After 9 pages of introduction, very long and confused, finally the authors start to describe their work. It is not clear to me whether the methodology is proposed or applied because often the references are not in the correct position in the text, that is, at the end of the paragraph to which they refer, but in the middle of the speech as if the sentences were paraphrases. Example: "The goal is to address a domain adaptation task (Kouw and Loog, 2019; Farahani et al., 2021) to reduce the cost of developing a reliable seismic catalog and database for a new given dataset with minimal initial human supervision." Do the citations refer to the entire methodology used, or do they refer only to the execution of that part of the task?

**We agree with the reviewer that the introduction is quite long and sometimes unclear. However, as mentioned previously, this is a result of the suggestions**

**from the 8 previous reviewers, each of whom had a different perspective on the work.**

**Regarding the references, we sometimes place them at the end of the text when referring to a general idea, or within a sentence when we are referring to a specific concept within that sentence, as is the case described by the reviewer. We have tried to italicize the cases where the concept within the sentence itself is important, in order to avoid misinterpretations.**

Line 215. The Authors declare "Such assumptions have important implications" but they spend few word to explain the difference between marginal distribution and conditional distribution. How did the Author choose the events that must belong to the two distributions? Did they compare the results with other choices?

**In accordance with another reviewer's suggestion, this section has been completely rewritten. The meanings of marginal and conditional distribution are now better explained.**

Line 223. "events showing characteristics similar" how the authors identify similarity? Can they quantify this choice? Similarity refers to some characteristics of the seismograms, of the frequency content, of amplitude, of magnitude, of location, of signal length, of coda waves, of body waves,….?

**In the context of this work, when we refer to similar characteristics, we mean events that have similar waveforms and spectral content. Translating this similarity into the feature space, we refer to points in the feature representation space located in nearby regions (Figure 2 in the manuscript). Since, as we have described throughout this letter and the manuscript itself, our analysis windows or frames are parameterized using a filter bank, what we are representing is the energy distribution in each of the bands covered by each filter. Therefore, similar events will occupy similar regions in the feature representation space.**

Line 230. The acronymous are used here for the first time and are not explained. Maybe a citation is needed here?

**According to the comments from other reviewers, this paragraph has been modified, and the references describing the baseline models have been included.**

Line 233. How long is the frame or the window? Does this choice affect the results? And how much?

**In this work, each analysis window or frame has a duration of 4 seconds, with an overlap of 3.5 seconds with the previous frame. The duration and overlap are parameters that can affect system performance. However, this study does not focus on these aspects, as we are using previously published models whose best results were achieved with these encoding characteristics. The**

**adjustment of the window size and the overlapping between them was extensively studied in Titos et al. (2018) and Titos, 2018 (Doctoral Thesis Dissertation).**

Line 260-265. This sentence should come earlier in the text. Section 3.1 is too general and do not help to understand the method. I propose to eliminate it or to include it in the subsequent sections where the methodology is explained.

**Section 3.1 has been partially rewritten. First, we have expanded the formal description of the domain adaptation problem. Second, we have improved the wording of the proposed methodology. Finally, regarding lines 260-265, we have simply included this information earlier in the text, specifically in the paragraph describing the recognition models used in the proposed methodology.**

Section 3.2.1 it is another introduction. Did the Authors mean that they used the approach of Weiss et al.? They should be more concise.

**In Section 3.2, we simply define the first experiment within our experimental framework. The reference to Weiss et al. corresponds to the concept of Transfer Learning itself. Essentially, what we aim to convey in this section is that instead of building a system from scratch, we will retrain existing models using the available data and labels from POPO2002. This approach is known as classical transfer learning.**

Line 320. "This section presents the results supporting the experiments outlined in the previous section" it is obvious. Please avoid these explanations in the text.

**Corrected**

Line 323. Did the Authors compare the automatic results with a manual inspection of the data automatically labelled in order to evaluate the accuracy between the automatic choice and your best accurate human one?

**The results and the accompanying images in the results section, where a detailed analysis is conducted, correspond to the manual inspection referred to by the reviewer.**

Line 361. "The y-axis corresponds to the real label or ground-truth and the x-axis corresponds to predicted labels." Is this the correct labelling of the master dataset to which the results must be compared to?

**The results in Table 7 correspond to the confusion matrix obtained by the different systems, using the manual annotations from the POPO2002 catalog as a reference.**

Line 411. "Once the construction of catalogs through transfer learning has been discussed, we are now ready to discuss the use of weakly supervised pseudo-labeling approaches." As Line 320.

**Removed**

Line 413. I beg to differ with this statement. Results are not clear and comparison between the automatic labelling of the master dataset with the manual labelling is missing or not well explained.

**The sentence the reviewer refers to in this comment is: "Thus, although system performances range between 85% and 90%, this does not always reflect a complete or unbiased seismic catalog. Rather than solely learning to characterize volcano dynamics based on an underlying physical model, the systems may be learning the information contained within the catalog itself. Consequently, catalog-induced learning could limit a system's ability to generalize, potentially obscuring information relevant to advancing our understanding of volcanic behavior." This sentence aims to convey that when a system is trained with a predefined seismic catalog (constructed under specific circumstances and for a particular purpose), the training process itself adapts the way the systems detect and classify different seismic events to minimize errors compared to the catalog annotations. In contrast, when using a pseudo-labeler built from a master database, the system detects and classifies the different events without the implicit human bias. We believe that this conclusion is concise and clear, and does not require a detailed discussion.**

Discussion section. This section is too long and include figures that belong to the results and that can be useful in earlier part of the manuscript. I suggest a deep reorganization of the paper.

**This article follows the structure of others published in this journal and the suggestions of several reviewers. In the results section, we present the outcomes obtained within the experimental framework. In the discussion section, the results are discussed in detail, and both the pros and cons of the methodology are argued. This is why most of the figures are found in this section, as the summarized results in tables are presented in the results section.**

In the manuscript the reference to the figure is sometimes written as Fig. and other times as Figure. Please check.

**Corrected. According to the journal's writing template(as far as we know), when the word "Figure" begins a sentence, it must be written in full. However, when "Figure" is part of a sentence, it can be referenced as "Fig." In any case, since this is a drafting error, we will consult with the journal to correct these issues before the final publication.**

**Answer to comments of Reviewer#4**

**Dear reviewer#4, We are very thankful for your thoughtful suggestions. Below, we present how we have addressed them.**

**General Comments**

In this second iteration of the manuscript, the authors have clearly devoted significant effort to refining and restructuring their work. The result is a substantially improved document that showcases clearer objectives, methods, and outcomes. Across all sections, the organization and writing style have been noticeably enhanced, making the overall manuscript much more coherent and accessible.

**The Introduction** is particularly strong, providing both a concise background and a clear statement of the research motivation. In addition, Section 2, which focuses on seismic signals and data catalogs, has been reconstructed in a way that captures the essential details of catalog construction and usage. This section now offers a thorough explanation of how seismological data is collected, cataloged, and analyzed, setting a solid foundation for the subsequent methodological discussion.

**The Methodology** section has also undergone a marked improvement compared to the previous version. The authors' decision to outline each step more systematically—especially how the three experiments are structured—makes it much easier for readers to follow the logic and replicate the work. Notably, the emphasis on **pseudo-labeling** as part of their weakly supervised learning strategy deserves commendation. By using a pre-trained model as a pseudo-labeler and then re-training with the newly labeled data, they demonstrate an innovative approach to semi-supervised or weakly supervised classification in seismo-volcanic signals.

**Regarding the Discussion**, one of the central points the authors address, which is particularly interesting for the field, is the relatively low recognition rate compared to existing reference catalogs. They offer a plausible explanation that these catalogs, while established, may be incomplete or biased toward particular classes of events. Consequently, a strict comparison against them can underestimate the efficacy of the new system.

Along the same line, the authors highlight the **quality vs. quantity** dilemma. While the weakly supervised methodology might introduce some degree of noise or misclassification, it also increases the overall number of detected events, thus expanding the catalog. According to their description, it would be ideal for future users of this methodology to strike a balance by conducting manual checks on a fair portion of newly labeled events to verify their authenticity. These checks not only help mitigate the risk of accumulating errors from pseudo-labels but also lend credence to the claim that genuinely overlooked events are being discovered. Nevertheless, **we recommend** that the authors (and future users) **explore**

**additional statistical consistency checks and cross-comparison with alternative detection methods** to further strengthen the reliability of these expanded catalogs in subsequent research projects. By systematically verifying or filtering pseudo-labeled events—through model agreement, confidence thresholds, statistical checks, and domain-expert reviews—one can reduce the risk of error accumulation and improve the quality of the final training data.

From a contextual usefulness standpoint, the authors argue that any additional events— correctly identified or carefully verified—enrich our understanding of volcanic processes, potentially offering earlier or more nuanced insights into volcanic unrest. They stress that while it is important to measure success against established reference catalogs, it is equally crucial to recognize the value in uncovering smaller or subtler events that might have gone undetected.

As a result, even if the system does not perfectly align with existing catalogs, it may enhance real-time monitoring, inform hazard assessments, and ultimately lead to more comprehensive research in volcano seismology.

Nonetheless, further elaboration on the potential pitfalls of pseudo-labeling, along with additional quantitative or expert-driven validations, would strengthen the overall argument.

Despite these minor weaknesses, this manuscript now provides a valuable contribution to the application of machine learning within volcano seismology. The authors' demonstration of how to construct and refine catalogs, leverage pre-trained models, and evaluate performance across multiple experiments will be extremely useful in guiding future research. Overall, the revision is a notable success, and the text should serve as a new reference for continued advances in the automated recognition and analysis of seismic-volcanic signals.

**About very Minor writing issues:**

• Introduction.

- line 52: A period "." is missing after "etc".

**Corrected.**

- line 54: Maybe lose instead of loss?

**Corrected.**

- line 54: an interesting topic: "monitoring systems loss effectiveness when recognizing events over time.." it would be ideal to include some references to support this point.

**Corrected. We have included references to two of our articles where we tested how the systems perform when using data from the same volcano, obtained from different seismic campaigns.**

- line 83: Deception misspelled.

**Corrected.**

- line 108: there is an extra period ".".

**Corrected.**

- line 110: "volcano" misspelled.

**Corrected.**

• Seismic data and catalogs.

- line 123: as stated in fig.1., the data was also collected in 1996 and 2001-2002?

**Corrected. It was a drafting error. Dato was collected in 1994-1995.**

- lines 150 and 151: are we using "Popocatépetl" with or without an accent?

**Corrected.**

---

## Author Response (AR3)

**Jun 12, 2025**

**Editor-in-Chief**

Natural Hazards and Earth System Sciences

Dear Editor,

I am pleased to submit this cover letter regarding the original research article (nhess-2024-102) entitled "Could seismo-volcanic catalogues be improved or created using weakly supervised approaches with pre-trained systems?" by Titos M., et al., for consideration in NHESS. We have carefully reviewed the feedback from reviewers and greatly appreciate the time and effort they have invested in evaluating our work.

We hope that these revisions, alongside the provided documentation of changes, meet the reviewers' expectations and adequately address their feedback.

Thank you for the opportunity to improve our work. We look forward to your advice.

Yours sincerely

Manuel Marcelino Titos Luzón
Postdoctoral Researcher, University of Granada, Spain
mmtitos@ugr.es.

**ANSWER TO THE REVIEWER'S COMMENTS**

In the following, we have provided detailed answers to the comments of the reviewers. The original texts from the reviewers are in normal font. Our answers are in bold font. We would like to take this opportunity to thank the reviewers for their valuable comments and for their time and resources.

**Answer to comments of Reviewer#1**

**We would like to thank reviewer#1 for the careful reading of this manuscript. Furthermore, below are those comments that need more clarification.**

Insufficient data have been used for any useful scientific inference.

**The major revision primarily focuses on the inclusion of more data. We would like to respectfully clarify that the primary objective of our manuscript is not to introduce a new model architecture. Rather, our work focuses on leveraging previously published and operational models as pseudo-labelers within a weakly supervised learning framework. This approach is intended to support the generation of seismic event catalogs with reduced manual effort—an aspect we consider central to the contribution of the study.**

**We fully acknowledge the value of the reviewer's suggestion to evaluate the approach on additional datasets. However, this is only feasible when reliable labeled data is available for comparison. In practice, the creation of high-quality seismic catalogs is a time-consuming and resource-intensive task, and such labeled datasets remain scarce. This challenge further motivates the goal of our work.**

**In this work, we have used the two labeled datasets that are currently available to our research group. Additionally, we contacted one of our collaborators, who kindly provided a dataset from a volcanic crisis along with its earthquake catalog. At this point, we can only express to the reviewer that we are open and willing to apply our method to any reliably labeled dataset they can recommend or share with us in order to fulfill their suggestion.**

**This is the third time we have been assigned a "major revision," and unfortunately, we do not have access to additional data. Seismic observatories are often reluctant to share their datasets, and when they do, the data is typically not labeled. It is simply not feasible for us to continue addressing this request without access to the necessary data.**

**Therefore, we kindly ask the reviewer to guide us on where and how we can obtain such datasets in order to carry out the suggested analysis. Without this, it will not be possible for us to further address this concern.**

**Answer to comments of Reviewer#2**

**We would like to thank reviewer#2 for the careful reading of this manuscript and the thoughtful comments that have improved the quality of this manuscript. Furthermore, below are those comments that need more clarification.**

Detailed Review Report
*General Impressions*
The authors clearly propose that traditional machine learning models for seismic event detection often carry biases due to training on specific, limited catalogs. Their approach, utilizing pseudo-labeling based on pre-trained systems to enhance model generalization, is compelling and well-motivated. The manuscript presents a highly relevant and interesting investigation into improving seismic-volcanic catalogs through weakly supervised machine learning techniques.

After multiple rounds of review, significant improvements have been made. However, readability and methodological clarity remain core concerns. Below, I outline detailed recommendations to address these issues constructively.

*- About Methodology Section.*
• Section 3.1 (Methodology Clarity):
The manuscript currently states that the proposed method aligns with the open-set domain adaptation paradigm, explicitly designed to handle novel event categories. However, the authors subsequently note a significant limitation: the method only labels events within categories already present in the master database. This limitation appears to directly contradict the previously stated open-set capability. I recommend clarifying this contradiction explicitly.

The authors should specify whether the approach is truly open-set (capable of detecting and handling unseen seismic categories) or acknowledge clearly that it is currently limited to closed-set scenarios.

Although the assumptions clearly indicate that label spaces may only partially overlap, and thus novel categories could be present, the authors later explicitly state their methodology can only label categories that exist in the master database. Therefore, the authors should clarify how their approach practically handles (or does not handle) the novel categories mentioned in their assumptions. If the method currently doesn't handle these novel categories, explicitly stating this limitation and distinguishing clearly between the theoretical scenario and the actual method implementation would strengthen the manuscript.

**We appreciate the reviewer's observation. We agree that there is a tension between the theoretical framing of our work within the *open-set domain adaptation* paradigm and the actual capabilities of the implemented method. Since we base the creation of new catalogs on a weakly supervised technique using models previously trained on master catalogs, what we are implicitly performing is a knowledge transfer between domains. It is true that across domains (volcanic environments), classes not present in the training catalogs may appear. This is why we acknowledge that the technique has limitations and to develop a more universal pseudo-labeler, a master database containing a broader range of seismic categories would need to be constructed. However, by examining the probability matrices, it is possible to identify when and**

**where such events appear. To conduct a comprehensive analysis of all events, regardless of whether they are included in the training catalogs, an unsupervised approach would be required—something that falls outside the scope of this article. Nevertheless, we have added a sentence in the manuscript clarifying this limitation:**
**"Although our method only labels categories present in the master catalog, potential novel classes in the target domain may still be revealed through analysis of the probabilistic detection matrices, especially when combined with unsupervised techniques for event discovery"**

• Main issues in the Methodology (Experiment 3.2.1):
1) Insufficient methodological details to reproduce the experiment
      o Problem: Currently, the authors only briefly mention:
- Three model architectures (RNN-LSTM, Dilated-LSTM, TCN),2)
- Pre-training on MASTER-DEC, then re-training with POPO2002

      However, readers might ask:
- How were the models pre-trained initially (hyperparameters, training set sizes, epochs, loss functions, etc.)?
- What specific transfer learning strategies were applied (e.g., layer freezing, fine-tuning, learning rate adjustments)?
- How exactly were data split (train-validation-test)?
- What were the evaluation metrics or validation methods?
- Did authors address class imbalance or category distribution?

Without these details, readers cannot reproduce the experiments. It seems that some important information about this is in section 4 (results). We suggest to the authors that change the text to the methodology section.

**This has been one of the major challenges the manuscript has faced. Given the large number of reviewers, maintaining a clear structure has proven impossible. Some reviewers requested that the methodology remain free of implementation details and suggested including those details in the results section. Others, noting that the manuscript does not present the development of a new model, recommended simply referencing the articles where each model's technical details are described. Regarding the transfer learning techniques, what we perform is a fine-tuning of all the parameters of the architectures in order to adapt each pre-trained model to the specific volcanic environment.**

**We have included the following sentence in Section 3.1 (Methodology – Re-training): "This approach applies a transfer learning strategy in which all model parameters were fine-tuned, experimenting with different learning rates and regularization techniques, and employing early stopping to prevent overfitting.**

2)Confusion caused by two alternatives, stating:
      o Problem: The authors propose two alternatives, stating:
- Option A: Only use the 5 categories in common with the MASTER-DEC catalog.
- Option B: Adapt the model output to accommodate all 7 categories present in POPO2002 by updating the output layer only.

But, critically:

- Authors state vaguely: "these two approaches have no major implications from a ML perspective".
- This statement is confusing because, practically, these two options have very different implications:

Option A completely excludes new categories, simplifying the task significantly. Option B involves at least minor model changes (output layer modification), and crucially, implies retraining with novel data categories (a clearly significant ML implication). This confusion significantly weakens methodological clarity.

**We appreciate the reviewer highlighting this point, as it is indeed important. When we mention that switching between 5 and 7 classes has no major implications, we are referring specifically to the fact that, in terms of performance, model complexity, and number of parameters, both configurations are practically equivalent. In both cases, the models retain the parameters of all layers except for the output layer, which increases from 5 to 7 units. The retraining process remains the same, with the only difference being that the model is now trained to recognize 7 classes instead of 5.**

**We have changed the text and included this sentence for the sake of the clarity: From a ML perspective, both approaches follow standard procedures, although they differ slightly in implementation. In the first case, where only five seismic categories are considered, the models are fully retrained using the new catalog. In the second case, which includes seven categories, the output layer is modified to accommodate the two additional classes, while the pre-trained parameters from the original model are retained. The model is then fine-tuned on the new data, allowing efficient adaptation without retraining from scratch.**

• Second Experiment (3.2.2):
Clearly distinguish the novelty of Experiment 2 by explicitly stating upfront that the primary difference from Experiment 1 is the source of labels (pseudo-labels rather than true annotations). Emphasizing this difference early in the description would enhance readability.

**We have included this sentence at the beginning of the paragraph to enhance readability: "The second experiment differs from the previous one primarily in the source of the labels used for training: instead of relying on true annotations, it leverages pseudo-labels generated by the pre-trained models themselves".**

- About Results Section.
• Lines 334-357: This methodological detail is beneficial and should be moved explicitly into the methodology section for clarity and improved reproducibility.

**As we previously mentioned, this information is placed in this section following the suggestion of earlier reviewers. Given the large number of reviewers who have revised the manuscript, it has been very challenging to establish a clear structure, and we have had to continuously relocate parts of the text between different sections. We kindly ask the reviewer that, if they are not comfortable with keeping this content here, to suggest again where it should be placed, and we will do our best to**

**accommodate the change. However, we believe that implementation details are better presented alongside the results rather than in the methodology section**.

• Line 362: The statement regarding "two experiments" conducted at this stage is confusing and should be simplified in the methodology section.

**What we mean by this statement is that we have conducted two separate experiments, one with 5 classes and another with 7. This is entirely independent of the methodology. Our intention is simply to demonstrate that, when retraining the models using the labels from a given database, the performance results remain high regardless of the number of classes**.

• Should the reader be beneficed with more transfer learning details? (e.g., fine-tuning strategies, freezing layers explicitly, loss functions and training epochs?).

**We have included new sentences along the manuscript to enhance de readability of the proposed technique.**

• First Experiment Results:
While the overall self-consistency result (e.g., 77.38% accuracy for the RNN-LSTM model) provides a general sense of model performance, the confusion matrix reveals important class-specific differences—most notably, the relatively low recall for VT events (0.51) compared to much higher values for noise (0.97) and other event types. This suggests that the model may be biased toward the dominant class (likely noise), potentially inflating the global performance metric. I recommend that the authors include additional evaluation metrics, such as precision, recall, and F1-score for each class, as well as macro-averaged or balanced accuracy scores. These would provide a more nuanced understanding of how well the model generalizes across all event types, especially the underrepresented or more challenging classes like VT. Including this information would strengthen the assessment of the model's real-world applicability in diverse seismic scenarios.

**Table 5 refers to the best results obtained in Table 4, which were achieved using a training split of 40% of the total data and considering 5 seismic classes. That said, the reviewer's analysis makes a lot of sense; however, we did not incorporate it into the manuscript because, in this case, weighted results or metrics such as the F1-score do not provide much additional information.**

**If we analyze Table 3 in detail, which is the basis for the retraining process, it shows that the number of VTE events — which are later the worst recognized — is 371, while BGN, one of the best-recognized classes, has 340 events. As can be seen, there is no strong imbalance. However, considering the nature of the events themselves and the spectral description given by the parametrization scheme used (based on filter banks on a logarithmic scale), noise-type events are much easier to discriminate than VTE events. As can also be seen in Figure 2, VTE events share regions of the**

representation space with many other event types, which complicates their recognition.

Finally, Figure 6 is another clear example of the complexity involved in detecting VTE events. Some of the VTE events labeled in POPO2002 do not consistently share spectral characteristics with VTE in MASTER-DEC. This may be because catalogs are often built using data from multiple seismic stations, with strong attenuation and source effects, as well as rules or conditions imposed for signal identification.

Therefore, the original labeling of an event does not always match the spectral content and waveform of the analyzed signal, since it may vary depending on the station being analyzed. As a result, if the analyzed signal does not align with the characteristics of the prototype event used to build the system, it will be labeled or associated with the prototype that probabilistically most resembles it. This behavior reduces the recognition rate for this seismic category.

Precision, Recall, and F1 Score for RNN-LSTM architecture

| Class | Precision | Recall | F1 Score |
|-------|-----------|--------|----------|
| BGN | 0.836 | 0.97 | 0.898 |
| TRE | 0.780 | 0.78 | 0.780 |
| HYB | N/A | 0 | 0 |
| VTE | 0.864 | 0.51 | 0.640 |
| LPE | 0.680 | 0.85 | 0.750 |

Precision, Recall, and F1 Score for Dilated-LSTM architecture

| Class | Precision | Recall | F1 Score |
|-------|-----------|--------|----------|
| BGN | 0.768 | 0.96 | 0.855 |
| TRE | 0.650 | 0.69 | 0.669 |
| HYB | N/A | 0 | 0 |
| VTE | 1.000 | 0.31 | 0.473 |
| LPE | 0.565 | 0.78 | 0.654 |

Precision, Recall, and F1 Score for TCN architecture

| Class | Precision | Recall | F1 Score |
|-------|-----------|--------|----------|
| BGN | 0.766 | 0.98 | 0.863 |
| TRE | 0.819 | 0.68 | 0.742 |
| HYB | N/A | 0 | 0 |
| VTE | 0.819 | 0.59 | 0.688 |
| LPE | 0.735 | 0.86 | 0.791 |

The weighted precision, recall, and F1-score are calculated by taking into account the number of events in each class as weights. This method adjusts for class imbalance by assigning more importance to classes with more samples. The weighted metric is computed as the sum of each class's metric multiplied by its number of events,

divided by the total number of events across all classes. This gives a more representative overall performance metric that reflects the actual distribution of the data.

$$\text{Weighted Metric} = \frac{\sum_{i=1}^{C} N_i \cdot M_i}{\sum_{i=1}^{C} N_i}$$

**C: total number of classes**

**Ni: number of events in class ii**

**Mi: metric value (e.g., precision, recall, F1-score) for class ii**

Weighted precision, recall and F1-score for each class using the TCN architecture.

| Class | Precision | Recall | F1-score |
|-------|-----------|--------|----------|
| BGN | 0.98 | 0.98 | 0.98 |
| TRE | 0.68 | 0.68 | 0.68 |
| HYB | 0.00 | 0.00 | 0.00 |
| VTE | 0.59 | 0.59 | 0.59 |
| LPE | 0.86 | 0.86 | 0.86 |

Weighted precision, recall and F1-score for each class using the Dilated-LSTM architecture.

| Class | Precision | Recall | F1-score |
|-------|-----------|--------|----------|
| BGN | 0.96 | 0.96 | 0.96 |
| TRE | 0.69 | 0.69 | 0.69 |
| HYB | 0.00 | 0.00 | 0.00 |
| VTE | 0.31 | 0.31 | 0.31 |
| LPE | 0.78 | 0.78 | 0.78 |

Weighted precision, recall and F1-score for each class using the RNN-LSTM architecture.

| Class | Precision | Recall | F1-score |
|-------|-----------|--------|----------|
| BGN | 0.97 | 0.97 | 0.97 |
| TRE | 0.78 | 0.78 | 0.78 |
| HYB | 0.00 | 0.00 | 0.00 |
| VTE | 0.51 | 0.51 | 0.51 |
| LPE | 0.85 | 0.85 | 0.85 |

**As a conclusion, we can say that the recognition of VTE is not so much influenced by the class imbalance of the dataset, but rather by the complexity of an event that shares spectral features with others throughout its temporal evolution.**

• Second Experiment Results:
While the weakly supervised fine-tuning improved global accuracy, the model's ability to detect meaningful seismic events—especially VT and LP types—remains limited, with VT nearly absent in the confusion matrix. The dominance of the noise class likely inflates the global metric. Additionally, the model's detection rate far exceeds the label count, which may reflect over-sensitivity rather than true discovery. More rigorous evaluation, including precision-recall analysis, event-level validation, or expert review of excess detections, would strengthen confidence in the weak supervision pipeline.

**We agree with the reviewer that validating the detected events or conducting an exhaustive expert review would increase confidence in the detections. However, this task is unfeasible from a human standpoint due to the large number of detections generated by the different architectures.**

**We would like to emphasize that Table 7 does not indicate that the models are incapable of detecting VTE and LPE, for instance. In fact, Table 8 shows the number of detections each architecture makes for each event type, confirming that there are many more detections than those reflected in the original catalog. What Table 7 indicates is that the detections of these events do not match with the catalog labels.This is why we performed an error analysis and provided graphical examples to support the reported recognition percentages.**

**Furthermore, to test the reliability of the results, we also conducted experiments on La Palma database where only earthquake events are annotated. We compared both matching and non-matching annotations in order to evaluate sensitivity and the false positive rate for this type of event. By analyzing Figure 9, we conclude that the vast majority of detected events match those in the catalog, considering them well recognized.**

• Third Experiment Results:
The use case of applying weakly supervised models during a pre-eruptive crisis is compelling and highlights the practical value of such approaches. However, the presentation of results—particularly the so-called "recognition results" table—is unclear. It is not evident whether the numbers reflect validated detections, raw counts, or comparisons to any ground truth. The sudden introduction of PhaseNet, while relevant, is also only partially integrated, with no evaluation metrics provided to contextualize its outputs or compare them to the proposed models. A more transparent and consistent presentation of results, including quantitative comparisons, ground truth validation, and clearer labeling of what each table or number represents, would greatly improve the interpretability and impact of this section.
While the discussion highlights VT confusion rates exceeding 60% in some cases, this appears to reference only the worst-performing model (Dilated-LSTM). The other models achieve higher recall (e.g., 59% for TCN), and the average across all three models is closer to 47%, not 40%. A more balanced summary would acknowledge this range to accurately reflect performance variability across architectures.

**We have modified the caption of Table 9 to indicate that it shows the number of earthquakes recognized at each station. As we argue in the text, the seismic catalog for this dataset contains 247 events. This study can be divided into two experiments. On the one hand, we analyze how many events are detected by PhaseNet, an AI-based model for seismic phase detection at each station, and on the other hand, how many are detected by our approach. For the PhaseNet experiment, we include the result tables and figures, and we also provide an analysis highlighting the cases where catalog events and detected earthquakes match (Figure 8b).**

**In the case of our models, we conduct a visual analysis of the results and present a discussion in the text to avoid including additional images and to keep the discussion more concise:**
**"Regarding the detection of events identified by the systems but not annotated in the catalog, on average, RNN-LSTM and Dilated-LSTM detected approximately 60 earthquake-type events, while TCN identified over 150. Figure 11 presents a couple of examples of such earthquakes. The PSDs reveal that they share characteristics consistent with those of earthquakes. However, as indicated by the probabilities shown at the top of the figure, their partial similarity in spectral content prevented them from being classified with higher confidence."**

**As we previously mentioned, conducting a thorough analysis of the results and maintaining a readable structure in the article has been challenging due to the large number of outputs. We believe that the way the results are currently organized is the simplest and most efficient way to analyze, compare, and discuss them.**

Summary about results:
Throughout the results and discussion sections, the manuscript refers to "confusion matrices" and reports numerical values (e.g., 0.51, 0.31, 0.59 for VT events across models) without clearly stating whether these represent recall or confusion rates. However, the structure of the matrices—particularly the fact that each row sums to one—strongly suggests that the values correspond to per-class recall, i.e., the proportion of correctly classified instances for each true class. This is the standard interpretation for row-normalized confusion matrices in the machine learning literature. The ambiguity around this point makes the discussion difficult to follow and may contribute to the impression of poor presentation. For instance, the statement that "confusion rates exceed 60%" appears to refer to only the worst-performing model and does not align with the higher recall values seen in other models unless the reader assumes a confusion rate = 1 - recall. For the sake of clarity and consistency, it is essential that the manuscript explicitly define how these matrices are computed and what the reported values represent. This will not only improve readability but also help readers interpret the results accurately.

**We have added the following sentence in Section 4 (Results), describing how the confusion matrices were constructed and what they represent: "For each experiment, tables describing the system performances in terms of accuracy, along with detailed confusion matrices are presented. These confusion matrices were constructed by comparing the model predictions against the labeled events in the catalog. This approach allows for a granular analysis of the classification behavior, revealing not**

**only the global accuracy but also class-specific performance, misclassification patterns, and possible confusion between seismic event types".**

While the qualitative example shown in Figure 4 is compelling and suggests the model is capable of discovering events missed during the initial labeling, these anecdotal demonstrations are not sufficient to validate the effectiveness of the weakly supervised system. To move beyond suggestive visuals and convincingly argue for the scientific value of these new detections, the study would benefit from a more rigorous validation approach—such as expert review, waveform similarity analysis, or cross-comparison with independent models like PhaseNet. Without such steps, the claim that these new detections are not false positives remains speculative and limits the broader impact of the proposed method.

**We fully agree with the reviewer. As we have previously argued, validating the detected events or conducting an exhaustive expert review would indeed increase confidence in the detections. However, this task is unfeasible from a human standpoint due to the large number of detections generated by the different architectures. This is precisely why we included the experiment and comparison with PhaseNet. It should be noted that PhaseNet cannot be used for comparison with other types of events beyond earthquakes, as it would not be meaningful. Moreover, we cannot compare segmentation performance since PhaseNet only detects P and S phases. Therefore, our analyses are necessarily limited to comparisons with the available labeled data.**

**We kindly ask the reviewer, if they are aware of any master and reliable annotated database to share it with us. We would be happy to run the experiments and provide a full analysis of the results accordingly.**

- About the Discussion Section:

• Line 416: Verify if percentages presented in the discussion exactly match the results section; discrepancies would confuse readers.

**Corrected**

While the qualitative example shown in Figure 4 is compelling and suggests the model is capable of discovering events missed during the initial labeling, these anecdotal demonstrations are not sufficient to validate the effectiveness of the weakly supervised system. To move beyond suggestive visuals and convincingly argue for the scientific value of these new detections, the study would benefit from a more rigorous validation approach—such as: expert review, waveform similarity analysis, or cross-comparison with independent models like PhaseNet (we'll talk about this later). Without such steps, the claim that these new detections are not false positives remains speculative and limits the broader impact of the proposed method.

**We completely share the reviewer's view. As discussed earlier, verifying the detected events or carrying out a thorough expert validation would certainly enhance the reliability of the results. However, given the sheer volume of detections produced by the various architectures, such an undertaking is not practically feasible. For this reason, we incorporated the experiment involving PhaseNet as a point of comparison. It is important to clarify that PhaseNet is specifically designed for detecting seismic phases in earthquake signals and is not suitable for evaluating other event types. Additionally, a direct comparison in terms of event segmentation is not applicable, as PhaseNet only identifies P and S phase arrivals. Consequently, our evaluations are necessarily confined to comparisons against the labeled data available in the catalog.**

The discussion attributes the weak performance of the model on volcano-tectonic events (VTEs) to discrepancies in labeling criteria, subjective annotation boundaries, and prototype mismatches. While labeling inconsistency is a known challenge in volcano seismology, VTEs are typically among the most well-defined and reliably detectable seismic signals due to their impulsive, high-frequency nature. Numerous existing models (e.g., PhaseNet) have shown robust detection of such events across different volcanoes. The fact that the system recovers only 5% of annotated VTEs suggests that the problem may lie more in the modeling strategy or prototype selection than in catalog inconsistency alone. A more balanced discussion should consider whether the weak supervision framework fails to generalize to realistic variability within VTEs and whether model or prototype refinement could improve performance.

**We also share the reviewer's conclusion and agree that many of the labels in the catalog were likely assigned based on information derived from multiple seismic stations. However, when analyzing a single vertical-component signal from a distant station—where attenuation and propagation effects are present—the waveform may not exhibit the typical spectral or shape characteristics associated with that event type. As a result, the system may fail to recognize it. This is supported by the La Palma experiment, where the system successfully detects earthquakes when the signals are clear.**

The comparison with PhaseNet in the third experiment raises concerns regarding methodology.
The authors assess PhaseNet's performance by comparing the number of detected phases across different score thresholds, arguing that only detections above 0.8 correspond well with the labeled dataset. However, this approach overlooks the fact that many valid seismic picks— especially low-amplitude or emergent phases—often have lower phase scores (e.g., 0.3–0.6), yet still align with cataloged arrivals. Furthermore, raw pick counts do not constitute a meaningful evaluation metric unless aligned with ground truth picks using a timing tolerance. To make a valid comparison, the authors should report precision, recall, and pick timing accuracy against the labeled dataset across multiple thresholds. Without this, the argument that PhaseNet underperforms is not well supported and may misrepresent the model's actual capabilities.

**We believe we may not have conveyed our point clearly enough. What we intended to express is that setting the probability thresholds above 80% greatly reduces the**

**number of detected phases, not that the detected events correspond only to phases with probabilities above 80%. The text literally states:**

*"For example, for values close to 80%, only approximately 722 P-phases and 503 S-phases at PLPI; and 282 P-phases and 216 S-phases at PPMA are detected. This significantly reduces the number of potential events that could be included in the catalog. Figure 8b shows the match between detections and the cataloged events. Of these 247 annotated events, PhaseNet detects 206 P-phases and 199 S-phases at PLPI; and 157 P-phases and 28 S-phases at PPMA, all without applying any probability threshold. Again, when setting the phase score threshold greater than or equal to 80%, the detections decrease to 163 P-phases and 164 S-phases at PLPI, and 116 P-phases and 21 S-phases at PPMA."*

**This perfectly aligns with the reviewer's conclusion that many phases have low scores. We think there may have been a misunderstanding regarding this point. Finally, Figure 8b presents the PhaseNet results, taking into account the timing of the P- and S-wave picks, exactly as the reviewer suggests. This comparison allows us to evaluate how well the detections align with the cataloged arrivals based on their temporal correspondence.**

- About Summary of Findings Section:
• Figure 10: Clearly label differences between rows 3 and 4.
**Corrected**

• Ensure consistent PSD plotting style across Figures 10, 11, and 12 for clarity.

**Figures 11 and 12 display a different PSD style because the objective of Figure 10 is specifically to allow the reader to easily identify visual differences between the smoothed PSD shape of the average and that of the analyzed event. This stylistic choice enhances the clarity of the comparison and supports the interpretation of the spectral content.**

**Other (Very) Minor Remarks:**
- Abstract:
• Lines 2 & 4: avoid unnecessary repetition of word "however" within the same paragraph; consider synonyms or rephrasing to improve readability.
**Corrected**
- Introduction:
• Line 37: a space is missing, "..crises.However"
**Corrected**
• Line 51: there is an extra space; "Canario et al., 2020 ;"
**Corrected**
• Lines 57-63: suggestion: another challenge is that upgrades and updates to seismic instrumentation over decades complicate the review of historical seismicity, as the digital signals may not share a consistent framework.
**Corrected**
• Lines 56, 66, 71, 95, etc.: There are inconsistencies in citation formatting throughout

the manuscript. Please ensure that references within parentheses follow the standard format, e.g., "(Weiss et al., 2016)2, rather than "(Weiss et al. (2016))".

**We are following the template guidelines, and if the manuscript is accepted for publication, we will coordinate with the editorial team to ensure the correct formatting. We are unsure why, despite adhering to the guidelines, the references are not being cited correctly**.

• Lines 168: space missing at "MASTER-DEC(1-50HZ)"

**Corrected**

- Methodology and experimental framework:

• Line 247: repetitive vocabulary again (stream).

**Corrected**

- Discussion.

• Lines 445-446: review grammar of "According to such table, on average, only 5% of the analysis windows labeled as VTE in the original catalog were recognized by the retrained systems."

**Corrected**

• Line 473: please check the text "(Fig. 7)a."

**We did not find any error in the text.**

---

## Author Response (AR4)

**July 16, 2025**

**Editor-in-Chief**

Natural Hazards and Earth System Sciences

Dear Editor,

I am pleased to submit this cover letter regarding the original research article (nhess-2024-102) entitled "Could seismo-volcanic catalogues be improved or created using weakly supervised approaches with pre-trained systems?" by Titos M., et al., for consideration in NHESS. We have carefully reviewed the feedback from **all two reviewers** and greatly appreciate the time and effort they have invested in evaluating our work.

We hope that these minor revisions, alongside the provided documentation of changes, meet the reviewers' expectations and adequately address their feedback.

Thank you for the opportunity to improve our work.

Yours sincerely

Manuel Marcelino Titos Luzón
Postdoctoral Researcher, University of Granada, Spain
mmtitos@ugr.es.

**ANSWER TO THE REVIEWER'S COMMENTS**

In the following, we have provided detailed answers to the comments of the reviewers. The original texts from the reviewers are in normal font. Our answers are in bold font. We would like to take this opportunity to thank the reviewers for their valuable comments and for their time and resources.

**Answer to comments of Reviewer#1**

**We would like to thank reviewer#1 for the careful reading of this manuscript and the thoughtful comments that have improved the quality of this manuscript. Furthermore, below are those comments that need more clarification.**

Please comment on how volcano-specific the results are, and how relevant they are to other volcanoes.

**To clarify this important aspect of the methodology and the method's applicability, we have modified Section 5.4 of the manuscript. In the revised version, we address the specificity and relevance of the master database in shaping the final results. We have included the following explanation: "One of the main strengths of the proposed system is its ability to recognize previously learned prototype events, even in volcanic environments that differ significantly from those present in the training datasets. This feature enhances its usefulness in reducing biases when creating or updating catalogs. The results suggest that training on a broader variety of volcanic settings with diverse event prototype distributions could improve recognition performance, fostering the development of more generalizable and less biased catalogs. Nonetheless, the system also presents limitations. Since the pseudo-catalogs are generated using models trained on a fixed set of known seismic categories, the system is forced to assign one of these categories to each analyzed window, even when the event does not match any known prototype. This constraint can lead to the mislabeling of truly novel events and, consequently, affect the performance of systems retrained using such pseudo-labels. Addressing this issue would require the creation of more comprehensive master databases that incorporate a wider range of event types, ideally from multiple volcanic settings. Moreover, determining the appropriate membership threshold for including events in the new pseudo-catalogs remains a key challenge. Low thresholds may increase sensitivity but also introduce many false positives—events that are dissimilar to any known prototype. Retraining the systems with these catalogs could reduce performance and detection accuracy. High thresholds, on the other hand, may improve specificity but may not be sufficient to allow the system to adapt to the new volcanic environment. This trade-off highlights the importance of post-analysis tools that assess detection confidence, which, in**

**addition to offering insights into the presence of potentially novel classes not covered by the original training data, also contribute to evaluating the reliability and effectiveness of the domain adaptation process by revealing how well the system distinguishes between learned and unfamiliar patterns in new volcanic environments—that is, how volcano-specific the results are and how relevant they may be to other volcanoes."**

**Answer to comments of Reviewer#2**

**Dear reviewer#2, we sincerely appreciate your thorough evaluation of and the valuable suggestions, which have significantly contributed to enhancing the clarity and quality of the manuscript.**

The authors have addressed the major concerns raised in previous review rounds, providing clarifications, methodological adjustments, and additional analysis that substantially improve the clarity and scientific value of the manuscript. The experiments are now better contextualized, the performance metrics have been calculated as requested, and the discussion reflects a more balanced interpretation of the model's strengths and limitations. With these revisions, I believe the manuscript is suitable for publication and will be a valuable contribution to the field of machine learning in seismology.

As a final recommendation, I suggest that the authors consider including the precision, recall, and F1-score metrics—either in the appendix or as supplementary material—to enhance transparency and allow for easier comparison with related studies.

**In this revised version of the manuscript, the authors have included F1- score metrics in the supplementary material.**